

# LPJ-GM 1.0: Simulating migration efficiently in a dynamic vegetation model

Lehsten, Veiko[a,b], Mischurow, Michael[b], Lindström, Erik[c], Lehsten, Dörte[b], Lischke, Heike[a]
[a] Dynamic Macroecology/Land change science, Swiss Federal Institute for Forest, Snow and
Landscape Research WSL, Birmensdorf, Switzerland
[b] Department of Physical Geography and Ecosystem Science, Lund University, Lund, Sweden
[c] Centre for Mathematical Sciences, Department for Mathematics Lund University, Lund, Sweden
[*] Corresponding author: veiko.lehsten@gmail.com

## Abstract

Dynamic global vegetation models are a common tool to assess the effect of climate and land use change on vegetation. Though most applications of dynamic global vegetation models use plant functional types, some also simulate species occurrences. While the current development aims to include more processes, e.g. the nitrogen cycle, the models still typically assume an ample seed supply allowing all species to establish once the climate conditions are suitable. Pollen studies have shown that a number of plant species lag behind in occupying climatological suitable areas (e.g. after a change in the climate) as they need to arrive at and establish in the newly suitable areas. Previous attempts to implement migration in dynamic vegetation models have allowed simulating either only small areas or have been implemented as post process, not allowing for feedbacks within the vegetation. Here we present two novel methods simulating migrating and interacting tree species which have the potential to be used for simulations of large areas. Both distribute seeds between grid cells leading to individual establishment. The first method uses an approach based on Fast Fourier Transforms while in the second approach we iteratively shift the seed production matrix and disperse seeds with a given probability. While the former method is computationally faster, it does not allow for modification of the seed dispersal kernel parameters with respect to terrain features, which the latter method allows.

We evaluate the increase in computational demand of both methods. Since dispersal acts at a scale no larger than 1 km, all dispersal simulations need to be performed at maximum at that scale. However, with the current available computational power it is not feasible to simulate the local vegetation dynamics of a large area at that scale. We present an option to decrease the required computational costs, reducing the number of grid cells where the local dynamics is simulated only along migration transects. Evaluation of species patterns and migration speeds shows that the simulation along transects reduces the migration speed, and both methods applied, on the transects, produce reasonable results. Furthermore, using the migration transects, both methods are sufficiently computationally efficient to allow large scale DGVM simulations with migration.

## 1. Introduction

A large suite of dynamic global vegetation models (DGVMs) is currently used to simulate the effects of climate and / or land use change on vegetation and ecosystem properties. These simulations result in projections (or hind-casts) of species ranges as well as changes in ecosystem properties such as carbon stocks and fluxes. Examples of these DGVMs include ORCHIDEE (Yue et al., 2018), LPJ-GUESS (Sitch et al., 2003), IBIS (Foley et al., 1998), (Sato et al., 2007), for a review of DGVM features see (Quillet et al., 2010).

While most DGVM applications use plant functional types (groups of plant species with similar traits
and responses to environmental conditions), here we only consider applications which explicitly
simulate tree species, e.g. (Hickler et al., 2012). These models typically assume that species can
establish at any site once the environmental conditions become suitable. However, in real ecosystems
species need not only to establish and replace existing vegetation, which the processes in gap models
describe successfully, but they also need to have a sufficient amount of seeds present at a given
location to successfully establish. Implicitly, current DGVMs assume that ample amounts of seeds of
all species are present in every location.
While this approach might seem reasonable in cases where the vegetation can keep up with climate
change (i.e. moving sufficiently fast to occupy areas which become suitable), there have been a
number of instances reported where a considerable migration lag occurred. For instance *Fagus*
*sylvatica* has been shown to have a considerable migration lag and is currently still in the process of
occupying its climatological optimum (Bradshaw and Lindbladh, 2005).
The implementation of migration into dynamic vegetation models is not only of interest for the
simulation of historical species ranges, it is also of interest for the projection of ecosystem properties
in the future since  migration lags might lead to uncertainties in projected ecosystem properties if the
wrong species community is predicted to occur at a certain site (Neilson et al., 2005). Especially,
given that the speed at which environmental conditions change currently is unprecedented at least over
the last centuries, effects of the migration lag of key species should be evaluated when projecting
ecosystem properties. This holds in particular for projections over several centuries.  For periods of
less than 50-100 years ahead, which corresponds to at most a few generations of  most tree species, the
explicit modelling of seed dispersal might be less important for simulating tree distributions, in
particular when taking into account the overwhelming influence of human activities.
Migration lags can be caused by different factors. Seed transport might only occur over limited
distances. But also low seed amounts and in particular long generation times can slow down
migration. Seed amount and generation time depend on the competition with other trees: a free
standing tree starts earlier to produce seeds and produces more than a tree of the same age in a closed
forest. The competitors, however, are also migrating, which leads to feedbacks between the species
(Snell et al., 2014).
Thus, for simulations over large areas covering long time spans, species migration – consisting of a)
local dynamics influenced by the environment, b) competition between species, and c) seed dispersal –
has to be taken into account simultaneously for several species.
Species migration has been implemented successfully in dynamic vegetation models working on
smaller extents and finer scales than DGVMs typically use, e.g. forest landscape models (FLMs;
review in Shifley et al, 2017), such as TreeMig, (Lischke et al., 2006), Landclim (Schumacher *et al.*,
2004), Landis (Mladenoff, 2004), or Iland (Seidl et al., 2012) or spatially explicit individual based
models such as LAVESI (Kruse et al., 2018).
In these models, seed dispersal is modelled in a straightforward way: seeds are distributed from each
producing to each receiving cell with a distance dependent probability. However, transferring these
approaches to DGVMs is problematic, due to a number of conceptual and technical difficulties.
DGVMs usually operate on a coarse spatial resolution to reduce computational load and input data
requirements. This neglects the spatial heterogeneity within the grid cells. Additionally, and even more
critical for implementing migration, it leads to discretization errors: if it is assumed that the forest
representing the grid cell is located in the centre of the cell, the seeds cannot move far enough to leave
the cell (given a typical cell size of 50km by 50km or 10km by 10km). If, on the other hand, a
uniformly distributed forest in the cell is assumed in the simulation some seeds reach the neighbour
cell with each time step, leading to a resolution dependent speed up of migration.
Also some specifics of model implementations might complicate the inclusion of migration in some
DGVMs. Many DGVM implementations are done in a way that for each grid cell all years are
simulated before the simulation of the next cell is started. This is done to minimize input-output effort
since the whole climate data for each cell is read in at once and it also eases parallelisation for multi-
core computers, since in this case each node is assigned a number of grid cells which the node
calculates independently of the other nodes without communication. However, for simulating seed
dispersal, all cells need to be annually evaluated. Additionally to the reasons mentioned before, most
DGVM applications use plant functional types which comprise typically species with very different
traits with respect to migration (e.g. dispersal vectors or seed properties). Hence introducing migration
would require to split up PFTs into smaller groups and to parameterise the additional properties.
There have been a number of attempts to integrate species migration in DGVMs (cf. Snell *et al.*, 2014,
and Discussion section). For example, Sato and Ise (2012) developed a DGVM where species could
potentially migrate between neighbouring cells with a fixed rate of about 1km/year while Snell (2014)
simulated migration as an infection process between patches and within each grid cell.
However, to the knowledge of the authors, there is no implementation of a migration scheme into a
DGVM which allows simulations with a large extent, takes migration within the grid cell into account
and includes feedbacks between all simulated species.
Here we present two methods to fill this gap, i.e. allow simulating species migration of several species
simultaneously. The methods are implemented into the LPJ-GUESS DGVM but can potentially also
be implemented into other DGVMs. Though they are tested here using a virtual landscape, they can be
applied for simulations of large areas given current computing resources.

## 2. Methods

### 2.1 The dynamic vegetation model LPJ-GUESS

LPJ-GUESS is a flexible framework for modelling the dynamics of terrestrial ecosystems from landscape to global scales (Sitch et al., 2003; Smith et al., 2001). Similar to most other DGVMs, it requires time series of climate data (precipitation, air temperature and shortwave radiation), soil conditions and carbon dioxide concentrations as input and explicitly simulates vegetation cover. While it uses plant functional types in most applications, some applications simulate tree species (e.g. Hickler et al., 2012; Lehsten et al., 2015). LPJ-GUESS explicitly simulates canopy conductance, photosynthesis, phenology, and carbon allocation. It uses a detailed individual-based representation of forest stand structure and dynamics. Each species (or PFT) has a specific growth form, leaf phenology, life history and bioclimatic limits, determining its performance and competitive interactions under the forcing conditions and realized ecosystem state of a particular grid cell (Sitch et al., 2003). A large body of publications describes the features of LPJ-GUESS in detail; here we concentrate on the changes that were applied to LPJ-GUESS version 4.0 (Lindeskog et al., 2013; Smith et al., 2014). To differentiate between the original version of LPJ-GUESS and our extended version (where we implemented the migration module) we refer to the extended version as LPJ-GM (short for LPJ-GUESS-MIGRATION).

### 2.2 Technical implementation

Standard LPJ-GUESS simulations are typically performed at a computing cluster with cells running on different nodes of the cluster without any interaction of the nodes. We implemented a distributed simulation using MPI (Clarke et al., 1994) with the grid cells communicating with a master process.

Seeds are produced potentially in each grid cell at the end of each year after the first 100 years (see below). The number of seeds produced is sent to the node computing the dispersal while all nodes wait for this master node to finish the calculation. This node sends the number of seeds that arrive at each grid cell back to all nodes to continue the calculation.

Similar to the standard version of LPJ-GUESS (Sitch et al., 2003; Smith et al., 2001), in the first 100 years no seed dispersal is performed and all species are allowed to establish and grow without seed limitation and without N-limitation to equilibrate the soil pools with carbon and nitrogen. This time period is used to sample NPP given a certain N deposition and climate to subsequently equilibrate the N pools of the soil and a fast spin-up of 40000 years approximated using the sampled rates of C assimilation (Smith et al., 2014). After this initialisation period all vegetation is killed and succession starts from a bare soil and now seed limitation is active.

In LPJ-GM seed dispersal is done on an annual basis. The amount of seeds produced is communicated to the master node at the end of each year. The master node re-distributes seeds over the whole spatial

domain according to the dispersal algorithm and communicates the amounts of arriving seeds back to
each grid cell. Seeds transferred to the grid cells are added to the seed bank which determines
establishment probability in environmentally-suitable cells (environmental suitability is determined by
means of environmental envelopes, containing amongst others minimum survival and establishment
temperatures; see Smith et al. 2001). All communications between the processes are done via MPI
protocol (Clarke et al., 1994).
LPJ-GUESS is a gap model with the typical successional vegetation changes. To even out
successional based fluctuations in ecosystem properties and to be able to simulate disturbances most
previous applications simulate a certain number of replicate patches per grid cell. All patches share the
same climate but potentially differ in their successional stage due to different timing of disturbances
and stochastic mortality. Conceptually, each patch has a size of 1000 m$^2$ but represents an area
depending on the resolution of the grid cell. Patches have no spatial position with respect to each other
and do not interact (Smith et al., 2001). In LPJ-GM we reduced the number of patches to one but
achieved the representative averaging by using explicitly placed small grid cells instead of statistical
units (replicate patches). For each large grid cell in the climate grid we simulate a large number of
cells of 1km$^2$ area resulting in a more than sufficient averaging of successional stages. LPJ-GUESS
simulations are typically performed with patch numbers around 10 (e.g. Smith *et al.*, 2001) but
depending on the aim of the simulation patch numbers have been increased even to 500 (e.g. Lehsten
*et al.*, 2016). In our setup even with 50 km corridors LPJ-GM represents a 0.5x0.5 degree cell with
200 simulation cells ranging at the higher end of the patch number per area compared to previous
simulations. We demonstrate this in Fig. 3 where a single 11 km by 11 km large grid cell is separated
in to 11 by 11 smaller grid cells with similar climate. The local dynamics and seed production is only
simulated along the transects (grey or green cells in left panel of Fig.3). As a next step the seed
production is interpolated onto all cells for which no local dynamics, was calculated and the seed
dispersal is simulated. Finally, seedling establishment is simulated, but only in the grid cells on the
corridors (more details for the different steps are given below).

## 2.3 Migration processes

### 2.3.1 Seed production

The seed production starts once the tree reaches maturity height and is scaled linearly with leaf area up
to maximum (LAI).
The seed number produced per tree is calculated as the product of the maximum fecundity multiplied
by the proportion of the current LAI to the maximum LAI and multiplied by the area per grid cell. For
example, the maximum fecundity of beech is 29000, the maximum LAI is 5 m$^2$ *m$^{-2}$ and the maturity
height is 14.4 m. Hence a tree of 15m height is above the maturity height, and with an LAI of 2.5 m$^2$
*m$^{-2}$ it will produce 29000*0.5/5=14500 seeds. No specific age of maturity is taken into account.
All seeds of a species produced S(x',y') at a location (x',y') within a year are available for seed
dispersal. Once seeds have entered the seed bank, no further dispersal is possible (they remain in the
seed bank). Though LPJ-GUESS keeps track of carbon allocated to the main plant compartments and
even allocates a certain amount of carbon to seeds (which is transferred to the litter pool, the soil pool
and finally the atmosphere), for simplicity we decided not to relate the seed production to the carbon
accounting at this point. Allocation rules including seed production and even mast fruiting effects
(synchronised strong increases in seed production e.g. similar to Lischke *et al.* 2006) could be
included in the future.
**2.3.2 Seed dispersal**
The produced seeds are distributed according to
$S_d(x,y) = \int S(x',y')k_s(x-x',y-y')\,dx'\,dy'$        (eq. 1).
$S(x',y')$ is the seed production, and $k_s(x-x',y-y')$ the seed dispersal kernel in euclidean
coordinates. The seed distribution $S_d(x,y)$, i.e. the input of seeds in location x, y is then obtained by
integrating over all possible locations $x',y'$ for arriving at $x,y$.
Thus, the seed distribution is given by the convolution ($**$) of the seed production and the seed
dispersal kernel:
$S_d = S ** k_s.$                               (eq. 2)
For this study we used the seed dispersal kernel and parameterization for *Fagus sylvatica* from
TreeMig (Lischke *et al.*, 2006). The seed dispersal kernel defines the probability of seeds arriving at a
sink cell *(x,y)* from the source cell $(x',y')$ with a certain distance z $= \sqrt{(x-x')^2 + (y-y')^2}$ .
The kernel is specified in a polar coordinate system,
$k_s(z,\theta) = k_s(z|\theta)k_s(\theta)$, with the radial distance $z$. The seeds follow a mixture of two exponential
distributions, the short and the long term dispersal, while the angular dispersion, $\theta$, is uniform in all
directions (in our case the angular dispersion $\theta$ is uniform, but if one is interested e.g. in implementing
wind directions this can be changed). Thus, the radial component of the kernel is given by
$k_s(z|\theta) = (1-\kappa)\frac{1}{\alpha_{s,1}}e^{-\frac{z}{\alpha_{s,1}}} + \kappa\frac{1}{\alpha_{s,2}}e^{-\frac{z}{\alpha_{s,2}}}, \kappa \in (0,1)$            (eq. 3)
while the angular term is given by
$k_s(\theta) = \frac{1}{2\pi} for \ \theta \in [0,2\pi]$                          (eq. 4.1)

$k_s(\theta) = 0 \: otherwise$ .                                       (eq. 4.2)

The dispersal kernel is defined by the species specific values for the proportion of long distance
dispersal $\kappa$ and the species expected dispersal distances $\alpha_{s,1}$ and $\alpha_{s,2}$ for the two kernels.
The species specific values for these parameters (0.99 for $\kappa_s$ and 25m and 200m for the two mean
dispersal distances $k_s$ for *Fagus sylvatica*) were taken from by Lischke *et al.* (2006).

### 2.3.3 Seed bank dynamics

The number of the seeds in the seed bank (i.e. the dormant seeds in the soil that can germinate in
subsequent years in each cell) is increased by the influx $S_d$ of seeds according to (eq. 1), and reduced
by the yearly loss of germinability (caused by decay of seeds; see supplementary material 4 for
parameter values) and the amount of germinated seeds at the end of each simulated year, similar to
TreeMig (Lischke et al., 2006).
For each grid cell and each year we prescribe whether the species requires seeds to establish. By not
requiring seeds for establishment we define refugia, or we define that the species' seeds are known to
be very far dispersed and hence no explicit simulation of establishment by seeds is required for this
species. Technically this is implemented by reading in a list for each cell containing a year from which
onwards a species' establishment is not limited by the availability of seeds.

### 2.3.4 Germination

LPJ-GUESS is a gap model and in the original version the number of newly established saplings only
depends on the amount of light reaching the forest floor (given that the cell has a suitable climate). In
LPG-GM we additionally limit the establishment of seedlings depending stochastically on the number
of available seeds. Hence the seed limitation is applied before the light limitation. The probability that
a species establishes is given in equation 5.
$P_{est} = S \, p_x \, P_{germ}$                   (eq. 5)
Where the $P_{est}$ is the probability of the species establishing, $S$ is the number of seeds and $P_{germ}$ is the
seed germination proportion. The extra parameter $p_x$ takes (implicitly) the area of each grid cell into
account. In our case we fixed this parameter to 0.01 after initial testing. Hence if in a certain year 100
seeds are in the seed bank and the germination rate is 0.71 (value for *Fagus sylvatica*) the probability
of establishment is 0.01*100*0.71=0.71.

## 2.4 Enhanced dispersal simulation

One way to simulate seed dispersal is to calculate the convolution of the matrix containing the seed production and the seed dispersal kernel (specified in eq. 1 and eq. 3). However, evaluating the convolution explicitly can be computationally expensive for seed dispersal kernels with long range.

### 2.4.1 Fast Fourier transformation method (FFTM)

An alternative is based on the convolution theorem and the Fast Fourier Transformation (FFT), a technique commonly used in physics, image processing and engineering (Strang, 1994), but rarely in ecology or (see e.g. Powell, (2001)  Shaw et al., (2006), Pueyo et al., (2008)).

This approach carries out the computations in the frequency domain, see Gonzales & Woods (2002). Here we use the notation $F\{S\} = \int e^{-iux-ivy} S(x,y)\, dx\, dy$ to denote the two dimensional Fourier transform of $S$ and correspondingly $F\{k_s\}$ the two dimensional Fourier transform of $k_s$. It then follows that the Fourier transform of the convolution equals the product of the Fourier transforms

$$F\{S ** k_s\} = F\{S\}F\{k_s\} \qquad\qquad \text{(eq. 6)}$$

Thus, it is possible to compute the convolution by applying the inverse Fourier transform to the products of the Fourier transforms

$$S ** k_s = F^{-1}\{F\{S\}F\{k_s\}\} \qquad\qquad \text{(eq. 7)}$$

This equation must be discretized before evaluating it on a computer. The discrete Fourier transform is computed using the Fast Fourier Transform (Cooley and Tukey, 1965), which has a computational cost of $O(N^2 log^2(N))$ in two dimensions. The discrete approximation of $S_d$ is then given by

$$S_d = F^{-1}\{F\{S\} \odot F\{k_s\}\} \qquad\qquad \text{(eq. 8)}$$

where $\odot$ is the element-wise (Hadamard product) multiplication of matrices.

Nowadays, software packages for FFT typically only compute positive frequencies. That means that we have to shift the frequencies prior to the element-wise multiplication of $F\{S\}$ and $F\{k_s\}$. This is illustrated in Fig.1, see also supplementary material S.2.

<Figure 1 to be placed here>

While this method allows including different wind distributions by changing the seed dispersal kernel (as long as they are valid for the whole simulated area), it does not allow to use different seed dispersal kernels at different locations, e.g. due to prevailing wind directions in valleys, due to barriers to animal transport like a motorway, or due to lower transport permeability in already forested areas.

### 2.4.2 Seed matrix shifting method (SMSM)

Another way to simulate seed dispersal is to simulate the seed movement between the cells explicitly by shifting the matrix containing the produced seeds by one position (repeatedly in all directions of the Moore neighbourhood; i.e. the surrounding eight cells) and simulating seed transport of a certain proportion of the seeds into the next cell. Each move can be viewed as an independent random variable. Repeating these moves thus corresponds to a random walk process. The Lindeberg's condition for sequences for sums of independent random variables ensures that the kernel will be Gaussian under general conditions (Shiryaev, 2016), with the expected value given by the sum of expected values for each random variable and similarly for the variance (see supplementary material S.1 for a formal proof and a derivation of the parameters of the resulting normal distribution).

If this is done repeatedly it allows an easy implementation of spatial explicit differences in seed dispersal kernel distributions, by adjusting the proportions of seeds being transported into the next cell according to a similarly sized matrix containing the area roughness or permeability. By this approach, barriers and even wind speeds in latitudinal and longitudinal directions can be implemented by adjusting the dispersal probabilities accordingly. After the distribution of the dispersed seeds is calculated, the seeds are added to the seed bank. An example calculation of the first three steps of the SMSM (in the final simulation 10 steps are performed) is given in the Supplement S.3.

## 2.5 Corridors

Seed dispersal acts at a rather fine scale compared to the usual scale at which DGVMs are run (LPJ-GUESS is typically run at a 0.5 to 0.1 degree longitude / latitude scale), though some regional applications use finer grids (e.g. Scherstjanoi et al., 2014). Given that the average long distance seed dispersal for example for *Fagus sylvatica* is 200 m (representing 0.002 degree longitude / latitude at the Equator), simulations at such a coarse scale will not be able to capture this process.

As a compromise between currently available computing resources and required simulation detail we choose a 1km scale at which we performed our simulations. However, even at this scale, simulating large areas for example within the European continent would result in a high computational effort.

Given that in some areas the landscape is rather homogenous while other areas have a variable terrain (or land use conditions), we test whether for homogenous landscapes it is sufficient to simulate the local dynamics only in latitudinal, longitudinal and diagonal transects (i.e. north-south, east west, as well as, northeast-southwest and northwest-southeast corridors) and how this will influence the migration speed. The corridors are 1 grid cell wide and regularly placed in the simulation domain. Their density can be chosen by defining the distance between the latitudinal and longitudinal corridors.

Although LPJ-GM only simulates local dynamics in the cells along the corridors, the seed matrix
needed to be filled for the dispersal calculation using the FFTM or the SMSM algorithm. We applied a
nearest neighbour interpolation of the seed production before performing the seed dispersal calculation
(theoretical considerations show that a distance weighted average would strongly speed up the
migration).

## 2.6 Simulation experiments

To test our newly developed migration module we simulated the spread of a single late successional
species (*Fagus sylvatica*) through an area covered by an early successional species (*Betula pendula*).
The species specific parameters for both species are given in the Supplement S.4. All grid cells and all
years in the simulated area had a static climate suitable for both species. Though the simulated domain
is quadratic in our case it could have any shape. Each cell in the simulated domain has been simulated
independently (except for the influx and outflux of seeds) from each other. For one specific simulation
using the SMSM method we assumed differences in the dispersal ability (e.g. more or less permeable
areas or physical barriers) while the climate on all grid cells is still static and favourable. The dispersal
ability of the landscape is displayed in Fig. 2. Areas colored white have zero permeability, hence no
seeds can reach these areas.

<Fig. 2 placed here>

Figure 3 demonstrates the sequence of  simulating vegetation dynamics on the corridors, interpolation
of seed production, seed dispersal on the entire grid and back via the seed input on the transects.
<Fig. 3 placed here>

Given the uniformity of the climate, there should be no variability in the migration speed caused by
differences in climatic conditions. We simulated the spread of *F. sylvatica* from a single grid cell in
the corner of the study area which represents the refugium. We tested several corridor distances
(between the parallel and between the diagonal corridors) for their effect on the migration speed. To
calculate the migration speed we first determined the migration distance. This was the distance
between the start point of the migration and the 95-percentile farthest point in the virtual landscape
where the leaf area index (LAI) of *F. sylvatica* was larger than 0.5. This migration distance was
subsequently divided by the simulated time elapsed since the start of the migration. To avoid founder
effects we neglected all points within first 5 km from the starting location (the refugium). The
simulations were performed over 3000 years and over an area of 100 by 100 cells of 1 $km^2$. Finally we
ran one simulation where we did not calculate the seed dispersal (but performed all communication
between cells and one run even without the communication), hence allowing us to estimate the
computation time demand for the seed dispersal calculation.

## 2.7 Performance evaluations

To estimate the performance of our methods against an implementation in which each grid cell
exchanges seeds with each other we developed a Matlab® script, since initial testing had shown that
such a procedure would be too slow to be implemented in LPJ-GUESS. Hence when evaluating the
performance differences from the script one has to bear in mind that these are calculated in a different
environment. However in a general sense we can see no reason why they should not reflect the
performance differences between the algorithms. The whole Matlab® script testing the performance
including the graphs is part of the Supplementary material.

## 3. Results

### 3.1 Explicit seed dispersal

The study comparing the performance of different migration mechanisms without the vegetation
dynamics, implemented in Matlab® , has shown that both the FFTM as well as the SMSM were
performing faster than the explicit dispersal from each grid cell to each other within the range of the
dispersal (last figure Supplement 2). This was especially pronounced if the area to be simulated was
increased. Though faster than the explicit dispersal method, the SMSM was still up to an order of
magnitude slower than the FFTM, in particular  for large simulations domains in Matlab® while the
FFTM and the SMSM required relatively similar amounts of time in the implementation in LPJ-
GUESS (tab.1).

### 3.2 FFTM simulations

Using the parameterization from TreeMig in a complete (no corridors) simulation area of 100 by 100
grid cells with the size of 1km$^2$ each resulted in a migration speed of 34 m per year for *Fagus sylvatica*
(Fig. 4).
<Figure 4 placed here>
Though the establishment in the model is stochastic, the simulated spread was relatively smooth. The
corridor distance of 10 km, 20 km and 50 km resulted in a reduced migration rate of 26, 28 and 28
m/year (compared to a simulation without corridors), respectively (Fig. 4, lower three rows of panels).
While in the simulation without corridors the variability of the migration speed was relatively low
(dots under the red line in upper left panel of Fig. 4), this variability was strongly increased when
corridors were simulated. This was caused by *F. sylvatica* migrating along the diagonal, reaching the
end point of the diagonal and then migrating along the longitudinal and latitudinal corridors into cells
which hd actually a shorter distance to the refugia than the endpoint of the diagonal.
The simulation time per grid cell in the whole area (range for which the seed dispersal was computed)
was increased by 12% by simulating the FFTM, but by using the corridors it was reduced to 36%,
22% and 12%, compared to simulating the full area (Tab. 1, col. 7). The proportion of computation
time used to perform the FFTM increased from 11% without corridors to 18%, 29% and 29% for
simulations with corridors every 10, 20 and 50 km. This estimate only includes the required time for
computing the FFT-based seed dispersal since the control run without seed dispersal still contained all
communication between cells. For the control run seeds were produced and sent to the master but the
master did not compute the seed dispersal, though still communicated with all other nodes to allow a
fair assessment of the computation time demand of the two methods (see Tab. 1). An additional run
without any communication resulted in a computation time similar to the run with communication.

## 3.3 Shifting seed simulations


Initial testing of the probability parameter for the SMSM suggested a value of $p = 5 * 10^{-7}$ to generate a
migration speed comparable to the migration speed for the FFTM based on the TreeMig
parameterization. Using the derivation presented in supplement 2 it is possible to calculate this
parameter for a Gaussian dispersal kernel. One can approximate any dispersal kernel by adding several
Gaussian kernel, however this would increase calculation time since the SMSM would have to be
performed several times. Therefore we decided to choose a parameter for the SMSM approximating
the migration speed rather than the seed dispersal kernel used in Lischke *et al.* (2006). This resulted in
a migration speed of 39 m/year for the filled area and 27m/year respective 29 m/year and 30m/year for
the 10 km, 20 km and 50km corridors (Fig. 5).
<Figure 5 placed here>
Similarly to the FFTM simulations, the migration speed was reduced for simulations with transects
(see table 1 for a summary). Also comparable to the FFTM based seed dispersal computation,
calculation time per grid cell in the whole area (range for which the seed dispersal is computed) was
increased by 16% by the simulation of dispersal, but reduced to 35%, 19% and 11% by using the
corridors. The proportion of calculation time spent for simulating the seed dispersal is comparable to
the proportion using the FFT, it was 16%, 19%, close to 23% and 32% (see Tab. 1).
Since the SMSM allows adjusting the probability depending on the seed transport permeability of the
terrain we also simulated the migration within a non-homogenous dispersal area. The results of this
simulation are displayed in Fig 6. The total computation time for this simulation was 46000 CPU*h
for 6000 years.

<Figure 6 placed here>
Though all cells of the virtual landscape had a similar climate, some cells were never occupied (see
Fig. 6) because the seeds were not able to reach them due to the different permeability (which might
not be reasonable for real world simulations but demonstrates the method). Migration speed was
different in different parts of the simulated area.
<Table 1 placed here>

# 4. Discussion

To our knowledge, in our study for the first time (tree) species migration has been implemented in a DGVM in a way that allows simulations of simultaneously migrating and interacting species for large areas.

## 4.1 Performance of new migration methods

The presented new methods for simulating migration in DGVMs showed a promising performance in different aspects.

The first is the gain of efficiency by the FFTM and the SMSM methods as compared to the traditional, straightforward approach to evaluate the seed transport from each cell to each other (last Fig in S.2). A two dimensional FFT can be obtained by successive passes of the one dimensional FFT, hence the complexity will be the one-dimensional complexity squared (Gonzalez and Woods, 2002).The computational complexity for the FFTM is $O(N^2 log^2(N))$ for a $N \times N$ grid discretizing the seed distribution, while the complexity of the direct implementation of the convolution approach in the SMSM is $O(2KRN^2)$ for a $N \times N$ grid discretizing the seed distribution and $R \times R$ kernel with $K$ being the number of iterations of the SMSM (for the derivation see supplementary material S.1). This can be computationally comparable to the FFTM for kernels with short range of $R$. Secondly, simulating the local dynamics only along the corridors instead of in the full area resulted in a similar migration pattern, and the simulated migration speed was similar to that of the simulation with full grid cell cover (though it is slower, caused by the stochasticity of the establishment, see table 1), but needed much less computing time (reduction of 88% for the corridors every 50km).

## 4.2 Comparison of the two dispersal methods

In this study we present two alternative methods for simulating dispersal, which differ in their properties. While the FFTM allows any type of seed dispersal kernel, the SMSM corresponds to a normal distribution kernel. Although other shapes of dispersal kernels can be approximated by weighted sums of normal distributions, of which each of them has to be simulated by an own SMSM, which will cause strong increases in computational demand. Additionally the SMSM restricts the long tail of the distributions by the number of iterations, as the seeds can travel only travel one grid cell per iteration step.

On the other hand, the advantage of the SMSM lies in its ability (contrary to the FFTM) to modify the parameters of the seed dispersal kernel spatially, depending on the terrain. If instead of applying a single permeability for all directions, a different permeability is applied for each of the 8 directions (e.g. north, northeast, east, etc.) this method also allows a spatially explicit consideration of wind directions (which is not possible for the FFTM, as it relies on a universal kernel applied to the entire area). Hence, depending on the aim of the analysis either one or the other or a combination of the algorithms is most suitable.

While not implemented here, it should be theoretically possible to use the FFTM (preferably with
corridors) for some homogenous parts of the simulated area and the SMSM for the remaining part in a
single simulation. As long as the seed donor areas for both methods are exclusive, and the areas in
which the seeds are allowed to disperse overlap at least with the width of the kernel, we can see no
reasons why this should not be feasible.

## 4.3 Comparison to other approaches

Our new species migration submodule FFTM uses for the first time an algorithm based on Fast Fourier
Transformation to simulate dispersal in a DGVM. Due to its efficiency, the FFT one of the
"workhorses" in mathematics, physics and signal processing (Strang, 1994). In ecology, there have
been a few applications using FFTs to simulate dispersal of pollen (e.g. for risk analysis, Shaw et al.
(2006), seeds (Pueyo et al., 2008) or even in a course compendium (Powell, 2001b)), but not as a
standard technique in DGVMs.
The SMSM, in turn, mimics the seed transport process itself in a simple and straightforward way,
which to our knowledge has also not been implemented in DGVMs either.
Both approaches are combined with features of modelling species migration that are already used in
other dynamic vegetation models (cf . Snell, 2014).
The cellular automaton KISSMig (Nobis & Normand, 2014), e.g. simulates the spread of single
species driven by a spatio-temporal grid of suitability, and by transitions to the nearest neighbour cells,
which is similar to one iteration in the SMSM. The suitability based models CATS (Dullinger *et al.*,
2012) or MigClim (Engler and Guisan, 2009) simulate a simple demography of single species and
explicitly the spread based on a seed dispersal kernel.
To also account for ecophysiology, the CATS model was combined with LPJ-GUESS in a post-
processing approach (Lehsten *et al.*, 2014). A spatio-temporally explicit suitability for a single species
was estimated from LPJ-GUESS simulations of the productivity of this species, assuming the presence
of the other species. This suitability was subsequently used within CATS to simulate migration. Such
a post-processing approach however does not include interactions between several migrating species.
Forest landscape models  have been developed to integrate such feedbacks between species as well as
dispersal (He et al., 2017; Shifley et al., 2017). These models simulate local vegetation dynamics with
species interactions, and dispersal by explicit calculation of seed or seedling transport probabilities
with dispersal kernels of different shapes (e.g. LandClim (Schumacher *et al.*, 2004), Landis
(Mladenoff, 2004), Iland (Seidl et al., 2012)). To capture spatial heterogeneity, they run at a
comparably fine spatial resolution (about 20-100m grid cells), allowing only the simulation of
relatively small areas due to computational demands.
To overcome such computational limits, several approaches for a spatial upscaling of the models have
been put forward. For example, the forest landscape model TreeMig can operate at a coarser resolution
(grid cell size 1000m) because it aggregates the within-stand- heterogeneity by dynamic distributions
and height classes (Lischke *et al.*, 1998), which allows applications at a larger scale, e.g. over entire
Switzerland (Bugmann et al., 2014) or on a transect through Siberia (Epstein *et al.*, 2007). Another
upscaling of TreeMig was achieved by the D2C method  (Nabel, 2015; Nabel and Lischke, 2013)
which simulates local vegetation dynamics only in a subset of cells that are dynamically determined as
representative for classes of similar cells. This method led to a computing time reduction of 30-85%
compared to the full simulation. This reduction is in a similar range for our transect method depending
on the configuration of the corridors.
In DGVMs, the discretization problem resulting from the need to upscale from the fine scale at which
migration processes act to the scale at which DGVMs work is very pronounced, because they are
designed to operate on very large extents (continents or the entire globe). Given the computational
demands of the simulations, they are therefore typically running at a coarse resolution for example 0.5
or 0.1 degree longitude / latitude, and simulate the vegetation dynamics at the centre of each of these
grid cells, assuming this point to be representative for the entire cell.
Snell (2014) approached the discretization problem for the DGVM LPJ-GUESS by   assuming that the
numerous replicates of the vegetation dynamics on a patch are randomly distributed over the area of
the grid cell (using 400 patches). Migration within the grid cell is treated similar to an infection
process, where the probability of a patch becoming infected (e.g. of the migrating species being able to
establish) depends only on the number of already invaded patches within the grid cell. Only once a
migrating species managed to establish in a certain proportion of the patches of the simulated grid cell,
further dispersal (explicit via a dispersal kernel) into surrounding grid cells is possible. Yet, there is no
spatial orientation of the patches within the grid cell and all simulations in this approach are strongly
resolution dependent. Simulations of large areas such as continents remain computational challenging
with this approach.
Our transect approach, similarly to the approach of Snell (2014), uses smaller representative spatial
units, 1km-cells, for a spatial upscaling. Since these small grid cells are arranged in contiguous
corridors, the migration along these corridors can be simulated without or with only a small
discretization error. The results indicate that also the error potentially introduced by the interpolation
to the rest of the area is small.
The two approaches that we present differ in their ability to simulate heterogeneous landscapes (in
terms of permeability). We suggest using the FFTM with corridors in homogenous landscapes (to
speed up the computation) and to use the SMSM without corridors in heterogeneous landscapes. In
cases where parts of the domain are heterogeneous (e.g. the regions around a mountainous  area) and
other, homogenous parts (e.g. lowlands), the cells can be arranged in a way that they cover the whole
area in the heterogeneous part and only corridors in the homogenous  part. In this setting the SMSM
can still be used for the whole domain and an improvement of computation time can be achieved by
only simulating the local vegetation dynamics in the homogenous parts of the domain. Thus, with our
approaches, we have combined several advantages of the before mentioned approaches: the seed
dispersal from forest landscape models, improved by the novel FFTM or SMSM and the
ecophysiology, structure and community dynamics of LPJ-GUESS. We furthermore found a
compromise between discretization and efficiency by the corridor method.

## 4.4 Potential further improvements

Despite the satisfying performance of the new methods in these first tests some aspects suggest further
development.

### 4.4.1 Computation time

Even with the computing time reduction by the corridor approach using a corridor of 50km distance,
the computing time required for the simulations including dispersal was still considerable. The reason
is that the number of cells on the corridors (where the local dynamics are simulated) is larger than the
number of replicates usually used in all the 1 or 0.5 degree grid cells simulated in traditional DGVMs.
For large-scale applications,  the approach should be further optimized, e.g. by choosing corridors
even further apart from each other in homogenous areas and adapting the corridor density to the large
scale (between grid-cell) heterogeneity of the terrain. The within grid-cell heterogeneity in turn can be
accounted for by deriving seed dispersal permeability, that can be used in the SMSM approach.
Another area of improvement lies in the technical implementation of the seed dispersal algorithm. In
the current implementation, the seed dispersal is performed at a single cpu, while all other cpus wait
until they receive the seeds. There are certainly ways to perform the seed dispersal computation on
several nodes to decrease the waiting time. Furthermore, in multi-species simulations the dispersal has
to be calculated for each migrating species. In this case, the dispersal of different species should be
calculated on separate nodes. Enlarging simulation areas generally resulted in longer runtimes for all
methods. Sometimes, however, the runtimes decreased in a pronounced way for the FFTM
(supplementary material, S.2). A cause for these decreases is that the efficiency of the FFT depends on
possible factorizations of the domain size (Bronstein, I.N., Semendjajew, K.A., Musiol, C., Mühlig,
1995).  For example, it is most efficient for domain sizes of $2^n$. Thus, a careful choice of the domain
size or of an FFT code doing that automatically promises to speed up the FFTM. The last figure in S.2
does not represent the differences in computation time between SMSM and FFTM as they are
measured on the computing cluster when performing the actual simulation. While in table 1, there was
only a marginal difference between the calculations of the two methods, the differences in the
Matlab® implementation presented in S.2 are up to an order of magnitude. It seems that Matlab® uses
a different optimization for calculating the Fast Fourier transformation even though both Matlab® as
well as the FFT libraries used on the computing cluster are based on the libraries provided by fftw.org.

### 4.4.2 Migration speed reduction by corridor approach

It is to be expected, that any sub-cell assumption results in discretisation errors. In our case the
assumption of a corridor reduced the migration speed. This needs to be taken into account when
evaluating the result of such studies. The design of the corridors might also not have been optimal,
maybe a corridor wider than a single cell might result in less decrease of migration speed. However,
these types of analysis are outside the scope of this study. One other aspect of using the corridors is
that while a late successional species (in our case *F. sylvatica*) has certainly no problems to establish
below the early successional species, in the case of an early successional species (e.g. *B. pendula*)
migrating into an area occupied by a late successional species, the corridors might decrease the
migration speed even more. An early successional species can only establish after sufficient light
reaches the ground, either due to the senescence of a tree of the established species or a disturbance
event. The narrow corridors might have strongly limited the availability of such grid cells. However
since early successional species have typically a good dispersal ability, this should not influence
simulations of tree migration following climate change (e.g. after the last glaciation).

### 4.4.3 Parameterisation of dispersal kernels and other plant parameters

In this study the focus was on developing and testing the novel methods, i.e. we did not attempt to
correctly simulate the spread of *F. sylvatica* over a defined time period. The calculated spread rates
were well below most of the spread rates in the literature. *F. sylvatica* has been estimated to migrate
with ca 100 m per year based on pollen analyses by Bradshaw and Lindbladh (2005). Although such
estimated high migration speeds could also be the result of glacial refugia located further north than
assumed (Feurdean et al., 2013), our estimates of the migration speeds of 20-30 m/year still seem
rather low. However, in this paper we aimed to implement tree migration by using the
parameterisation of TreeMig in a DGVM and thereby allow large scale simulations. Our estimated
migration rates of 20-30 m per year are very close to the migration rates estimated for this
parameterisation for TreeMig by Meier et al., (2012) which estimated a value of 22 m per year. Hence,
though we implemented the migration module into a conceptually very different model, the resulting
migration rate remained relatively similar.
To perform model runs estimating the migration speed of any species would require a fine tuning of
the, age of maturity, seed production, dispersal parameters, germination rates, and seed survival
(which are very rough estimates in TreeMig; Lischke et al., 2006) to generate the observed migration
e.g. by comparing to migration rates based on pollen records. Unfortunately, though all of these
parameters are most likely strongly influencing the migration rates, they are not only hard to find in a
study performed with similar methods for all tree species, they are likely to be highly variable
depending on growth conditions and even provenance of the individual tree. However for a large scale
application at least the sensitivity of these parameters should be evaluated.
In our model, we assumed seed production to start at a fixed, species specific height of maturity which
accounts for a developmental threshold, but also growth and thus for environmental conditions
(similar to TreeMig, Lischke et al.2006). Other studies used age of maturity as a trigger to start seed
production, which has been shown to be important to determine tree migration rates (e.g. Nathan et al.,
2011). The aim of this study was not a full sensitivity analysis but a study showing that a similar
approach as Lischke et al. (2006) results in comparable migration rates.We will implement the option
to use age of maturity in the next version of LPJ-GM.
Applications of our approach to simulate migration in the future are only suitable if the migration
speed of any species is substantially faster than the migration speed that we reach for *F. sylvatica* (due
to time periode for which climate projections are available). Furthermore, independent of the used
model, migration simulations are only suitable if the species are not typically planted, as in many
commercial forests.

## 4.5 Potential for applications

The test simulations were performed at a virtual landscape of 100km by 100km, but eventually the
method is aimed to allow large scale simulations over several millennia. Regarding memory
requirements, this is possible of currently available hardware: Test runs with landscapes of 4000 by
4000 grid cells (i.e. the size of Europe) performed without technical problems at least regarding the
memory requirement (given 62 GB of RAM). The considerable computational cost however requires a
relatively high amount of computing time, which might be reduced by efforts for speeding up (due to
efficient parallelisation) of the FFTM (currently the FFTM is performed on a single node while the
remaining nodes are idle, one could use all nodes to perform the FFTM) or by even further apart
corridors.

## 5. Conclusions

The presented novel approaches offer high potential to simulate the spatiotemporal dynamics of
species which are migrating and interacting with each other simultaneously. The approaches are not
restricted to LPJ-GUESS, but can in in principle be applied to other DGVMs or FLMs which simulate
seed (or seedling) production and explicit regeneration. The presented methods need to be improved in
terms of computing performance to allow simulations of tree migration at continental scale and over
paleo time scales. Our study also shows that the estimates for seed dispersal kernels for the major tree
species need to be revised to allow simulations of forest development for example over the Holocene.

## 6. Author contributions

VL, DL and HL designed the study, VL performed the simulations and the statistical analysis. MM and EL contributed to the study design, MM also performed large parts of the coding.  EL developed the formal proof in Supplementary material S.1 and the computation performance related estimates in the Conclusion section. All authors contributed to the writing of the article.

## 7. Competing interests

The authors declare that they have no conflict of interest.

## 8. Acknowledgements

This study was funded by the Swiss National Science Foundation project CompMig, Nr . 205321_163223 .

## 9. Code and Data availability

The code generating the figures in Supplementary material 2 are part of the material. The used DGVM LPJ-GUESS containing the migration module can be requested from the author.

The data behind all figures will be published on the DataGURU server (dataguru.lu.se; doi:10.18161/migration_lehsten_2018) upon acceptance of the paper.

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

# 11. Contents of the supplementary material



## 12. Figures and tables




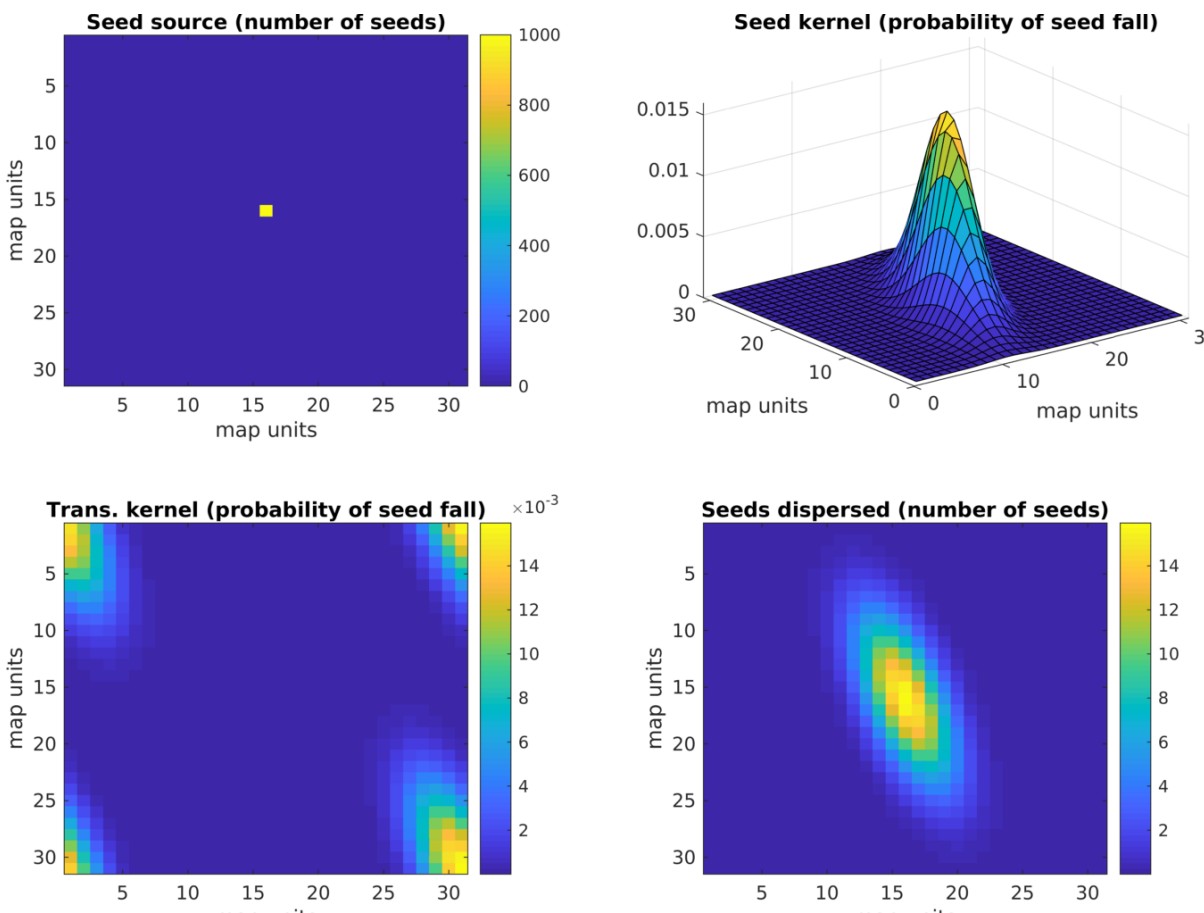


Fig. 1. Upper left panel: seed source. Upper right panel: example of a seed dispersal kernel (here a
non-symmetric kernel is assumed), lower left panel: transformed seed dispersal kernel, lower right
panel: seed distribution after convolution.

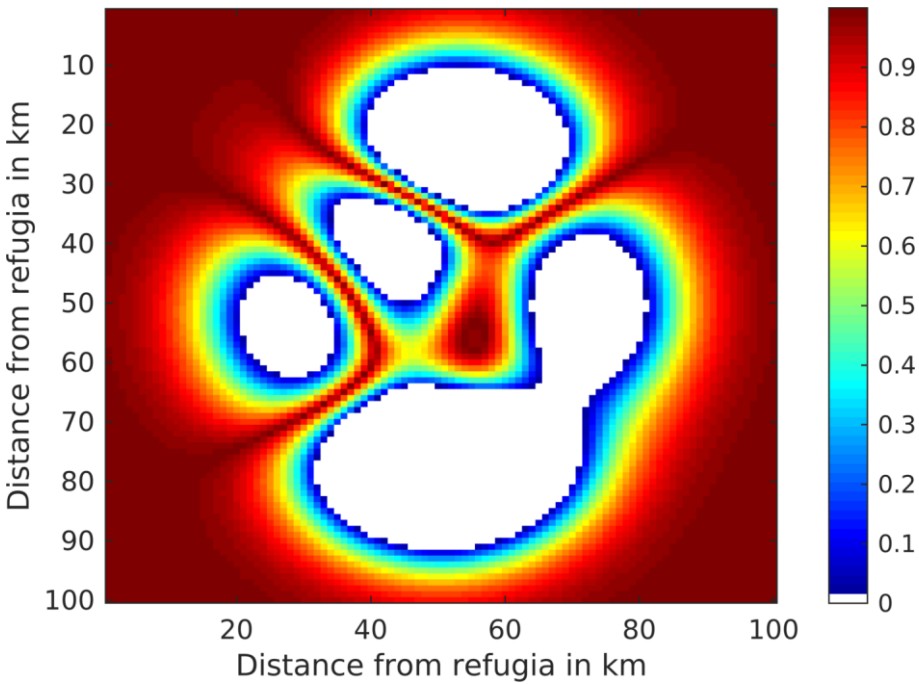


Fig. 2: Seed dispersal permeability for SMSM simulation tests. Each time the seed matrix is shifted, the probability of entering the new cell (which in our test is set to $5*10^{-7}$) is multiplied with the seed dispersal permeability of the new potentially entered cell.


Local dynamics, including seed production, on corridors

Seed production interpolated (blue), on grid

Seed dispersal, on grid

Seed input, on corridors

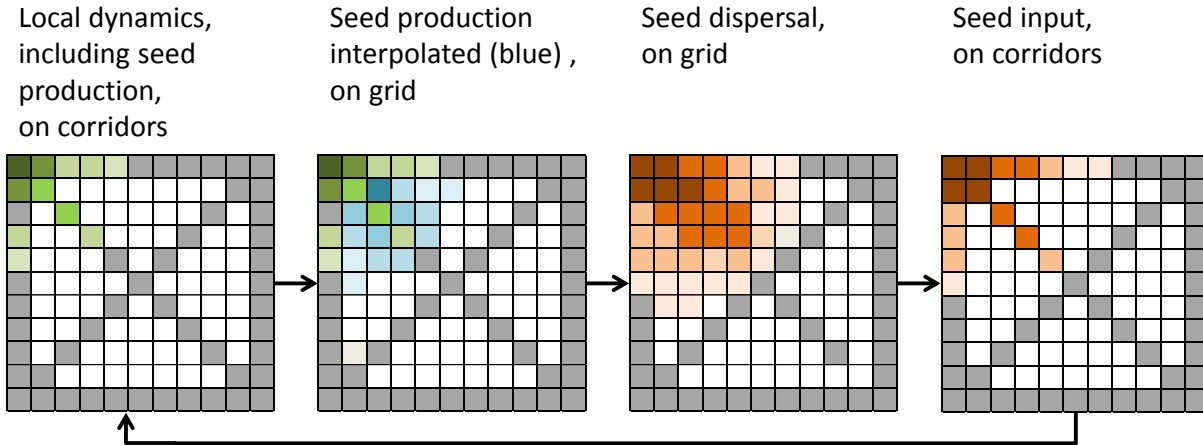

Fig. 3. Example of a simulated grid with transects (grey). In each time step the local vegetation dynamics including the seed production (green) is calculated on the transects. Then the seed production of each species is interpolated from the transects to all non-transect grid cells (blue) and then dispersed on the entire grid (brown). The seed input on the transect cell then enters the local dynamics in the next time step.

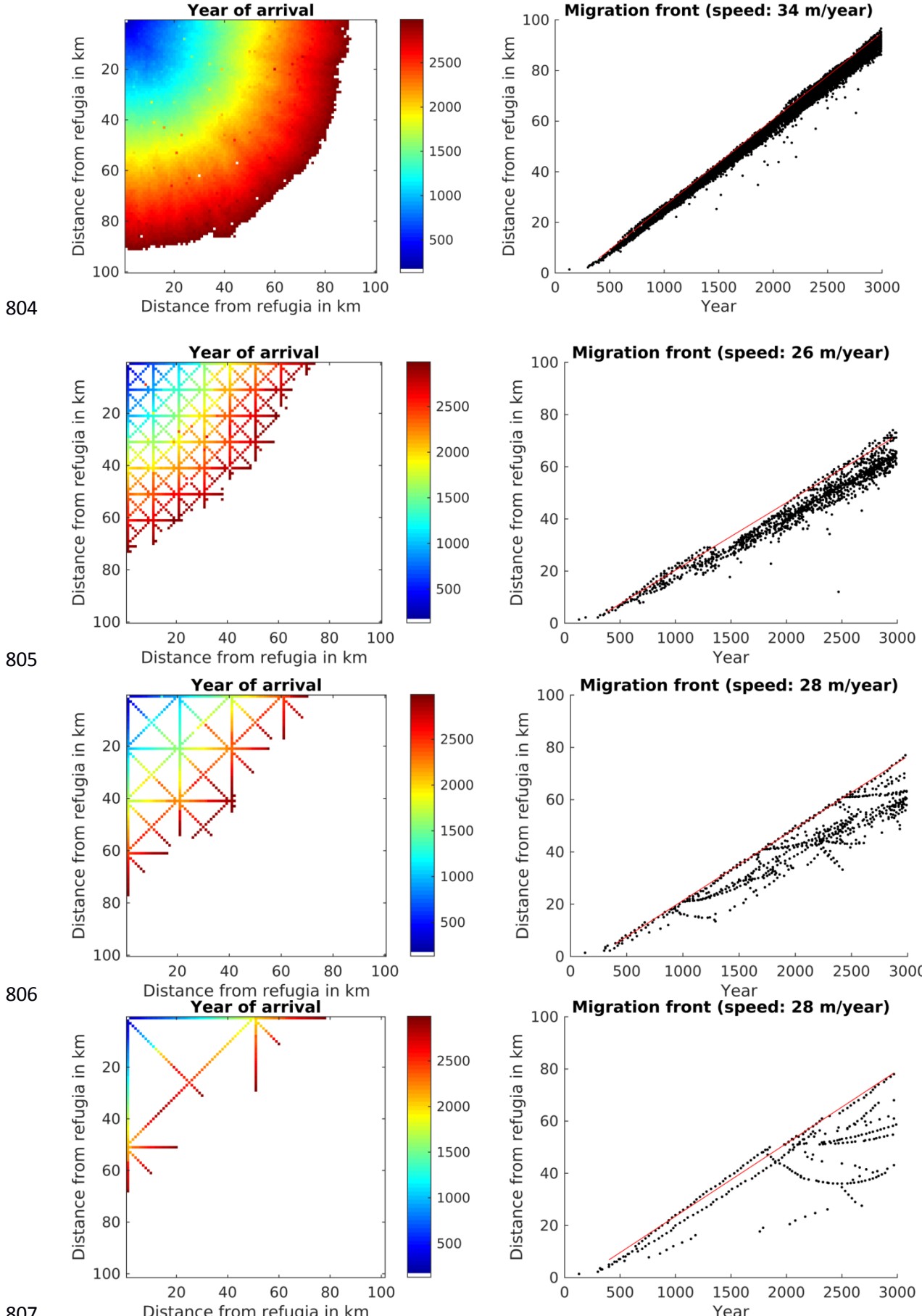





Fig. 4 Spread of *Fagus sylvatica* through an area of 100 * 100 grid cells with static climate using the
FFTM algorithm with no corridors or corridors every 10km, 20km or 50km. The left panels display
the time when *F. sylvatica* first reached an LAI of 0.5. *F. sylvatica* is allowed to establish freely only
in the upper left corner. The right panels show the distance of the grid cells with LAI 0.5 for *F.*
*sylvatica* from the starting point. The red line indicates the 95 percentile of the grid cells farthest away
from the starting point. The migration speed is calculated as slope of this line, taking only grid cells at
least 5 km away from the starting point into account to avoid some initial establishing effects.

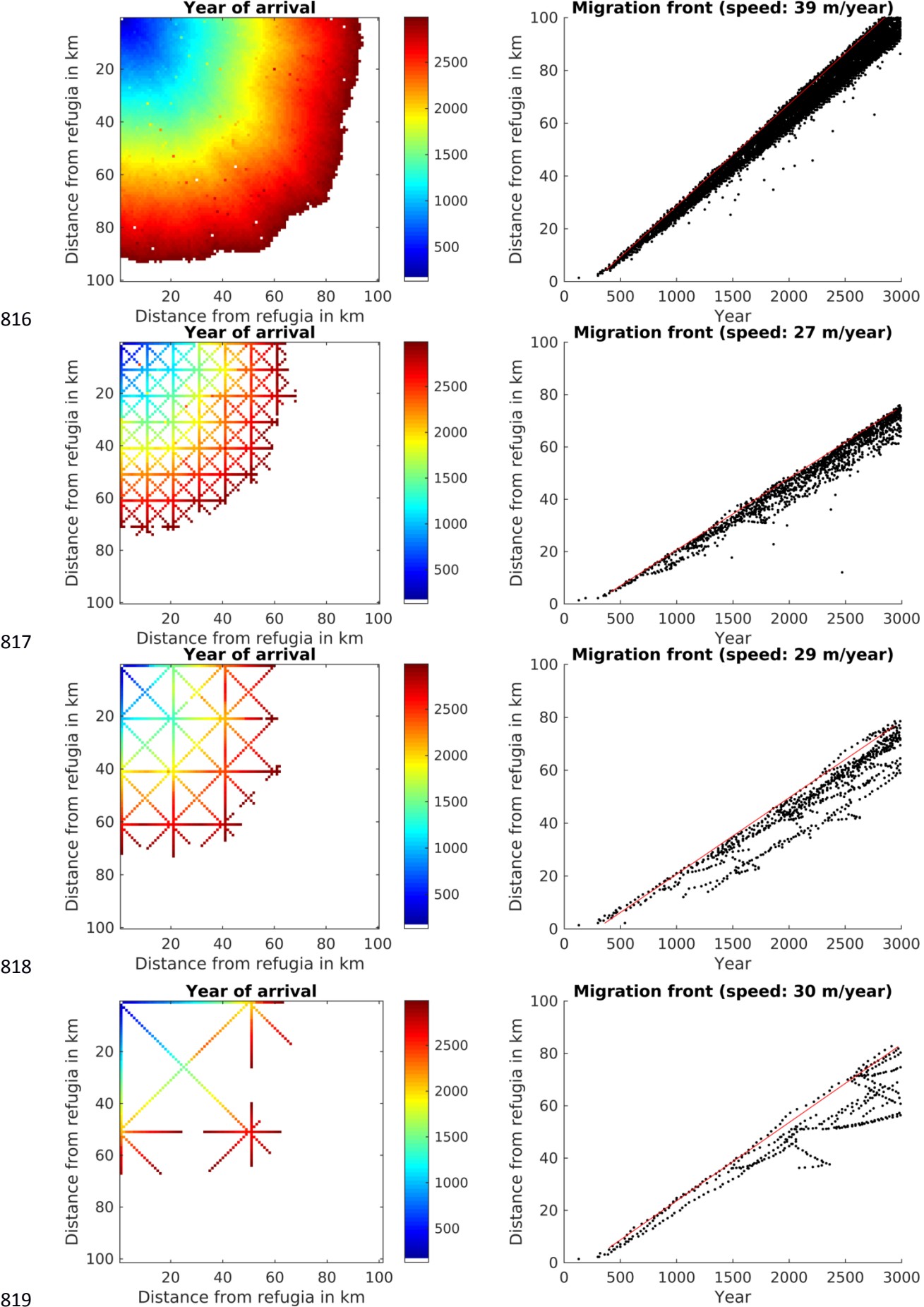





Fig. 5 Spread of *Fagus sylvatica* using the SMSM through an area of 100 * 100 grid cells with
identical climate, using the full area (upper row of panels) or corridors every 10$^{th}$, 20$^{th}$ or 50$^{th}$ cell. For
more explanation see Fig. 3.

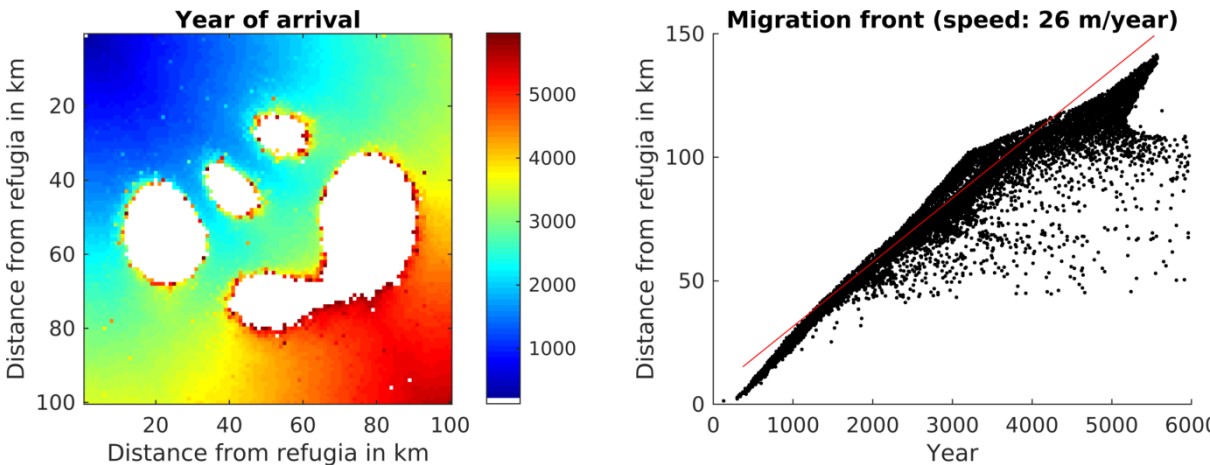

Fig. 6 Spread of *Fagus sylvatica* using the SMSM method through an area of 100 * 100 grid cells with
identical climate but probability of seed fall is set to 0.00005 multiplied with the spatially explicit seed
dispersal permeability value as shown in Fig. 2. Note that we increased the simulation time to 6000
years in order to have F. sylvatica establishing in all areas.

Table 1. Summary of migration speeds and calculation time. A corridor distance of 0 indicates no
corridors but an area completely filled with grid cells. The simulated grid cells column lists the
number of cells for which LPJ-GM calculates the population dynamics, in all simulations the
simulation domain (for which the seed dispersal was calculated) had a size of 10000 grid cells and all
simulations were performed over 3000 years. The last line lists a simulation identical to the others
except that no seed dispersal was calculated to allow estimating the computation time demand for this
operation.

| Seed dispersal mode | Corridor distance (cells) | Simulated grid cells (corridor cells) | Migration speed, m/year | Computation time (CPU h) | Comp. time change per corridor grid cell compared to sim. without dispersal (CPU h) | Total comp. time change for whole domain compared to sim. without dispersal (CPU h) | Percentage of CPU time for dispersal | Decrease due to corridor simulation |
|---|---|---|---|---|---|---|---|---|
| FFTM | 0 | 10000 | 34 | 1800 | +12% | +12% | 11% | |
| FFTM | 10 | 3330 | 26 | 650 | +22% | -59% | 18% | 64% |
| FFTM | 20 | 1765 | 28 | 400 | +41% | -75% | 29% | 78% |
| FFTM | 50 | 977 | 27 | 220 | +41% | -86% | 29% | 88% |
| SMSM | 0 | 10000 | 39 | 2000 | +25% | +19% | 16% | |
| SMSM | 10 | 3330 | 27 | 700 | +31% | -59% | 19% | 65% |
| SMSM | 20 | 1765 | 29 | 400 | +41% | -77% | 23% | 81% |
| SMSM | 50 | 977 | 30 | 220 | +41% | -86% | 32% | 89% |
| Non | 0 | 10000 | 0 | 1600 | 0% | 0% | 0% | |
