# Peer review of "LPJ-GM 1.0: Simulating migration efficiently in a dynamic vegetation model"

_Geoscientific Model Development, 2018_

## Short Comment (SC1) · 12 Sep 2018

Dear authors,

In my role as Executive editor of GMD, I would like to bring to your attention our Editorial version 1.1:

http://www.geosci-model-dev.net/8/3487/2015/gmd-8-3487-2015.html

This highlights some requirements of papers published in GMD, which is also available on the GMD website in the 'Manuscript Types' section:

http://www.geoscientific-model-development.net/submission/manuscript_types.html

In particular, please note that for your paper, the following requirements have not been

met in the Discussions paper:

- "The main paper must give the model name and version number (or other unique identifier) in the title."

- "All papers must include a section, at the end of the paper, entitled 'Code availability'. Here, either instructions for obtaining the code, or the reasons why the code is not available should be clearly stated. It is preferred for the code to be uploaded as a supplement or to be made available at a data repository with an associated DOI (digital object identifier) for the exact model version described in the paper. Alternatively, for established models, there may be an existing means of accessing the code through a particular system. In this case, there must exist a means of permanently accessing the precise model version described in the paper. In some cases, authors may prefer to put models on their own website, or to act as a point of contact for obtaining the code. Given the impermanence of websites and email addresses, this is not encouraged, and authors should consider improving the availability with a more permanent arrangement. After the paper is accepted the model archive should be updated to include a link to the GMD paper."

Please include the version number of LPJ-GM in the title of the revised manuscript. Additionally, please include information how to optain the LPJ-GM Code in the Code Availability Section. Note, that it is not sufficient to only state that the code is available from author without stating reasons, why publication is not possible.

Yours,

Astrid Kerkweg

---

## Author Comment (AC1) · 18 Sep 2018

1. Response to title: We will change the title to Simulating migration in dynamic vegetation models efficiently with LPJ-GM 1.0

2. Response to availability of LPJ-GM 1,0

LPJ-GM is based upon LPJ-GUESS adding one module performing all calculations and new distribution of seeds for the simulation of the migration. There are a few other changes mainly with technical character in other modules.

LPJ-GUESS is an established modelling platform with a large community of users worldwide. Code is available via the portal and procedures managed by Lund University, which has been in place for more than 10 years. Currently more tan 20 persons

in Lund are working on with LPJ-GUESS, so we expect this to be a viable project for the foreseeable future. Code is provided freely to bona fide research users subject to the condition that it is not distributed further, new users being referred back to the portal. Details are available at www.nateko.lu.se/lpj-guess. Thus the code is easy to get, but not open source in a strict sense.

The unit developed for this ms will be made available publicly with an assigned DOI it is attached here as a supplement (migration.cpp). However, the implementation of all procedures was already provided in the attached Matlab script which performs a seed dispersal using the two suggested procedures.

Please also note the supplement to this comment:
https://www.geosci-model-dev-discuss.net/gmd-2018-161/gmd-2018-161-AC1-supplement.zip
* * *

---

## Referee Comment (RC1) · Anonymous Referee #1 · 19 Sep 2018

Overall Comments:

The paper describes two approaches, for simulating seed dispersal in global-scale dynamic vegetation models. Vegetation migration in response to climate change (both past and future) is a major area of research, and the ability to simulate dispersal in DGVMs would be a major advance. There is certainly scientific merit in this manuscript, however there are numerous issues that need to be addressed. In general, since this is a paper about model development, more details about the model and justifications for their choices, need to be included.

Major Comments

Seed production (Section 2.3.1) For the entire section – "each grid cell", does this refer

to the large grid cell or the smaller grid cells within?

L157 – "no specific age of maturity is taken into account". Maturation age has been shown to be one of the most important factors for determining tree migration rates (e.g., Nathan et al. 2011; Snell 2014). This is especially relevant for trees, as most tree species delayed maturation. Please include a justification for why this was not included.

Nathan R, Horvitz N, He YP, Kuparinen A, Schurr FM, Katul GG. 2011. Spread of North American wind-dispersed trees in future environments. Ecology Letters, 14: 211-219.

Snell RS. 2014. Simulating long distance seed dispersal in a dynamic vegetation model. Global Ecology and Biogeography, 23: 89-98.

L159 – please clarify what is meant by "seed bank", and perhaps use another term. In ecology, seed bank has a very specific meaning (i.e., the dormant seeds in the soil that can germinate in subsequent years). How long do seeds stay in this "seed bank"? Does each grid cell have their own seed bank? So seeds enter after dispersal has already occurred? Or is this a central seedbank that all grid cells have access to?

L160-160 – please provide a justification for why you chose the LAI approach for seed production, and not the carbon allocation approach already implemented in LPJ-GUESS. In addition, please include some more information for how LAI is used to determine the number of seeds? What value was chosen for maximum fecundity? Is this species specific?

Seed bank dynamics (Section 2.3.3)

L191-193 – this explanation is not sufficient. What is the difference between yearly loss of germinability and the amount of germinated seeds? Is there a single seed bank for each large grid cell, or each smaller grid cell inside? L194-198 – this is confusing.

Germination (Section 2.3.4)

[Figure]

L202 – 208 – why did you want to add more limitations to establishment? What is the biological justification for this? What does "we fixed this parameter to 0.01 after initial testing" mean? What properties did you evaluate? What does this parameter do? And how does your new limitation interact with the already implemented light limitation (i.e., does this filter happen before or after)?

Corridors (Section 2.5)

This entire section is also very confusing - looking at the figures helped, but the text needs to be clarified. Are these corridors the large grid cells, or the smaller grid cells inside? Or both? L260 – 263 – How is the 1 km scale chosen, appropriate for a species with an average long distance dispersal of 200 m? Only a very, very small proportion of seeds would be able to travel 1 km or more. The next section (L285), mentions "parallel and diagonal corridors". What does this mean? This should be described in this section, with some additional details provided.

Results, Explicit seed dispersal (Section 3.1)

It is not clear what results this section is talking about, nor how it relates to the rest of the manuscript. Referring to "pre-studies" is not helpful (i.e., these results are not part of the current manuscript? So why are they included?).

There are also no values in here at all. How much faster did the FFTM or SMSM perform compared to the explicit dispersal?

Also, this is the first mention of a Matlab script (perhaps should be mentioned in the methods?). Since (I assume) the Matlab script doesn't include the additional processes from LPJ-GUESS, how comparable are these results to what you would get in LPJ-GUESS?

Minor Comments

L34 – not "at least", which implies 1 km or greater. But should be "at maximum" implying that 1 km is the greatest size that can be used.

L 37 – what is "it"?

L39 – what "both methods" are you referring to here? The comparison of the Fast Fourier transformation vs the iteratively shifting seed matrix, or the comparison of between the simulations with all grid cells, versus the corridors?

L39 – what does "reliable" mean?

L59 – awkward wording.

L59-79 – both of these paragraphs are missing appropriate references. They have none, but include several statements which need to be referenced.

L95 – although this is explained in more detail in the discussion, it would be helpful to have this information in the introduction. (i.e., what did previous approaches do, and why were they limiting).

L108 – 110 – a few more details about LPJ-GUESS? This one sentence is vague and particularly unhelpful for understanding what this model does.

L127-128 – if all vegetation is killed, and now seed dispersal is active, you MUST have some vegetation or you won't have any seeds?? Where does the first generation come from?

L142-150 – please clarify this how this occurred. So instead of one grid cell with multiple patches, you simulated one grid cells with multiple grid cells? But these smaller grid cells, had spatial locations and could interact with each other (unlike patches)? This was my interpretation, but this needs to be clearer. A conceptual figure would help.

L188-189 – need more details about how these parameters were "roughly estimated" if this approach is to be applied in other models or for different species.

L286-289 – this sentence is confusing.

[Figure]

Figure 5 is not clear – what is causing the white areas? Neither the results nor the methods addresses what the simulation set up was that could cause this pattern. The explanation "no seeds were able to reach them" is not true, as seeds obviously reached all the way around the white circles in the center (i.e., beech arrived in year 2500, but then never migrated in?).

Numerous small grammatical errors throughout (this is not a complete list, just a few examples). L25

L51-52, "to have a sufficient amount of seeds"

L72 – unnecessary "of"

L86 – unnecessary "the"

L202 – "depending stochastically depending"

L305 – "Using at a distance of"

L347 – "..we are the first that manage to implement. . ."

L369

---

## Author Comment (AC2) · 20 Sep 2018

On behalf of the author team I want to thank the reviewer for the time invested in this extensive review.

Below is a point by point response to the comments raised by the reviewer. We will address the points in a revised version once invited by the editor to provide a new version of the paper.
Overall Comments: The paper describes two approaches, for simulating seed dispersal in global-scale dynamic vegetation models. Vegetation migration in response to climate change (both past and future) is a major area of research, and the ability to simulate dispersal in DGVMs would be a major advance. There is certainly scientific merit in this manuscript, however there are numerous issues that need to be addressed. In general, since this is a paper about model development, more details about the model and justifications for their choices, need to be included.

Major Comments: Seed production (Section 2.3.1) For the entire section – "each grid cell", does this refer to the large grid cell or the smaller grid cells within?

Response: There are no smaller gridcells within gridcells in our model. Whenever we mention gridcell we mean the entire gridcell since we have not subdivided them. We simply use very small (compared to typical LPJ-GUESS uses) gridcells. In each run a gridcell all gridcells have the same size. However, we are only simulating a sparse field of gridcells, meaning that while all gridcells are taken into account for the seed dispersal, some of them (often many of them) are not simulated by LPJ-GUESS-GM, hence all seeds landing there will not germinate. We will stress this out in the paper.

L157 – "no specific age of maturity is taken into account". Maturation age has been shown to be one of the most important factors for determining tree migration rates (e.g., Nathan et al. 2011; Snell 2014). This is especially relevant for trees, as most tree species delayed maturation. Please include a justification for why this was not included.

Nathan R, Horvitz N, He YP, Kuparinen A, Schurr FM, Katul GG. 2011. Spread of North American wind-dispersed trees in future environments. Ecology Letters, 14: 211-219. Snell RS. 2014. Simulating long distance seed dispersal in a dynamic vegetation model. Global Ecology and Biogeography, 23: 89-98.

[Figure]

Response: In our implementation we are not aiming to make a model to fit the real world but rather to see whether our way to implementation leads to comparable results compared to TREEMIG. To be able to compare the results we tried to stick to the way it was implemented in TREEMIG, and there the start of seed production was based on the total height, hence this is what we implemented as well. Height is also a good indicator of maturity. However we agree with the reviewer that age has certainly its merits and might be a more suitable variable in certain conditions. We have in earlier versions used age as a trigger to start seed production. In the next version the user will be able to switch between tree age and tree height as a trigger for seed production. This information is missing in the paper. We will add this reasoning to the paper. The mentioned LAI is used to calculate the amounts of seed similar to Lischke et al 2006.

Comment: L159 – please clarify what is meant by "seed bank", and perhaps use another term. In ecology, seed bank has a very specific meaning (i.e., the dormant seeds in the soil that can germinate in subsequent years).

Response: This is exactly what we mean. We will use this phrase to define seed bank in the text.

Comment: How long do seeds stay in this "seed bank"?

Response: The annual loss of seed germability hence the decay time for the seeds is taken from the values in the cited TREEMIG publication ( for Fagus it is 0.8). It is a species-specific value though currently the values are similar for all since there is little literature comparing the different species.

Comment: Does each grid cell have their own seed bank?

Response: Yes. So seeds enter after dispersal has already occurred?

Response: Yes.

Comment Or is this a central seedbank that all grid cells have access to?

Response: No. Seed bank dynamics (decay, loss due to germination and new arrival due to seed dispersal). Is calculated independently for each gridcell. We have decided to keep the seed bank description short since we took exactly the same approach as in TREEMIG. However we will explain it more exhaustively in the revised version of the paper.

Comment: L160-160 – please provide a justification for why you chose the LAI approach for seed production, and not the carbon allocation approach already implemented in LPJ- GUESS.

Response: The currently implemented carbon allocation in LPJ-GUESS allocates a fixed amount of carbon to reproductive tissue which is then added to the litter pool. Here the basic unit is carbon rather than seed number, which is what we work with. The main reason why we did not go the way to change the amount of carbon allocated to reproductive tissue depending on seed weight and seed number is twofold. Firstly we did not want to change the carbon dynamics within LPJ-GUESS as this is the result of a lot of fine adjustments and any change in the NPP allocation requires a substantial testing afterwards to assure that no unwanted effects occur. Secondly it would have meant that we also need to take the lateral exchange of carbon between cells into account. Currently there are a number of checks that assure that the carbon cycle is closed. Apart from transferring more data between cells, we would have had to adjust these checks as well. So in essence we agree that it would be more reasonable to deduct the carbon for the seeds from the NPP and to transfer it to the adjacent cells (though I guess the amount of carbon actually transported outside a cell is very small compared to the one that stays inside). For the purpose of demonstrating the migration mechanism (focus of this paper) we consider it not necessary but in the next version of LPJ-GM in which we plan to simulate historical tree migration we will implement exactly this. We will comment on this in the paper.

Comment. In addition, please include some more information for how LAI is used to determine the number of seeds? Response: We will add the equation and the

parameters to the paper.

Comment: What value was chosen for maximum fecundity?

Response: We used the same value as used in TREEMIG, we will add them to the paper.

Comment: Is this species specific?

Response: Yes.

Comment: Seed bank dynamics (Section 2.3.3) L191-193 – this explanation is not sufficient. What is the difference between yearly loss of germinability and the amount of germinated seeds?

Response: At the start of each year the amount of seeds that survived the last year is calculated. From this number the amount of germinating seeds is subtracted using a species specific fraction. Then new seeds arrive from the same gridcell and surrounding gridcells that are added to the seed bank the the cycle begins again. We will explain this in more detail in the paper.

Comment: Is there a single seed bank for each large grid cell, or each smaller grid cell inside? L194-198 – this is confusing.

Response: As there are no smaller grid cells within gridcells, there is one seed bank for each grid cell, of which all have the same size.

Germination (Section 2.3.4) Comment: L202 – 208 – why did you want to add more limitations to establishment? What is the biological justification for this? What does "we fixed this parameter to 0.01 after initial testing" mean? What properties did you evaluate? What does this parameter do? And how does your new limitation interact with the already implemented light limitation (i.e., does this filter happen before or after)?

Response: This parameter is a parameter which relates to the area of the gridcell (in which the seeds spread out). In LPJ-GUESS all simulations are done per square meter.

If the same number of seeds land in a larger area there will be less per square meter. As we do not change cell size here we fixed this parameter. Basically LPJ-GUESS simulates a certain amount of seedlings to establish each year depending on the amount of light reaching the forest floor. What we do here is to calculate a probability that the establishment event that LPJ-GUESS simulates happens depending on the amount of seeds available, hence we only decrease LP-GUESS's internal establishment. We will make this more clear in the paper.

Comments: Corridors (Section 2.5) This entire section is also very confusing - looking at the figures helped, but the text needs to be clarified. Are these corridors the large grid cells, or the smaller grid cells inside? Or both?

Response: As there are no small gridcells and large gridcells, but only one size of gridcells, the corridors are those gridcells in which the full LPJ-GM dynamics is calculated. The cells outside the corridors take part in the seed dispersal routine, but no vegetation dynamics is calculated in them. We will make this more clear in the text and mention repeatedly that there is only one size of gridcell however only on those on the corridors the vegetation dynamic is calculated by LPJ-GM.

Comment: L260 – 263 – How is the 1 km scale chosen, appropriate for a species with an average long distance dispersal of 200 m? Only a very, very small proportion of seeds would be able to travel 1 km or more.

Response: Yes it is a small proportion, but it is sufficient to establish the species in the next cell. But we agree, there is a discretization error involved, as in every spatial simulation. We will discuss that.

Comment: The next section (L285), mentions "parallel and diagonal corridors". What does this mean? This should be described in this section, with some additional details provided.

We thought it would be clear when looking at the figures. There are basically two

types of gridcells (all with the same size). First there are the cells for which LPJ-GM calculates full vegetation dynamics and seed production (type 1). Secondly there are cells (type 2) for which LPJ-GM assumes a seed production similar to the nearest neighbor for which full vegetation dynamics is calculated. Hence there is a complete matrix of seed production for which one of the two described algorithms (FFTM and SMSM) is applied to calculate seed dispersal. Only in those cells for which LPJ-GM calculates the vegetation dynamics these seeds can cause trees to establish and to produce new seeds. These two types of cells are arranged in a way that the type 1 cells form a corridor surrounded by type 2 cells. Since the diagonal corridors are also parallel we agree that we wording is unfortunate. We will in the revision call them north-south, east-west, northeast-southwest and northwest-southeast corridors and explain this more extensively

Comment:

Results, Explicit seed dispersal (Section 3.1) It is not clear what results this section is talking about, nor how it relates to the rest of the manuscript. Referring to "pre-studies" is not helpful (i.e., these results are not part of the current manuscript? So why are they included?).

Response: This pre-study is not part of the manuscript but part of the supplementary material. And it is mentioned in the paragraph. The term pre-study is misleading and we will replace it and directly point to the supplement. We will mention the matlab supplement in the Methods where we will add a part about the performance of the different algorithms. We will also highlight the results in the Results section and discuss this in the Discussion section in the revised version.

Comment: There are also no values in here at all.

Response: Running times for FFTMS and SMSM are compared to a large detail in the table 1. We will point to in in the text here as well.

How much faster did the FFTM or SMSM perform compared to the explicit dispersal?

Response: In the early stadium of the work we did an explicit seed dispersal in LPJ-GM. It became quickly obvious that given current computation ability this would not allow us to simulate larger areas. We then developed the two methods (FFTM and SMSM). Re-implementation of the explicit dispersal algorithm only to show that it will be much slower would take a considerable time. Therefore we went a different way, by implementing FFTM and SMSM and explicit dispersal mechanism into a Matlab script. This allowed to concentrate on the running time needed for the dispersal mechanism, which is the since the dispersal algorithm is the focus of this paper. A direct comparison of running times required for the seed dispersal for the different algorithms for different area sizes is done in the script and the results are plotted in the pdf. The script also allows the user to simply cut and past the code into Matlab and play around with it.

Comment: Also, this is the first mention of a Matlab script (perhaps should be mentioned in the methods?).

We will do so. It was actually just meant as an add on to aid explaining the methods since if reading an implementation helps to be implement it in any other model. But since it is also used to evaluate running times we will cover it in the methods and results section in the revised version.

Comment: Since (I assume) the Matlab script doesn't include the additional processes from LPJ-GUESS, how comparable are these results to what you would get in LPJ-GUESS?

Response: In table 1 we are comparing the running times of the FFTM and the SMSM with the running time where seed production is calculated but no seed dispersal is performed, while in the Matlab script we are comparing only the seed dispersal calculations of the two new mechanisms and the explicit mechanism leaving out the vegetation dynamics. Hence though one cannot precisely calculate the difference, one can make a rough estimate. Table one lists the percentage of the time used for the dispersal

for the FFTM method, Supplement 2 shows the increase in computation time for the dispersal algorithm between the FFTM and the explicit seed dispersal hence, by multiplying the time needed for the FFTM win table 1 with the factor from the figure in the Matlab script one can estimate the total difference. So as an example: The simulation of 100*100 cells with LPJ-GM uses in total 1800 cpu*h of which ca 200 cpu*h are used for the calculation of the dispersal (11%; table 1). According to the graph in supplement 2 the explicit simulation needs one order of magnitude longer than the FFTM hence a rough guess would be that instead of 200 cpu*h as used for the FFTM, an explicit seed dispersal would need 2000 cpu*h, which would increase the total required simulation time for 100*100 cells to 4000 cpu*h. One can see in the plot in the supplementary material, that with larger areas, the differences between calculation time of FFTM and supplementary material increases to two orders of magnitude. Hence for larger areas the calculation of the seed dispersal would dominate the required calculation time even more.

We will add this to the discussion.

Minor Comments L34 – not "at least", which implies 1 km or greater. But should be "at maximum" implying that 1 km is the greatest size that can be used.

Response: We will change this

Comment L 37 – what is "it"? It stands for 'simulating the local dynamics' We will change this.

Comment L39 – what "both methods" are you referring to here? The comparison of the Fast Fourier transformation vs the iteratively shifting seed matrix, or the comparison of be- tween the simulations with all grid cells, versus the corridors?

Response: we mean FFTMS and SMSM and will name them in the sentence to make this clear. The corridors are not a method but a way of placing cells.

L39 – what does "reliable" mean?

Response: It means that for both methods (FFTMS and SMSM) comparable results are gained by calculating either corridors or the whole area. We will rephrase this.

L59 – awkward wording. Response: We will rephrase this.

L59-79 – both of these paragraphs are missing appropriate references. They have none, but include several statements which need to be referenced.

Response: The first paragraph expressed mainly the viewpoint of the authors, but we will try to find other papers which expressed similar viewpoints. The second paragraph clearly states facts and we will add the appropriate references in the revised version.

L95 – although this is explained in more detail in the discussion, it would be helpful to have this information in the introduction. (i.e., what did previous approaches do, and why were they limiting).

Response: We had in fact moved the review of other methods from the introduction to the discussion but will provide a small summary paragraph in the introduction section in the revised version.

L108 – 110 – a few more details about LPJ-GUESS? This one sentence is vague and particularly unhelpful for understanding what this model does.

Response: LPJ-GUESS has been described in 200+ publications (though most of them focus on a small side aspect or a new development that did not become part of the standard version). Finding the right amount of information given to the reader is tricky. We will extend the model description.

L127-128 – if all vegetation is killed, and now seed dispersal is active, you MUST have some vegetation or you won't have any seeds?? Where does the first generation come from?

Response: As described in 194-198, for some cells seed limitation does not apply until a certain point in time. These cells are the refugia of the species which have free

establishment. In the example simulations these are the cells in the upper left corner. Hence after the clearing of the vegetation trees can establish freely here, produce seeds that can subsequently disperse to the surrounding cells.

L142-150 – please clarify this how this occurred. So instead of one grid cell with multiple patches, you simulated one grid cells with multiple grid cells? But these smaller grid cells, had spatial locations and could interact with each other (unlike patches)? This was my interpretation, but this needs to be clearer. A conceptual figure would help.

Response: We simulated only one patch per grid cell. However in typical LPJ-GUESS simulations, grid cells are as big as the climate data dictates, e.g. 0.5 up to 2.5 degree longitude/latitude. In our simulation, grid cells are small compared to the standard LPJ-GUESS size. Hence for the area simulated in one standard LPJ-GUESS simulation gridcell (with several patches), LPJ-GM would place many gridcells (with one patch). So there is not one gridcell with multiple gridcells, but just many small grid cells. We thank the reviewer for the suggestion to make a conceptual figure and will provide it in the revised version.

L188-189 – need more details about how these parameters were "roughly estimated" if this approach is to be applied in other models or for different species.

Response: The term 'roughly estimated' indicated that there is a high uncertainty connected to these values. There are parameters estimated for the main European tree species in Lischke et al. 2006.

L286-289 – this sentence is confusing.

Response: we will break up the sentence and make it more clear in the revised version.

Figure 5 is not clear – what is causing the white areas? Neither the results nor the methods addresses what the simulation set up was that could cause this pattern. The explanation "no seeds were able to reach them" is not true, as seeds obviously reached

all the way around the white circles in the center (i.e., beech arrived in year 2500, but then never migrated in?).

Response: The main purpose of the simulation displayed in Figure 5 was to demonstrate a how the method performs when a certain seed dispersability is defined caused by a certain terrain. To demonstrate the effect we created areas in which the dispersability for the seeds were set to zero, hence they could not enter the grid cell. This is shown in the methods section in Figure 2 which shows the probability of entering a new grid cell. In the blue areas in figure 2 the cells have a zero probability of seeds entering from neighboring cells hence seeds can not enter the cell.

In the Results just above Figure 5 we currently write:

"Since the SMSM allows adjusting the probability depending on the seed transport permeability of the terrain we also simulated the migration within a non-homogenous dispersal area. The results of this simulation are displayed in Fig 5."

In the description of the SMSM we write:

If this is done repeatedly <the seed shifting algorithm> it allows an easy implementation of spatial explicit differences in seed dispersal kernel distributions, by adjusting the proportions of seeds being transported into the next cell according to a similarly sized matrix containing the area roughness or permeability. By this approach, barriers and even wind speeds in latitudinal and longitudinal directions can be implemented by adjusting the dispersal probabilities accordingly.

<Figure 2 placed here>

In the results "Simulation experiments" section we describe the simulation run:

"For a specific simulation using the SMSM method we assumed differences in the dispersal ability (e.g. more or less permeable areas or physical barriers) while the climate on all grid cells is still static and favourable.

The reviewer correctly remarks that the information concerning this simulation is spread out, to aid the reader we will refer to this simulation as 'terrain simulation' as it is already called in the Matlab script. We will also add a paragraph in the discussion section explaining that the white features are caused by the low permeability of the cells leading to zero probability of seeds entering the cells and hence to no establishment of trees.

Comment:

Numerous small grammatical errors throughout (this is not a complete list, just a few examples). L25 GMDD Interactive comment L51-52, "to have a sufficient amount of seeds" L72 – unnecessary "of" L86 – unnecessary "the" L202 – "depending stochastically depending" L305 – "Using at a distance of" L347 – "..we are the first that manage to implement. .

Response: We will change these grammatical errors and check the text again for further grammatical issues.

―――――――――――――――――――

---

## Referee Comment (RC2) · Anonymous Referee #2 · 21 Sep 2018

The paper presents two methods for simulating tree species migration, newly implemented in the dynamic global vegetation model LPJ-Guess. I find the paper mostly well written and generally an interesting scientific contribution.

What I, based on the presented material, cannot consent to is the reoccurring statement that the model can be used for continental simulations of multiple interacting species, nor that it is suitable for DGVMs beyond special cases (i.e. species simulations in Europe). Most DGVMs use plant functional types with mixed dispersal/reproduction traits, particularly when used for large spatial applications. The example application deals with only two species, and only the dominant late successional tree species Fagus Sylvatica is tracked, which has a quite narrow dispersal kernel. Furthermore, the application deals with a homogenous landscape. From what the authors show and

write I am not convinced that/how a continental simulation with multiple interacting and dispersing species would/can be possible. From the paper I understood that using FFTM with widely spread transects would not be appropriate in heterogonous areas. SMSM with terrain, on the other hand, would not save enough computation time to be applicable on continental area. Is the plan to use FFTM with transects in homogeneous areas and SMSM in heterogeneous areas? But if so, how would these algorithms then communicate with each other in a continental simulation?

Given that the paper, the presented ideas and the LPJ-GM implementation are already a substantial contribution, I recommend reconsidering the (over?) statements regarding continental applications and DGVMs e.g. in the last sentence of the abstract and particularly the first sentence in the discussion and talk about DVMs with species and spatial extents exceeding applications of a few ha, which is a good and sound contribution. Another way could be closing the explanatory gaps, i.e. (1) discussing issues with DGVMs and how DGVMs, which usually use plant functional types (PFTs) for large scale applications, could be parameterised for the algorithms, (2) discuss the costs/difficulties of an application with a realistic number of interacting species, with differing dispersal traits and (3) explaining how a realistic continental simulation could be assembled with the FFTM and/or the SMSM simulation, given spatial fragmentation and spatial heterogeneity.

General comments

1. In many places in the text the authors state that transect simulations lead to similar/only slightly reduced migration speed. However, from the figures/table it seems to be quite a significant underestimation, and the less transects the worse the underestimation of the migration speed (>20%; i.e. in the 3000y application >600y delay). I recommend stating this clearly and to discuss the consequences.

2. The authors claim that they fulfil the stochastic requirements because they have 200 or so 1km2 grid-cells when comparing to the usual 0.5°x0.5° grid cell. However,

this only holds if the spatial heterogeneity caused by the stochastic disturbances and stochastic mortality does not affect tree species migration. In the example application the authors choose Fagus Sylvatica, a dominant late successional tree species, and I can imagine that for this species the stochasticity might indeed play a minor role. However, what in case of e.g. pioneer, less dominant/more specialised species? These might depend on disturbed areas for establishment – is the transect approach valid for such species? I would find it very helpful to see how the stochasticity and the few available transect cells might affect the spread of such species.

3. It is correct that if applied globally DGVMs usually use 0.5° grid-cells, however when applied as DVMs on continents or regionally the resolution is usually much smaller. See e.g. the dispersal experiments by Snell (2014), and the simulation of European potential natural vegetation with LPJ-Guess (Hickler et al. 2012).

4. I would recommend referring a bit more to relevant literature in some parts of the text, since several of the ideas/methods have already been discussed/used elsewhere. (I mentioned some references in the specific comments list below).

5. What I miss in the current introduction is a bit more on why migration is missing in DGVMs. The authors state that one reason is the '1D' property, i.e. that cells are not interacting and thus the computation costs of making them interacting. But what should also be mentioned is the problem of parameterisation: DGVMs usually use PFTs, often compiled of species with various different traits with respect to migration (dispersal vectors, competitiveness, generation times, ...) (e.g. Snell et al., 2014). Furthermore, if I understood it correctly, the example simulation is for 3000y and the tracked species migrates ∼100km in that time. Several of the criticized studies with DGVMs (e.g. "land use change on vegetation and ecosystem properties") would use well below 3000y; mostly around 100/200y – given the comparable cheapness of 1D simulations and the mentioned constraints due to parameterisation: wouldn't a 'no dispersal' simulation be sufficient for many simulations with large spatial extent and coarse resolution?

6. I would appreciate a more detailed description of the SMSM method. Maybe an illustration? Would this method work with a species with a more pronounced long distance dispersal tail than Fagus Sylvatica? What would this mean regarding computation costs? How to parameterise the SMSM? Could a setting like Fig 5 be simulated with transect at all? Looking at the supplementary figure it seems that the matrix shift method with a terrain has a very small computational gain?

7. I would appreciate more discussion of the limitations and a clearer directive how to apply the algorithms for a continental simulation, if possible. When reading the text I got the feeling that the remedy for the FFTM limitations (heterogeneity/fragmentation, wind directions) is to use the SMSM, but that this method, particularly if used with terrain, is not performant enough for continental applications. Some more buzzwords for the limitation section: parameterisation of SMSM; species parameterisation; fragmentation when using the FFTM; what about ecosystems with many species (i.e.. tropics). Reduction of migration speed by >20%, i.e. in the 3000y simulation > 600y delay.

8. The editor provided me with the model code. Unfortunately I was not able to understand how the simulations were done. There are no hints on how the simulations were conducted, nor was I able to identify the configuration file (instruction script (ins)?) used for the simulations or to find out how/where the transects were defined. I know that it is cumbersome but in the spirit of "good scientific practice" it might be nice to provide and mark the configurations files?

Specific comments:

l.1: Maybe consider to adapt the title, since LPJ-GM does not necessarily lead to a more efficient simulation of migration in dynamic vegetation models per se – e.g.: "Simulating migration in dynamic vegetation models efficiently on the example of LPJ-GM" or maybe better "Simulating migration efficiently in the dynamic vegetation model LPJ-GM"

l.21: Most DGVMs do not use species but plant functional types (PFTs)

l.31: From the last Figure in the supplementary, SMSM with terrain seems to be much slower than FFTM?

l.40: "Furthermore, with the transect method both methods..."?

l.49: DGVMs assume that some instance (i.e. species) of the PFT can establish

l.51: Something seems not correct with the embedded sentence – maybe that instead of the?

l.53 & 63: Anyway DGVMs usually do not simulate species but only PFTs

l.60: When considering ecosystem properties in the future hardly any study would make projections »100y, maybe 200y, but the example in this study uses 3000y. Wouldn't – based on what is shown in this paper – a "no migration between large grid-cells (0.5°) assumption" be appropriate for studies with ~100-200y?

l.76: Another example of a model even accounting for wind speed and direction: LAVESI-WIND (Kruse et al. 2018)

l.82: What does the spatial heterogeneity refers to in this context – soil and climate? If I understood the set-up right LPJ-GM also does not account for such heterogeneities within the grid cell, only to such with regards to species interactions and stochasticity?

l.88-90: e.g. Fisher et al. (2010) call these one-dimensional DVMs

l.94: why every time-step? LPJ-GM only does it once per year?

l.100: If I understood it correctly the presented simulations only simulate two species

l.101: What would a simulation with several species look like, does each need one FFT/SMSM? What are the resulting costs?

l.110-111: Please list a few key references describing LPJ-GUESS 4.0

l.119: Above and below this node is called master

l.123: "no seed dispersal"-> "no seed limitation"?

l.130: There are species producing seeds throughout the year (see e.g. Owens 1994, Brokaw, 1998)

l.140: Here or generally in LPJ-Guess?

l.153-157: How is this similar to Lischke et al., 2006? Lischke et al. (2006) do not mention LAI but state: "The number of seeds S produced per year by each tree depends on its height, species and mast seeding period."?

l.175: For Fagus Sylvatica?!

l.181-182: But wouldn't the implementation of wind direction lead to anisotropy and therefore make the FFTM not applicable anymore (E.g. Neupane (2015))?

l. 185: maybe use $\theta$ and add the $\theta = 1$ in the text below?

l.186: long term -> long distance?

l.197: but how is the number of seeds defined in this case, since in the next para it is stated that the establishment of seedlings depends on the number of available seeds?

l.202: "depending stochastically depending"

l.206: "seed bank per and the germination" remove the per?

l.216: The authors should definitely mention that the method has also already been broadly applied in simulating dispersal. E.g. have a look at Powel (2001) + shortly googling I e.g. found Pueyo et al. (2008) and Prasad et al. (2013) and I assume there are more.

l.235: "different wind distributions" -> only if they are valid for the whole simulated area, or?

l.242: How is this proportion determined?

l.242: 1km2 cell?

l.249: How often is this done/ needs to be done to account for long distance dispersal? What happens with the seeds at the boundary of a simulation area?

l.255: Figure 2 is not cited in the main text (only in Fig. 5).

l. 266: Wouldn't the heterogeneous landscape be much more crucial to test the applicability of the methods?

l.274-275: And? But?

l.276: Out of interest: how many CPUs were used/ what computing environment? Would it be possible to add a 'no dispersal and no communication' (i.e. a 1D) simulation for comparison?

l.303: 1km2 grid cells

l.306: somewhat? >20%!

l.309: Maybe add the numbers for the variability

l.312: Which probably also explains the patterns in the migration front?!

l.314: When FFT when FFTM?

l.318-319: How do the simulations compare to a simulation without communication between grid cells, i.e. 1D simulations?

l.323: How to specify this parameter when not having a FFTM simulation at hand? How for SMSM with terrain, does this require a simulation without terrain before? What are the cost reductions then?

l.335-337: I would find it valuable to have the simulation times for the terrain simulation in the table, too!

l.341: This would probably not work with transects?!

l.358: K is the number of iterations?!

l.361: "a very similar migration pattern" I would delete the "very"

l.362: it is slower by 20%!

l.362-363: in l. 310-312 the authors state that its slower because of the migration path? How do these two different explanations contribute?

l.364: how to parameterise "explicit considerations of wind directions"

l.376: Something is missing in this sentence

l.379: Maybe in a DVM? But not in ecology and not to simulate dispersal; the authors should mention some applications - as mentioned above: have a look at Powell (2001) + other references such as Pueyo et al. 2008; Prasad et al. 2013 + I imagine there are much more.

l.384: "DGVMSs"

l.410-412: 63-85% instead if 85%? I would maybe remove this quantitative comparison. In my understanding the size of the reduction will be dependent on the model and the set-up of the simulations, i.e. on a variety of factors, such as the number of simulated and dispersing species, the resolution, settings of the applied algorithms, etc., and since its two different models and probably very different simulation set-ups, it seems to me to be comparing apples and oranges?

l.413: more pronounced than what or where?

l.416: maybe 0.5 and 1.0?

l.449: From the last Figure in the supplementary it seems that SMSM simulations with terrain are comparably much slower. Is it possible to speed them up with transects?

l.462: I would not call 20% slightly

l.457: "FTTM" -> "FFTM"

l.465: unfortunate – I think this would be really interesting, especially when simulating fragmented landscape or non-dominant species

l.481: missing ")"

l.486: What was the tested set-up? I assume FFTM? I.e. no 'terrain'? Transects with 50km distance? How many competing/migrating species? All grid cells homogeneous? How many years?

l.488: "considerable computation costs" – what were they in the tests (e.g. CPU h per simulated y)? Are continental applications possible, or are they not possible?

l.488: plural and singular mix: "a high ... amounts" + what does "of the FFT as the local simulations" mean?

l.498: what do you mean with "truly mechanistic"? I recommend deleting this statement

l.503: "related estimates the Conclusion section"?

l.613: Again "10" + something with the formatting

l.627: Please provide a legend – even if the figure is only schematic

l.635: When looking at the Figure and reading 2.4 I wondered where the 5*10-7 came from and how this parameter is determined? – Finally I found some information in 3.3 Fig.3 and 4: I would appreciate if the y-axis of the distance plots on the right would have similar scales, this would really help for comparison

l.647-648: difficult sentence – maybe: "only taken into account for grid cells ..."?

l.661: Comparing the dark blue spots in Fig . 2 and the white ones in Fig. 5 the Figures seem to be mirrored along the diagonal?

l.664: cpu*h = CPU h?

l.671: FFTM with 10: shouldn't this be 64% instead of 67%?

References

Brokaw, N. V. (1998). Cecropia schreberiana in the Luquillo mountains of Puerto Rico. The Botanical Review, 64(2), 91-120.

Fisher, R., McDowell, N., Purves, D., Moorcroft, P., Sitch, S., Cox, P., Huntingford, C., Meir, P., and Woodward, F. I. (2010). Assessing uncertainties in a second-generation dynamic vegetation model caused by ecological scale limitations, New Phytol., 187, 666–681.

Hickler, T. , Vohland, K. , Feehan, J. , Miller, P. A., Smith, B. , Costa, L. , Giesecke, T. , Fronzek, S. , Carter, T. R., Cramer, W. , Kühn, I. and Sykes, M. T. (2012). Projecting the future distribution of European potential natural vegetation zones with a generalized, tree species‐based dynamic vegetation model. Global Ecology and Biogeography, 21: 50-63.

Kruse, S., Gerdes, A., Kath, N. J., and Herzschuh, U. (2018). Implementing spatially explicit seed and pollen dispersal in the individual-based larch simulation model: LAVESI-WIND 1.0, Geosci. Model Dev. Discuss., https://doi.org/10.5194/gmd-2018-31, in review.

Neupane, R. C. (2015). Modeling Seed Dispersal and Population Migration Given a Distribution of Seed Handling Times and Variable Dispersal Motility: Case Study for Pinyon and Juniper in Utah. Owens, J. N. (1995). Constraints to seed production: temperate and tropical forest trees. Tree Physiology, 15(7-8), 477-484.

Powell, James (2001). Spatio-temporal models in ecology; an introduction to integro-difference equations. Utah State University, Logan (UT).

Prasad, Anantha M., et al. (2013). Exploring tree species colonization potentials using a spatially explicit simulation model: implications for four oaks under climate change. Global Change Biology 19.7, 2196-2208.

Pueyo, Y., et al. (2008). Dispersal strategies and spatial organization of vegetation in

arid ecosystems. Oikos 117.10, 1522-1532.

Snell, R. S. (2014), Simulating long-distance seed dispersal in a dynamic vegetation model, Glob. Ecol. Biogeogr., 23(1), 89–98, doi:10.1111/geb.12106.

Snell, R. S., Huth, A., Nabel, J. E. M. S., Bocedi, G., Travis, J. M. J., Gravel, D., Bugmann, H., Gutiérrez, A. G., Hickler, T., Higgins, S. I., Reineking, B., Scherstjanoi, M., Zurbriggen, N. and Lischke, H. (2014), Using dynamic vegetation models to simulate plant range shifts, Ecography (Cop.)., 37(12), 1184–1197, doi:10.1111/ecog.00580.
* * *

---

## Referee Comment (RC3) · Anonymous Referee #3 · 25 Sep 2018

Lehsten et al. present a nice and timely study focusing on the implementation of migration into dynamic global vegetation models. They show a way to connect established assumptions of seed dispersal based on former studies with two approaches of enhanced seed dispersal based on Fast Fourier transformation and seed matrix shifting. Together with allowing seed dispersal through specified spatially equidistant corridors they show a nice way of how to reduce computation time while losing some accuracy even though there is no real validation presented. The approach has the potential to be applied in the different DGVMs existing today and is an important contribution to their development. My comments mainly concern 1) the reproducibility of their method and 2) the realism of being able to conduct continental scale simulations.

General comments:

[Figure]

Computation time: My biggest concern here is that the authors simulated only one species migrating. Therefore, they were able to simulate only one growing patch per 1 km2 grid cell. When increasing the species number it is definitively important to also increase the number of growing patches (and probably decrease their size to enable local competition or in other words to avoid an unrealistic growing patch overarching competition which is one of the major selling points of a gap model like in LPJ GUESS typically using 100-1000 m2 growing patches) in return increasing the computation time. The computation time for a continental scale simulation with many different species still has to be determined and could be topic of a follow up study. Germination rate: It is unclear how sensitive the presented results are in connection to the germination rate used. Obviously the germination rate must influence the speed of migration. The rate of germination directly influences the competitiveness of each species and therefore its dispersal. Age of maturity: Even though I am totally ok with not taking into account an age of maturity to keep the findings of this study as simple as possible, it is again very obvious that this variable strongly influences the speed of migration. Therefore, this topic needs an extra space in the discussion or some results in the supplement showing e.g. the influence of assuming a minimum age of maturity.

Specific comments:

Line 132-134: How does the seedbank determine establishment probability and how is environmentally-suitable determined? I would like at least a brief explanation of this crucial aspect.

Line 136 – 150:

a) So what I see in figure 3 and 4 is that for the 50km corridor approach you have 6 corridors per 0.5° grid cell (or do neighboring cells share corridors)? And these corridors need 200 simulation cells each of them 1km$^2$ in size? Assuming that a 0.5° grid cell is 50x50km I wonder where the positions of your corridors actually are. At the boarders of each grid cell and also diagonal through the middle? It would be helpful to

see this in Fig. 3 and 4.

b) You are able to use only 1 patch per cell, because you are only simulating 1 species migrating. It is important to explain in the discussion that you definitely need more (and probably smaller) patches if you consider more species. It actually scales with species number. Therefore, computation time would be much higher as well. This is contradicting potential continental simulations.

Line 155-157: Even though you cite Lischke et al. I would like to see a brief explanation of the "maximum fecundity" method.

Line 157: Have you performed tests for age of maturity? I guess setting an age of maturity would lower the speed of migration. I am totally ok with not taking this into account, but it would be good to pick up this issue in the discussion e.g. under 4.4.3.

Line 163-164: Please provide explanation and reference for mast fruiting effects.

Line 188 – 189: So do you use the values for Fagus sylvatica?

Line 191 – 193: Where can I find values of "loss of germinability"? If these are specific values from Lischke et al. I would suggest to list them in a table as well as similar parameter values. This would really help to reproduce the study.

Line 194-198: I have my problems understanding this whole part. 1. "A year is defined for each species and grid cell before which seed bank constraints are ignored". I do not understand this sentence. 2. I also do not understand the second sentence. I believe you talk about the initial conditions and refugia. It is probably a very crucial part for migration simulations so please provide a few more sentences of explanation.

Line 207: Explain "age cohort". It has not appeared before and is important to understand the approach.

Line 199-208:

a) Please provide equations and an according explanation instead of an example in

line 208.

b) Moreover, I am quite sure that the germination rate strongly influences results. It is probably important for the speed of migration and definitively for competition and therefore equilibrium biomass. What do you mean with "initial testing"? I don't expect a comprehensive full explanation in the text, but I would like to understand why you have chosen certain parameter values.

Line 302 – 303: As suggested above. Please provide the numbers of your parametrization.

Line 442 -443: Is it possible to give a comparison here? What is the computation time for the same setup with standard LPJ-GUESS? I see your comparisons in table 1, but they all refer to simulations which use a master.

Line 488 – 490: Have you estimated the CPU hours for this setup? Would be an interesting information.

Technical comments:

Line 22-24: Indicate that this sentence is about plants in the real world

Line 62-64: For me this is one of the major selling points and I would put it in the abstract as well. You decide.

Line 202: 2 times the word "depending"

Line 203 -205: Confusing sentence. "The probability that a species establishes is proportional to the seed number in the seed bank multiplied by . . .". Wrong formulation.

Line 206: The word "year" is missing.

Line 242: I would not expect that every reader knows what a Moore neighborhood is?
* * *

---

## Author Comment (AC3) · 9 Oct 2018

Dear reviewer, we want to thank for the time spend for this thorough review. Please find a step by step response below. All line numbers will be entered once the final version of the ms is ready.
Comment: What I, based on the presented material, cannot consent to is the reoc-curring state- ment that the model can be used for continental simulations of multiple interacting species, nor that it is suitable for DGVMs beyond special cases (i.e. species simulations in Europe).

Response: We agree that the performance of the dispersal algorithm as presented cre-ates a doubt that a continental simulation is possible. Given that we clearly make this claim should require for us to present an algorithm that would have a suitable perfor-mance. The main aim of this paper was to introduce the two algorithms into the DGVM and to perform continental scale simulations in a upcoming work. Hence we did not in-vest a lot of time into optimization, except for a paragraph in the discussion for simplicity of the paper. We have now used two more options to optimize the speed that we evalu-ate in the Matlab script which allowed us to increase the performance by more than an order of magnitude of the FFTM versus the explicit simulation (one of the improvements was already part of the LPJ-GM simulations and was not included in the Matlab script for readability),while the other (which only improves by 30% to 50%) is currently not implemented. We will rephrase all statements where we state that our implementation can be used for continental applications and write that it has the potential to be used for large areas and also has a lot of potential for performance improvements. Again the intention was to present the two algorithms for seed dispersal, while any continental scale simulation experiments would require to also present a completely new param-eterization of some of the trees, as well as many other aspects hence we would like to refrain from performing continental scale simulations for now. We completely agree with the second statement. Given that we worked a lot in the past with species simula-tions we completely forgot that the currently most common application of DGVMs uses PFTs and there it is not necessary to include seed dispersal.

Comment: Most DGVMs use plant functional types with mixed dispersal/reproduction traits, particularly when used for large spatial applications. The example application deals with only two species, and only the dominant late successional tree species Fagus Sylvatica is tracked, which has a quite narrow dispersal kernel. Furthermore, the application deals with a homogenous landscape. From what the authors show and write I am not convinced that/how a continental simulation with multiple interacting and dispersing species would/can be possible. Response: Again we completely agree that only for simulations on the species level, an inclusion of seed dispersal is useful. Currently all dispersal simulation is performed at the master node, if the dispersal simulations for each species are performed at one node per species there should be (theoretically) no reasonable reduction in performance. We also repeat that we have not proven that we can simulate at a continental scale, but only that we are 2-3 orders of magnitude faster in simulating seed dispersal compared to the explicit simulation. Whether this is sufficient for a continental simulation will be shown in successive work and we will rephrase this claim.

Comment: From the paper I understood that using FFTM with widely spread transects would not be appropriate in heterogonous areas. SMSM with terrain, on the other hand, would not save enough computation time to be applicable on continental area. Is the plan to use FFTM with transects in homogeneous areas and SMSM in heterogeneous areas? But if so, how would these algorithms then communicate with each other in a continental simulation? Response: Though not formulated in the paper (for simplicity) yes this is the main idea. Depending on the parameters of the species specific dispersal kernel, there is a maximum distance that the seeds are transported (theoretically there is no such limit, but given the strong decrease of the tail, this assumption has no influence on the final result). One option is to define a certain area as heterogeneous. The seeds produced in this area are dispersed by the SMSM algorithm, while the seeds of the remaining area are dispersed by the FFTM. Though for both the seed production is only taken into account for the assigned areas the seed fall will be calculated for the area plus an edge surrounding the area with the width of half a maximum kernel width. In a last step the dispersed seeds of the two methods are added to a final distribution of seeds. This way there is no complicated communication of the two algorithms, but the edges of the areas that are simulated overlap while the areas of the

seed production that go into the algorithm do not. Since the reviewer asks for it we will present this reasoning in the Discussion.

Comment: Given that the paper, the presented ideas and the LPJ-GM implementation are already a substantial contribution, I recommend reconsidering the (over?) statements regarding continental applications and DGVMs e.g. in the last sentence of the abstract and particularly the first sentence in the discussion and talk about DVMs with species and spatial extents exceeding applications of a few ha, which is a good and sound contribution. Response: We will tune down our statements and clearly show what level of areal coverage we show the algorithms being able to simulate.

Comment: Another way could be closing the explanatory gaps, i.e. (1) discussing issues with DGVMs and how DGVMs, which usually use plant functional types (PFTs) for large scale applications, could be parameterised for the algorithms, (2) discuss the costs/difficulties of an application with a realistic number of interacting species, with differing dispersal traits and (3) explaining how a realistic continental simulation could be assembled with the FFTM and/or the SMSM simulation, given spatial fragmentation and spatial heterogeneity. Response: We will do this in the Discussion. General comments 1. In many places in the text the authors state that transect simulations lead to similar/only slightly reduced migration speed. However, from the figures/table it seems to be quite a significant underestimation, and the less transects the worse the underestimation of the migration speed (>20%; i.e. in the 3000y application >600y delay). I recommend stating this clearly and to discuss the consequences. Response: An underestimation of 20% might look great but given that most current approaches completely ignore seed dispersal and also that the parameterization of the seed dispersal kernel comes with quite an uncertainty puts the 20% in perspective. The currently reached migration speeds are very likely to be too low as described in the Discussion, which is probably caused by uncertainties in the parameterization of the seed dispersal kernel. However as the aim was to implement the two methods in a DGVM we aimed to keep the kernel similar to the parameterization within TREEMIG to be able to compare

results.

Comment: 2. The authors claim that they fulfil the stochastic requirements because they have 200 or so 1km2 grid-cells when comparing to the usual 0.5 x0.5 grid cell. However,this only holds if the spatial heterogeneity caused by the stochastic disturbances and stochastic mortality does not affect tree species migration. In the example application the authors choose Fagus Sylvatica, a dominant late successional tree species, and I can imagine that for this species the stochasticity might indeed play a minor role. However, what in case of e.g. pioneer, less dominant/more specialised species? These might depend on disturbed areas for establishment – is the transect approach valid for such species? I would find it very helpful to see how the stochasticity and the few available transect cells might affect the spread of such species. Response: Pioneer species are typically fast migratory species, hence they would typically be able to colonize the area before the late successional species arrives (if both occupy the same climate space). If on the other hand the late successional species has a larger climate range you are correct that the early successional species will be hindered in its migration into the few spots that are available over a short time. This is true for both a full simulation as well as a simulation along transects. The fact that the seed survival of early successional species is typically higher due to the lower seed mass, should allow the spread still in the small temporal successional gaps. Again the focus here is to introduce the methods. A parameterization of the species that results in a migration speed comparable to observed values is outside the range of this study. However we agree that this is an important point and will discuss it in the limitations and further work section. Comment: It is correct that if applied globally DGVMs usually use 0.5 grid-cells, however when applied as DVMs on continents or regionally the resolution is usually much smaller. See e.g. the dispersal experiments by Snell (2014), and the simulation of European potential natural vegetation with LPJ-Guess (Hickler et al. 2012). 4. I would recommend referring a bit more to relevant literature in some parts of the text, since several of the ideas/methods have already been discussed/used elsewhere. (I mentioned some references in the specific comments list below). Response: The

maximum resolution is dictated by the climate data available. Given that LPJ-GUESS is parameterized using CRU climate data the simulations at coarse scale are typically performed at 0.5 degree or sometimes using the CRU climatology to bias correct the CRU timesseries it is performed at 0.25 or 0.1 degree. Even when run with 10 patches and at 0.1 degree would result in 250 patches per 0.5 degree which is comparable to the 200 simulations that we perform at any 50 by 50 km cell. However we will mention that non-global simulations and especially regional simulations use finer grid cells.

Comment: What I miss in the current introduction is a bit more on why migration is missing in DGVMs. The authors state that one reason is the '1D' property, i.e. that cells are not interacting and thus the computation costs of making them interacting. But what should also be mentioned is the problem of parameterisation: DGVMs usually use PFTs, often compiled of species with various different traits with respect to migration (dispersal vectors, competitiveness, generation times, ...) (e.g. Snell et al., 2014). Response: We will mention this.

Comment : Furthermore, if I understood it correctly, the example simulation is for 3000y and the tracked species migrates 100km in that time. Several of the criticized studies with DGVMs (e.g. "land use change on vegetation and ecosystem properties") would use well below 3000y; mostly around 100/200y – given the comparable cheapness of 1D simulations and the mentioned constraints due to parameterisation: wouldn't a 'no dispersal' simulation be sufficient for many simulations with large spatial extent and coarse resolution? Response: As already mentioned before (and highlighted in the Discussion) the migration speed that we are calculating are way too small compared to measured values. The aim of this paper was to introduce the method and here we choose to measure our success in response to the migration speed simulated by TREEMIG since we used a similar dispersal kernel. Any real world application will require a new parameterization of the kernel to gain a realistic speed. Currently most simulations are not 'no dispersal' simulations but 'extremely fast dispersal' simulations (given that they use free establishment). Otherwise they could not show any response

of vegetation on climate. For short term studies where the time horizon is well below the generation time of the species there is a limited use of a dispersal kernel. Especially for Europe where land use is dominating the vast majority of the landscape and at least in the northern part also plants (often alien) species in forest any kind of simulations assuming semi-natural conditions are questionable. However to understand current tree distribution in those parts which are still semi-natural and especially to understand forest species history taking seed dispersal into account might be important. We will mention this reasoning in the Discussion.

Comment: I would appreciate a more detailed description of the SMSM method. Maybe an illustration? Would this method work with a species with a more pronounced long distance dispersal tail than Fagus Sylvatica? What would this mean regarding computation costs? How to parameterise the SMSM? Could a setting like Fig 5 be simulated with transect at all? Looking at the supplementary figure it seems that the matrix shift method with a terrain has a very small computational gain? Response: Thank you for suggesting to add a figure that will illustrate the SMSM, we will do so in the revised version. In general the SMSM method is a relatively direct implementation of seed dispersal, by moving seeds from one cell to another with a certain probability. Comment: I would appreciate more discussion of the limitations and a clearer directive how to apply the algorithms for a continental simulation, if possible. When reading the text I got the feeling that the remedy for the FFTM limitations (heterogeneity/fragmentation, wind directions) is to use the SMSM, but that this method, particularly if used with terrain, is not performant enough for continental applications. Some more buzzwords for the limitation section: parameterisation of SMSM; species parameterisation; fragmentation when using the FFTM; what about ecosystems with many species (i.e.. tropics). You read correctly that FFTM is not able to handle landscapes resulting in heterogeneous seed dispersal while SMSM has strong performance constrains. As a matter of fact, the FFTM is still applicable if the barrier is larger than the kernel width since it will place seeds there but if the cells are not suitable then the seeds will not germinate (Baltic sea, alps). The SMSM is only required in areas where we have (or rather know) different

dispersal tail lengths depending on the terrain. This might be the case in some valleys in the alps, where seed dispersal acts mainly along the valleys, but not the mountain, but given the typical resolution of the output for continental studies this might not be necessary to apply the SMSM at all, while for finer scale studies SMSM might be the best choice. We will discuss this in the Discussion section

Comment: Reduction of migration speed by >20%, i.e. in the 3000y simulation > 600y delay. 8. The editor provided me with the model code. Unfortunately I was not able to understand how the simulations were done. There are no hints on how the simulations were conducted, nor was I able to identify the configuration file (instruction script (ins)?) used for the simulations or to find out how/where the transects were defined. I know that it is cumbersome but in the spirit of "good scientific practice" it might be nice to provide and mark the configurations files? Response: The transects are defined in the gridlist. Basically while a typical LPJ-GUESS gridlist contains only columns one for the longitude and one for the latitude, in LPJ-GM there are additional columns in which for each species a time is given in which free establishment is allowed see below for the start of a gridlist. 23 50 TeBS,101.0,IBS,101.0 23.01 50 TeBS,100.0,IBS,10000.0 23.02 50 TeBS,100.0,IBS,10000.0 The first line indicates that at position 23 degree longitude, 50 degree latitude both species TeBS (temperate broadleaved Summer green tree or beech and IBS Intermediate shade tolerant broadleaved Summer green tree; or birch) are allowed free establishment (hence no seed limitation) at the year 101, which is one year after the initialization phase for nitrogen initialization. Hence this cell would form a refugia for beech. The next cell is located at 23.01 degree longitude and 50 degree latitude and beech is only allowed free establishment after the year 10000, hence not within the simulated time of 3000 years, it can only establish at this site if seeds arrive there. Birch is allowed to establish at this site without seed limitation. The transects are defined in a way that only the cells that are on the transects are listed in the gridlist.txt. This way of defining them might not be the most elegant one, but since the current setup of LPJ-GUESS simply cuts the gridlist into as many pieces and distributes them into different directories in which the simulation is performed, this way

I did not had to read in a separate file for the refugia definition, and I am sure that the information is linked to the gridcell. The configuration script (the ins file) is similar to the one used in Hickler et al. except that it contains these additional entries at the global level. ! migration INSTRUCTION years_total 3000 ! How many years the dispersal simulation is performed domain 23 50 0.01 0.01 ! which domain is simulated and with what resolution param "size_lat" (num 100) ! how many cells are in the domain along the latitude param "size_lon" (num 100) ! how many cells are in the domain along the longitude dispersal_patchsize 0.99 ! How big a single patch is. if_dispersal_fft 1 ! whether FFTM dispersal is performed if_dispersal_float 0 ! whether SMSM dispersal is performed if_dispersal_ext_fft 0 ! whether another variant of FFTM (not described in the paper) is performed stochastic_seed_est_scaler 0.01 ! scaler for the patch size output_interval 10 ! in years save space since not all years are needed in the output

Each species contains the following extra parameters which are taken from TREEMIG(here are the values for beech): max_fecundity 29. ! maximum fecundity min_height_for_maturity 14.4 ! minimum height for maturity germination_rate 0.3 ! rate of seeds germinating per year max_seed_age 3.3 ! maximal survival times for seeds in seed bank short_range_disp_frac 0.99 ! fraction of seeds that go into short seed dispersal short_disp_alpha 25 ! parameter for short distance dispersal long_disp_alpha 200 ! parameter for long distance dispersal

I also would prefer to make the whole model code publicly available. However current policies within the modelling consortium only allows to give access to model code after individual contact with the author. I decided that my unit containing the implementation of the code for the actual migration will be made publicly available (as a supplement to this paper), but there are of course some other small bits and technical issues, like for example the MIP related code that is located in other units.

Specific comments: l.1: Maybe consider to adapt the title, since LPJ-GM does not necessarily lead to a more efficient simulation of migration in dynamic vegetation models per se – e.g.: "Simulating migration in dynamic vegetation models efficiently on the

example of LPJ- GM" or maybe better "Simulating migration efficiently in the dynamic vegetation model LPJ-GM" Response: Yes we will change the title to "Simulating migration efficiently in the dynamic vegetation model LPJ-GM 1.0" Comment: l.21: Most DGVMs do not use species but plant functional types (PFTs) Response: We will mention this. Line. . . Comment: l.31: From the last Figure in the supplementary, SMSM with terrain seems to be much slower than FFTM? Response: Yes it is especially now that we have optimized the Matlab code (though at the expense of readability) it is. We will delete the word 'marginally'. However it is still faster than an explicit seed exchange. Line : . . . Comment: l.40: "Furthermore, with the transect method both methods"? Response: We will introduce the 'with the transect methods' . We will replace continents with large regions, since we have not really shown that continents can be simulated with our method. Line:. . . Comment: l.49: DGVMs assume that some instance (i.e. species) of the PFT can establish

Response: We will add a sentence before stating that while most DGVM applications use PFTs which are not suitable for the seed dispersal simulations due to different seed dispersal mechanics within the same PFT, we are concentrating here on applications simulating explicit tree species. Line: . . .

l.51: Something seems not correct with the embedded sentence – maybe that instead of the? Response: No here that would give a different meaning. We inserted an 'a' and hope the sentence is now easier to read. Line:. . .. Comment: l.53 & 63: Anyway DGVMs usually do not simulate species but only PFTs Response: Since we already write in line . . .. that we are only considering species simulating DGVMs we consider this covered.

Comment: l.60: When considering ecosystem properties in the future hardly any study would make projections Âż100y, maybe 200y, but the example in this study uses 3000y. Wouldn't – based on what is shown in this paper – a "no migration between large grid-cells (0.5 åŮ̧ ) assumption" be appropriate for studies with âĹij 100-200y? Response: We mention this limitation at line :. . .. "However, given that most studies using future

climate simulate only 50 or 100 years ahead, which is way below the generation time of trees, and because of human activities which plant many tree species outside its native range, the use of explicit modelling of seed dispersal in DGVMs might be limited for studies of future tree distribution." Comment: l.76: Another example of a model even accounting for wind speed and direction: LAVESI-WIND (Kruse et al. 2018) Response: Thanks for pointing us to the paper, we are citing it now in line …. Comment: l.82: What does the spatial heterogeneity refers to in this context – soil and climate? If I understood the set-up right LPJ-GM also does not account for such heterogeneities within the grid cell, only to such with regards to species interactions and stochasticity? Response: Yes: Soil and climate, mountains blocking seed transport as well. This sentence should simply highlight that a simple transfer of migration speeds calculated with models at fine scale into models at coarse scale is challenging.

Comment: l.94: why every time-step? LPJ-GM only does it once per year? Response: Yes of course we mean annually. We change this at line ….

Comment: l.100: If I understood it correctly the presented simulations only simulate two species. Response: Yes in our example simulation only two species are simulated but the method can simulate more species in a real application case.

Comment: l.101: What would a simulation with several species look like, does each need one FFT/SMSM? What are the resulting costs? Response: In the current simulation time that we present in the table we are actually simulating the seed dispersal of both species independently (though it would of course be faster to only simulate one species). Yes each species needs its own FFTM or SMSM to be performed if the migration of several species is to be evaluated, however they could be potentially performed at separate nodes which would decrease calculation time again. We are mentioning this in the section where we discuss the performance (Line 453 to 455 in the old ms). Comment: l.110-111: Please list a few key references describing LPJ-GUESS 4.0 Response: We included Smith 2014 and Lindeskog 2013 which are the main references describing the 4.0 version. Comment: l.119: Above and below this

node is called master Response: We now call it master here as well. Comment: .123: "no seed dispersal"-> "no seed limitation"? Response: Yes we added this as well. Line . . . l.130: There are species producing seeds throughout the year (see e.g. Owens 1994, Brokaw, 1998) Response: Yes this is one of the discretization errors that we have to make. Given unlimited computing power and knowledge of weather conditions and plants behavior, we would perform the FFTM or SMSM daily over the time when seeds are produced. However, as a first improvement of the situation in which most models do not consider seed dispersal at all, we suggest to simulate at an annual time step. Comment: l.140: Here or generally in LPJ-Guess? Response: It is variable but this is the recommended size. Comment: l.153-157: How is this similar to Lischke et al., 2006? Lischke et al. (2006) do not men- tion LAI but state: "The number of seeds S produced per year by each tree depends on its height, species and mast seeding period."? Response: We also use the height of maturity, but no mast seeding period, while Lischke et al. scaled the seeds with height we did scale them with LAI, you are correct this is not the same and we have taken that sentence away as it was meant introduce into the chapter but it is necessary. Comment: l.175: For Fagus Sylvatica?! Response: Added. Actually we also simulated seed dispersal for Birch but since Birch is set to no seed limitation that does not matter. Comment: l.181-182: But wouldn't the implementation of wind direction lead to anisotropy and therefore make the FFTM not applicable anymore (E.g. Neupane (2015))? Response: The FFTM can apply any shape of seed dispersal kernel, it can just not change it with the landscape. Hence certain wind directions are possible like the kernel used for illustration, which is also skewed, for example by wind. Neupane simualates effects of the landscape on fruit dispersing birds. Such an effect would have to be modelled by the SMSM. If the different wind directions in different parts of the domain (e.g. caused by a certain terrain) is to be taken into account, this also needs to be done by SMSM.

l. 185: maybe use $\theta$ and add the $\theta = 1$ in the text below? We will adjust this equation. Comment: l.186: long term -> long distance? Response: Changed. Comment: l.197: but how is the number of seeds defined in this case, since in the next para it is stated

that the establishment of seedlings depends on the number of available seeds? LPJ-GUESS calculates the number of established individuals per species depending on the light reaching the forest floor. LPJ-GM takes this value and either and sets it to zero depending on the presence (or rather absence) of seeds. In case of establishment free from seed limitation (in our case the birch), this step is not performed. Hence the species can always establish depending only on the light reaching the forest floor. Comment: l.202: "depending stochastically depending" Response: Thanks for spotting this repetition. It is fixed. Comment: l.206: "seed bank per and the germination" remove the per? Response: Done Comment: l.216: The authors should definitely mention that the method has also already been broadly applied in simulating dispersal. E.g. have a look at Powel (2001) + shortly googling I e.g. found Pueyo et al. (2008) and Prasad et al. (2013) and I assume there are more. Response: We are now mentioning that there are a number of applications which already use ffts to simulate dispersal and cite a few of them. Line . . . .

Comment l.235: "different wind distributions" -> only if they are valid for the whole simulated area, or? Response: We added this remark in line: . . . . l.242: How is this proportion determined? A few lines later we point to a derivation of the parameters (in this case this proportion) in the supplementary material S.1.

Comment: l.242: 1km2 cell? Response: Yes in our application all cells have one km^2 extent. Comment: l.249: How often is this done/ needs to be done to account for long distance dispersal? Response: Currently this is done 10 times hence we are reaching a maximum of 10 km. Comment: What happens with the seeds at the boundary of a simulation area? Response: For both the FFT as well as the SMSM simulation we extend the area by one kernel width to avoid / minimize edge effects. Basically all seeds that land of the seed domain are lost.

Comment: l.255: Figure 2 is not cited in the main text (only in Fig. 5). Response: Thanks for spotting this. This sentence must have gotten lost in one of the internal revisions. We now refer to Figure 2 in the description of the simulations. Line . . . .

Comment: l. 266: Wouldn't the heterogeneous landscape be much more crucial to test the appli- cability of the methods? Response: No the idea is to only use the corridors in homogenous landscapes and to speed up the simulation there. In heterogeneous landscapes this simplification is not suitable. Hence we only test the corridors in homogenous landscapes. Comment: l.274-275: And? But? Response: And we do not want to strongly increase the migration speed. We have spent a lot of time trying to come up with a better solution like some kind of weighted average, however so far we have not found one. Hence we prefer to have a reduction by 20% hence a conservative estimate rather than a strong increase which also was heterogeneous within the simulated area depending on the arrangement of the corridors. We have not given up the hope to come up with a better solution in a real world application. Comment: l.276: Out of interest: how many CPUs were used/ what computing environment? Response: We used 200 nodes (with 20 nodes per CPU) at the LUNARC computing facilities. Comment: Would it be possible to add a 'no dispersal and no communication' (i.e. a 1D) simulation for comparison?

Response: Yes we are working on it and will add it to the final version of the paper

Comment: l.303: 1km2 grid cells Response: We added this. Comment: l.306: somewhat? >20%! Response: We have removed the somewhat. Comment: l.309: Maybe add the numbers for the variability Response: What we mean here is visual realization that the distance of the points increases. Since we do not use a mean value to estimate the migration front, it is hard to quantify this variability since we have a different variability above compared to below the line Comment: l.312: Which probably also explains the patterns in the migration front?! Response: Yes that is the reason. Comment: l.314: When FFT when FFTM? Response It should always be FFTM. Thanks for spotting this. Comment: l.318-319: How do the simulations compare to a simulation without communication between grid cells, i.e. 1D simulations? Response: We are currently running this simulation and will add this comparison in the table in the final paper. Comment: l.323: How to specify this parameter when not having a FFTM simulation at

hand? Response: Here the aim was to parametrize the SMSM in a way that we have a similar migration speed compared to FFTM. In a practical application one would have a certain dispersal kernel and the derivation in Supplement 2 allow to estimate the parameter to fit a Gaussian kernel. It is also possible to transform the Gaussian kernel to any other shape by adding several Gaussian kernel. If would wanted to do this we would have increased the calculation time for the SMSM. Hence we opted for a more practical approach to get comparable results with the two and still keep the kernel and parameterization from TREEMIG. Comment: How for SMSM with terrain, does this require a simulation without terrain before? Response: Well as stated before, one can mimic the function used in the FFTM or one can use a Gaussian dispersal function to start with and calculate the parameter for the SMSM from the distribution. However in our case we wanted to be comparable to TREEMIG, so we choose their function and parameterization. And to avoid to use several Gaussian to approximate the function used in TREEMIG we simply tested in an homogenous area. We are now mentioning this in the text on line: . . . Comment: What are the cost reductions then? Response: When the final dispersal kernel is approximated by stacking several Gaussian dispersal kernel the SMSM has to be performed several times.

Comment: l.335-337: I would find it valuable to have the simulation times for the terrain simulation in the table, too! Response: All SMSM calculations are with terrain, though the terrain is a homogenous grid or 1s. In the Matlab script we have differentiated between simulation of SMSM with terrain (one extra multiplication) and without, however since we are not planning to use any SMSM without terrain and since the LPJ-GM code always does a terrain, we decided to remove the SMSM without terrain from the Matlab script. Comment: l.341: This would probably not work with transects?! Response: Yes, the transects have to be chosen in a way that they are not disrupted by barriers that are larger than the dispersal kernel. The main idea behind using the transects is to use them only in heterogeneous areas where you would simulate the whole area. However some parts of the typically squared domain might be homogenous so one might to choose to use transects there as well. Comment: l.358: K is the number of iterations?!

[Figure]

Response: Yes thanks for spotting that we did not explain this. We now added it to the text at line . . . Comment:

l.361: "a very similar migration pattern" I would delete the "very" Response: Done. Comment: l.362: it is slower by 20%! Response: Yes but given the differences in the literature of migration speed within and between measured and simulated migration speed as well as the uncertainty in the parameters of the seed dispersal kernel this is still relatively similar. Comment: l.362-363: in l. 310-312 the authors state that its slower because of the migration path? How do these two different explanations contribute? The stochasticity leads to an increase in migration speed if there are surrounding cells right and left that can contribute via diagonal seed exchange to the cells along the transect. We are currently testing the effect of transects being wider than a single cell, but the results of this would make the ms more complex and we will present them in the next application.

l.364: how to parameterize "explicit considerations of wind directions" Basically one could calculate different Gaussian distributions in different directions and according to the wind distribution in one area using the considerations in Supplement 2. Comment: l.376: Something is missing in this sentence Response: we added an 'or the other' to make it clearer Comment: l.379: Maybe in a DVM? But not in ecology and not to simulate dispersal; the authors should mention some applications - as mentioned above: have a look at Powell (2001) + other references such as Pueyo et al. 2008; Prasad et al. 2013 + I imagine there are much more. Response: Yes we agree, we meant DGVMs, this is certainly misleading and we are now relating to some other applications in the introduction. Comment l.384: "DGVMSs" Response: Thanks for spotting this. Comment: l.410-412: 63-85% instead if 85%? I would maybe remove this quantitative compar- ison. In my understanding the size of the reduction will be dependent on the model and the set-up of the simulations, i.e. on a variety of factors, such as the number of simulated and dispersing species, the resolution, settings of the applied algorithms, etc., and since its two different models and probably very different

simulation set-ups, it seems to me to be comparing apples and oranges? Response: Yes we certainly agree that there are a variety of factors influencing this and therefore it might be more suitable to not quantify it here. Instead we write that our method leads to a reduction in a similar range depending on the configuration of the corridors (Line #CONFIGURATION) Comment: l.413: more pronounced than what or where?

Response: Thanks for spotting this, the sentence that this was referring to was lost in an internal revision. Line … Comment: l.416: maybe 0.5 and 1.0? Response: No here we actually mean 0.1 There are some applications at 1 degree and even 2.5 degree, but when vegetation or even species are in the focus the finer scales are more common. Comment: l.449: From the last Figure in the supplementary it seems that SMSM simulations with terrain are comparably much slower. Is it possible to speed them up with transects? Response: The idea is to have only those parts where the area is very complex or in which we are actually able to define different seed dispersal kernel to be used with the SMSM, all other areas should use the FFTM. The SMSM did also speed up if used with corridors. The values in table 1 are actually calculated with the extra one multiplication required for the SMSM with terrain. We therefore decided to remove the SMSM-without-terrain from the figure in Supp.2. Comment: l.462: I would not call 20% slightly Response: With respect to the uncertainty both in the parameters available for the seed dispersal kernel as well as the estimates of migration speed in the literature from pollen analysis 20% is still a low uncertainty. However we are removing the word slightly. Comment: l.457: "FTTM" -> "FFTM" Response: Thanks for spotting this.

l.465: unfortunate – I think this would be really interesting, especially when simulating fragmented landscape or non-dominant species Response: We absolutely agree and we are already performing test simulations for a further study. Comment: 481: missing ") Response Thanks for spotting this. Comment: l.486: What was the tested set-up? I assume FFTM? I.e. no 'terrain'? Transects with 50km distance? How many competing/migrating species? All grid cells homogeneous? How many years? Response: We

tested using the FFTM (hence without terrain) using 4000 by 4000 grid cells, running for a few years only. Looking at the numbers in table 1 shows that running a full scale simulation with 21000 years over the 3463 $\frac{1}{2}$ degree cells that we typically use for European runs would take a long time: 1800 (CPUh per 100000 cells and 3000 years) /100000(cells in the MS)*3463(half degree cells in Europe) *50*50(rough estimation of how many 1km cells are in a half degee cell)) *21000 years in LGM simulation / 3000 years in testsimulation gives us roughly 10 mill CPUh. Given that my current account allows me 45000 CPUh a month that is currently not feasible and that is why we suggest the transect method. (Actually there might be even more time needed given that there are more than 2 species in the final runs). Comment: l.488: "considerable computation costs" – what were they in the tests (e.g. CPU h per simulated y)? Are continental applications possible, or are they not possible? Response: See above. If the corridors are clever placed yes they are possible and if a more efficient parallelization of the FFT is implemented. In this ms we are not providing a proof for this (we will do in the next where we aim to perform a European simulation). Therefore we do not refer to continental runs anymore. The statement at this point is meant to say that from a memory requirement there is no problem performing the FFTM over large areas. Comment: l.488: plural and singular mix: "a high Response: Thanks for spotting this. ... amounts" + what does "of the FFT as the local simulations" mean? Response: amounts : see calculations above. FFT as local simulation: Currently only one node is taking care of the calculation of the FFT. One could theoretically perform the FFT at each node and use one master node only for collecting the amount of dispersed seeds and performing the communication. Hence there is still some untapped optimization potential. Comment: l.498: what do you mean with "truly mechanistic"? I recommend deleting this statement Response: We meant that the migration rates are a result of the dispersal kernel and establishment in a mechanistic way. We agree that the term might be misleading and have removed it. Comment: l.503: "related estimates the Conclusion section"? Response: Thanks for spotting that there is an 'in' missing.

Comment l.613: Again "10" + something with the formatting Response: Thanks for

spotting this. It is changed. Comment: l.627: Please provide a legend – even if the figure is only schematic Response: We will add a legend. Comment: l.635: When looking at the Figure and reading 2.4 I wondered where the 5*10-7 came from and how this parameter is determined? – Finally I found some information in 3.3 Response: See responses above. It is a fitted parameter to make the two methods result in comparable migration speed. Fig.3 and 4: I would appreciate if the y-axis of the distance plots on the right would have similar scales, this would really help for comparison Response: We will change the axes. Comment: l.647-648: difficult sentence – maybe: "only taken into account for grid cells ..."? Response: Thanks for the suggestion.

Comment: l.661: Comparing the dark blue spots in Fig . 2 and the white ones in Fig. 5 the Figures seem to be mirrored along the diagonal? Comment:

l .664: cpu*h = CPU h? Response: Changed. Comment: l.671: FFTM with 10: shouldn't this be 64% instead of 67%? Response: Thanks for spotting this typo.

References Brokaw, N. V. (1998). Cecropia schreberiana in the Luquillo mountains of Puerto Rico. The Botanical Review, 64(2), 91-120. Fisher, R., McDowell, N., Purves, D., Moorcroft, P., Sitch, S., Cox, P., Huntingford, C., Meir, P., and Woodward, F. I. (2010). Assessing uncertainties in a second-generation dynamic vegetation model caused by ecological scale limitations, New Phytol., 187, 666–681. Hickler, T. , Vohland, K. , Feehan, J. , Miller, P. A., Smith, B. , Costa, L. , Giesecke, T. , Fronzek, S. , Carter, T. R., Cramer, W. , Kühn, I. and Sykes, M. T. (2012). Projecting the future distribution of European potential natural vegetation zones with a generalized, tree speciesâ ÌȨ A ËǦ Rbased dynamic vegetation model. Global Ecology and Biogeography, 21: 50-63. Kruse, S., Gerdes, A., Kath, N. J., and Herzschuh, U. (2018). Implementing spatially explicit seed and pollen dispersal in the individual-based larch simulation model: LAVESI-WIND 1.0, Geosci. Model Dev. Discuss., https://doi.org/10.5194/gmd-2018-31, in review. Neupane, R. C. (2015). Modeling Seed Dispersal and Population Migration Given a Distribution of Seed Handling Times and Variable Dispersal Motility: Case Study for Pinyon and Juniper in Utah. Owens, J. N. (1995). Constraints to seed

production: temperate and tropical forest trees. Tree Physiology, 15(7-8), 477-484. Powell, James (2001). Spatio-temporal models in ecology; an introduction to integro-difference equations. Utah State University, Logan (UT). Prasad, Anantha M., et al. (2013). Exploring tree species colonization potentials using a spatially explicit simulation model: implications for four oaks under climate change. Global Change Biology 19.7, 2196-2208. Pueyo, Y., et al. (2008). Dispersal strategies and spatial organization of vegetation in arid ecosystems. Oikos 117.10, 1522-1532. Snell, R. S. (2014), Simulating long-distance seed dispersal in a dynamic vegetation model, Glob. Ecol. Biogeogr., 23(1), 89–98, doi:10.1111/geb.12106. Snell, R. S., Huth, A., Nabel, J. E. M. S., Bocedi, G., Travis, J. M. J., Gravel, D., Bug- mann, H., Gutiérrez, A. G., Hickler, T., Higgins, S. I., Reineking, B., Scherstjanoi, M., Zurbriggen, N. and Lischke, H. (2014), Using dynamic vegetation models to simulate plant range shifts, Ecography (Cop.)., 37(12), 1184–1197, doi:10.1111/ecog.00580. Interactive comment on Geosci. Model Dev. Discuss., https://doi.org/10.5194/gmd-2018-161, 2018.

---

## Author Response (AR1)

**Authors response to "Simulating migration in dynamic vegetation models efficiently with LPJ-GM 1.0"by Lehsten et al.**

We would like to thank all reviewers for their efforts in reviewing our manuscript. Below is a point by point list of the responses and changes in the manuscript.

Anonymous Referee #1

Overall Comments:
The paper describes two approaches, for simulating seed dispersal in global-scale dynamic vegetation models. Vegetation migration in response to climate change (both past and future) is a major area of research, and the ability to simulate dispersal in DGVMs would be a major advance. There is certainly scientific merit in this manuscript, however there are numerous issues that need to be addressed. In general, since this is a paper about model development, more details about the model and justifications for their choices, need to be included.

Major Comments:
Seed production (Section 2.3.1) For the entire section – "each grid cell", does this refer to the large grid cell or the smaller grid cells within?

*Response:*
*There are no smaller gridcells within gridcells in our model. Whenever we mention gridcell we mean the entire gridcell since we have not subdivided them. We simply use very small (compared to typical LPJ-GUESS uses) gridcells. In each run all gridcells have the same size. However, in some runs we are only simulating a sparse field of gridcells, meaning that while all gridcells are taken into account for the seed dispersal, some of them (often many of them) are not simulated by LPJ-GUESS-GM, hence all seeds landing there will not germinate.*
*We will stress this out in the paper.*

L157 – "no specific age of maturity is taken into account". Maturation age has been shown to be one of the most important factors for determining tree migration rates (e.g., Nathan et al. 2011; Snell 2014). This is especially relevant for trees, as most tree species delayed maturation. Please include a justification for why this was not included.

Nathan R, Horvitz N, He YP, Kuparinen A, Schurr FM, Katul GG. 2011. Spread of North American wind-dispersed trees in future environments. Ecology Letters, 14: 211-219.
Snell RS. 2014. Simulating long distance seed dispersal in a dynamic vegetation model. Global Ecology and Biogeography, 23: 89-98.

*Response:*

*In our implementation we are not aiming to make a model to fit exactly the real world but rather to see whether our way to implementation leads to comparable results compared to TREEMIG. To be able to compare the results we tried to stick to the way it was implemented in TREEMIG, and there the start of seed production was based on the tree height, hence this is what we implemented as well. Height is also a good indicator of maturity. However we agree with the reviewer that age has certainly its merits and might be a more suitable variable in certain conditions. We have in earlier versions used age as a trigger to start seed production. In the next version the user will be able to switch between tree age and tree height as a trigger for seed production. We added this information to the paper in line 570. The mentioned LAI is used to calculate the amounts of seed similar to Lischke et al. (2006).*

Comment:
L159 – please clarify what is meant by "seed bank", and perhaps use another term.
In ecology, seed bank has a very specific meaning (i.e., the dormant seeds in the soil that can germinate in subsequent years).

*Response:*
*This is exactly what we mean. We now use this phrase to define seed bank in the text.*

Comment:
How long do seeds stay in this "seed bank"?

*Response:*
*The annual loss of seed germeability hence the decay time for the seeds is taken from the values in the cited TREEMIG publication (for Fagus it is 0.8). It is a species-specific value though currently the values are similar for all since there is little literature comparing the different species (described under 2.3.3. Seed bank dynamics).*

Comment:
Does each grid cell have their own seed bank?

*Response: Yes. We mention this now in the text.*

So seeds enter after dispersal has already occurred?

*Response: Yes.*

Comment:
Or is this a central seedbank that all grid cells have access to?

*Response: No. Seed bank dynamics (decay, loss due to germination and new arrival due to seed dispersal). Is calculated independently for each gridcell.*
*We have decided to keep the seed bank description short since we took exactly the same approach as in TREEMIG. We have explained it more exhaustively (at line 221).*

Comment:
L160-160 – please provide a justification for why you chose the LAI approach for seed production, and not the carbon allocation approach already implemented in LPJ-GUESS.

*Response:*
*The currently implemented carbon allocation in LPJ-GUESS allocates a fixed amount of carbon to reproductive tissue which is then added to the litter pool. Here the basic unit is carbon rather than seed number, which is what we work with. There are two main reasons why we did not go the way to change the amount of carbon allocated to reproductive tissue depending on seed weight and seed number. Firstly we did not want to change the carbon dynamics within LPJ-GUESS as this is the result of a lot of fine adjustments and any change in the NPP allocation requires a substantial testing afterwards to assure that no unwanted effects occur. Secondly it would have meant that we also need to take the lateral exchange of carbon between cells into account. Currently there are a number of checks that assure that the carbon cycle is closed. Apart from transferring more data between cells, we would have had to adjust these checks as well. So in essence even though we agree that it would be more reasonable to deduct the carbon for the seeds from the NPP and to transfer it to the adjacent cells (though I guess the amount of carbon actually transported outside a cell is very small compared to the one that stays inside), for the purpose of demonstrating the migration mechanism (focus of this paper) we consider it not necessary but in the next version of LPJ-GM in which we plan to simulate historical tree migration we will implement exactly this. We are commenting on this in the paper under 2.3.3 Seed Bank Dynamics.*

Comment.
In addition, please include some more information for how LAI is used to determine the number of seeds?
*Response:*
*We added an example calculation of seed production to the paper (under the section Seed production ).*

Comment:
What value was chosen for maximum fecundity?

*Response:*
*We used the same value as used in TREEMIG, we have added it to the paper (under Seed production).*

Comment:
Is this species specific?

*Response: Yes.*

Comment:
Seed bank dynamics (Section 2.3.3)
L191-193 – this explanation is not sufficient. What is the difference between yearly loss of germinability and the amount of germinated seeds?

*Response:*
*At the start of each year the amount of seeds that survived the last year is calculated.*
*From this number the amount of germinating seeds is subtracted using a species specific fraction.*
*Then new seeds arrive from the same gridcell and surrounding gridcells that are added to the seed bank the cycle begins again.*
*We explain this now in more detail in the paper.*

Comment:
Is there a single seed bank for each large grid cell, or each smaller grid cell inside? L194-198 – this is confusing.

*Response: As there are no smaller grid cells within gridcells, there is one seed bank for each grid cell, of which all have the same size.*

Germination (Section 2.3.4)
Comment:
L202 – 208 – why did you want to add more limitations to establishment? What is the biological justification for this? What does "we fixed this parameter to 0.01 after initial testing" mean? What properties did you evaluate? What does this parameter do? And how does your new limitation interact with the already implemented light limitation (i.e., does this filter happen before or after)?

*Response:*
*This parameter relates to the total area of the gridcell (in which the seeds spread out). In LPJ-GUESS all simulations are done per square meter. If the same number of seeds land in a larger area there will be less per square meter. As we do not change cell size here we fixed this parameter. Basically LPJ-GUESS simulates a certain amount of seedlings to establish each year depending on the amount of light reaching the forest floor. Here, we calculate a probability that the establishment event that LPJ-GUESS simulates happens depending on the amount of seeds available, hence we only decrease LP-GUESS's internal establishment. So the seed limitation filter is applied before the light limitation filter. We made this more clear in the paper at line 235.*

Comments:
Corridors (Section 2.5)
This entire section is also very confusing - looking at the figures helped, but the text needs to be clarified. Are these corridors the large grid cells, or the smaller grid cells inside? Or both?

*Response:*
*As there are no small gridcells and large gridcells, but only one size of gridcells, the corridors are those gridcells in which the full LPJ-GM dynamics is calculated. The cells outside the corridors take part in the seed dispersal routine, but no vegetation dynamics is calculated in them.*
*We made this clearer in the text and mention repeatedly that there is only one size of gridcell however only on those on the corridors the vegetation dynamic is calculated by LPJ-GM. To make this clearer to the reader we added a new figure (Fig. 3).*

Comment:
L260 – 263 – How is the 1 km scale chosen, appropriate for a species with an average long distance dispersal of 200 m? Only a very, very small proportion of seeds would be able to travel 1 km or more.

*Response: Yes it is a small proportion, but it is sufficient to establish the species in the next cell. But we agree, there is a discretization error involved, as in every spatial simulation. We mention this at line 355.*

Comment:
The next section (L285), mentions
"parallel and diagonal corridors". What does this mean? This should be described in this section, with some additional details provided.

*Response:*

*We hope this is now clearer with the additional figure 3.*
*There are basically two types of gridcells (all with the same size). First there are the cells for which LPJ-GM calculates full vegetation dynamics and seed production (type 1). Secondly there are cells (type 2) for which LPJ-GM assumes a seed production similar to the nearest neighbor for which full vegetation dynamics is calculated. Hence there is a complete matrix of seed production for which one of the two described algorithms (FFTM and SMSM) is applied to calculate seed dispersal. Only in those cells for which LPJ-GM calculates the vegetation dynamics these seeds can cause trees to establish and to produce new seeds. These two types of cells are arranged in a way that the type 1 cells form a corridor surrounded by type 2 cells. Since the diagonal corridors are also parallel we agree that we wording is unfortunate. In the revised version we mention that they are north-south, east-west, northeast-southwest and northwest-southeast corridors and explain this more extensively*

Comment:

Results, Explicit seed dispersal (Section 3.1)
It is not clear what results this section is talking about, nor how it relates to the rest of the manuscript. Referring to "pre-studies" is not helpful (i.e., these results are not part of the current manuscript? So why are they included?).

*Response:*
*This pre-study is not part of the manuscript but part of the supplementary material. And it is mentioned in the paragraph. The term pre-study is misleading and we have replaced it and now directly point to the supplement. We mention the Matlab supplement in the Methods where we added a part about the performance of the different algorithms. We also highlight the results in the Results section and discuss this in the Discussion section in the revised version.*

Comment:
There are also no values in here at all.

*Response: Running times for FFTMS and SMSM are compared to a large detail in the table 1. We have pointed to that in the text as well.*

How much faster did the FFTM or SMSM perform compared to the explicit dispersal?

*Response:*
*In the early stadium of the work we did an explicit seed dispersal in LPJ-GM. It became quickly obvious that given the current computation ability this would not allow us to simulate larger areas. We then developed the two methods (FFTM and SMSM). Re-implementation of the explicit dispersal algorithm only to show that it will be much slower would take a considerable time. Therefore we went a different way, by implementing FFTM and SMSM and explicit dispersal mechanism into a Matlab script. This allowed to concentrate on the running time needed for the dispersal mechanism, which is the focus of this paper. A direct comparison of running times required for the seed dispersal for the different algorithms for different area sizes is done in the script and the results are plotted in the pdf. The script also allows the user to simply cut and past the code into Matlab and play around with it.*

Comment:
Also, this is the first mention of a Matlab script (perhaps should be mentioned in the methods?).

*Response:*
*We now have a whole section to do so. It was actually just meant as an add on to aid explaining the methods since reading an implementation helps to be implement it in any other model. But since it is also used to evaluate running times we have covered it in the methods and results section in the revised version (Sections 2.7 and 3.1).*

Comment:
Since (I assume) the Matlab script doesn't include the additional processes
from LPJ-GUESS, how comparable are these results to what you would get in LPJ-
GUESS?

*Response:*
*In table 1 we are comparing the running times of the FFTM and the SMSM with the running time where seed production is calculated but no seed dispersal is performed, while in the Matlab script we are comparing only the seed dispersal calculations of the two new mechanisms and the explicit mechanism leaving out the vegetation dynamics.*
*Hence though one cannot precisely calculate the difference, one can make a rough estimate. Table 1 lists the percentage of the time used for the dispersal for the FFTM method, Supplement 2 shows the increase in computation time for the dispersal algorithm between the FFTM and the explicit seed dispersal. Hence, by multiplying the time needed for the FFTM in table 1 with the factor from the figure in the Matlab script one can estimate the total difference. As an example: The simulation of 100\*100 cells with LPJ-GM uses in total 1800 cpu\*h of which ca 200 cpu\*h are used for the calculation of the dispersal (11%; table 1). According to the graph in supplement 2 the explicit simulation needs one order of magnitude longer than the FFTM, hence a rough guess would be that instead of 200 cpu\*h as used for the FFTM, an explicit seed dispersal would need 2000 cpu\*h, which would increase the total required simulation time for 100\*100 cells to 4000 cpu\*h.*
*One can see in the plot in the supplementary material, that with larger areas, the differences between calculation time of FFTM and supplementary material increase to two orders of magnitude.*
*Hence for larger areas the calculation of the seed dispersal would dominate the required calculation time even more. We added that the performance differences in the Matlab script are only to be seen as rough estimates in the Methods section dedicated to the Matlab script.*

Minor Comments
L34 – not "at least", which implies 1 km or greater. But should be "at maximum" implying
that 1 km is the greatest size that can be used.

*Response: We changed this.*

Comment
L 37 – what is "it"?
*Response: It stands for 'simulating the local dynamics' We changed this.*

Comment
L39 – what "both methods" are you referring to here? The comparison of the Fast
Fourier transformation vs the iteratively shifting seed matrix, or the comparison of be-
tween the simulations with all grid cells, versus the corridors?

*Response: we mean FFTMS and SMSM and will name them in the sentence to make this clear.*

*The corridors are not a method but a way of placing cells.*

L39 – what does "reliable" mean?

*Response:*
*It means that for both methods (FFTMS and SMSM) comparable results are gained by calculating either corridors or the whole area. We will rephrase this.*

L59 – awkward wording.
*Response: We rephrased this.*

L59-79 – both of these paragraphs are missing appropriate references. They have none, but include several statements which need to be referenced.

*Response:*
*The first paragraph expressed mainly the viewpoint of the authors, but the second paragraph clearly states facts and we added an appropriate reference.*

L95 – although this is explained in more detail in the discussion, it would be helpful to have this information in the introduction. (i.e., what did previous approaches do, and why were they limiting).

*Response: We had in fact moved the review of other methods from the introduction to the discussion but will provide a small summary part of the Snell et al.(2014) approach in the introduction section in the revised version.*

L108 – 110 – a few more details about LPJ-GUESS? This one sentence is vague and particularly unhelpful for understanding what this model does.

*Response:*
*LPJ-GUESS has been described in 200+ publications (though most of them focus on a small side aspect or a new development that did not become part of the standard version). Finding the right amount of information given to the reader is tricky. We have extended the model description.*

L127-128 – if all vegetation is killed, and now seed dispersal is active, you MUST have some vegetation or you won't have any seeds?? Where does the first generation come from?

*Response: As described in 225-230, for some cells seed limitation does not apply until a certain point in time. These cells are the refugia of the species which have free establishment. In the example simulations these are the cells in the upper left corner. Hence after the clearing of the vegetation trees can establish freely here, produce seeds that can subsequently disperse to the surrounding cells.*

L142-150 – please clarify this how this occurred. So instead of one grid cell with multiple patches, you simulated one grid cells with multiple grid cells? But these smaller grid cells, had spatial locations and could interact with each other (unlike patches)? This was my interpretation, but this needs to be clearer. A conceptual figure would help.

*Response:*
*We simulated only one patch per grid cell. However in typical LPJ-GUESS simulations, grid cells are as big as the climate data dictate, e.g. 0.5 up to 2.5 degree longitude/latitude.*
*In our simulation, grid cells are small compared to the standard LPJ-GUESS size. Hence for the area simulated in one standard LPJ-GUESS simulation gridcell (with several patches), LPJ-GM would place many gridcells (with one patch). So there is not one gridcell with multiple gridcells, but just many small grid cells. We thank the reviewer for the suggestion to make a conceptual figure and included it in the revised version (fig.3).*

L188-189 – need more details about how these parameters were "roughly estimated" if this approach is to be applied in other models or for different species.

*Response:*
*The term 'roughly estimated' indicated that there is a high uncertainty connected to these values. We removed the roughly estimated as we want to indicate that they were not estimated by us, but by Lischke et al. 2006.*

L286-289 – this sentence is confusing.

*Response: we broke up the sentence and made it clearer.*

Figure 5 is not clear – what is causing the white areas? Neither the results nor the methods addresses what the simulation set up was that could cause this pattern. The explanation "no seeds were able to reach them" is not true, as seeds obviously reached all the way around the white circles in the center (i.e., beech arrived in year 2500, but then never migrated in?).

*Response:*
*The main purpose of the simulation displayed in Figure 5 was to demonstrate a how the method performs when a certain seed dispersability is defined caused by a certain terrain.*
*To demonstrate the effect we created areas in which the permeability for the seeds were set to zero, hence they could not enter the grid cell. This is shown in the methods section in Figure 2 which shows the probability of entering a new grid cell. In the blue areas in figure 2 the cells have a zero permeability hence seeds cannot enter the cell. We have now updated the colorscheme of figure 2 to highlight the areas with zero permeability and added a sentence to the methods section explaining this.*

Comment:

Numerous small grammatical errors throughout (this is not a complete list, just a few examples). L25

L51-52, "to have a sufficient amount of seeds"

L72 – unnecessary "of"
L86 – unnecessary "the"
L202 – "depending stochastically depending"
L305 – "Using at a distance of"
L347 – "..we are the first that manage to implement. .

*Response:*
*We have changed these grammatical errors and check the text again for further grammatical issues.*

Anonymous Referee #2

Comment:
The paper presents two methods for simulating tree species migration, newly imple-
mented in the dynamic global vegetation model LPJ-Guess. I find the paper mostly
well written and generally an interesting scientific contribution.
*Response: Thank you for this summary.*

Comment:
What I, based on the presented material, cannot consent to is the reoccurring state-
ment that the model can be used for continental simulations of multiple interacting
species, nor that it is suitable for DGVMs beyond special cases (i.e. species simulations
in Europe).

*Response:*
*We agree that the performance of the dispersal algorithm as presented by now creates a doubt that*
*a continental simulation is possible. Given that we clearly make this claim should require for us to*
*present an algorithm that would have a suitable performance. The main aim of this paper was to*
*introduce the two algorithms into the DGVM. We aim to perform continental scale simulations in*
*an upcoming work. Hence we did not invest a lot of time into optimization, except for a paragraph*
*in the discussion, for simplicity of the paper. We will rephrase all statements where we state that our*
*implementation can be used for continental applications and write that it has the potential to be*
*used for large areas and also has a lot of potential for performance improvements. Again the*
*intention was to present the two algorithms for seed dispersal, while any continental scale*
*simulation experiments would require to also present a completely new parameterization of some of*
*the trees, as well as many other aspects hence we would like to refrain from performing continental*
*scale simulations for now.*
*We completely agree with the second statement. Given that we worked a lot in the past with species*
*simulations we completely forgot that the currently most common application of DGVMs use PFTs*
*and there it is not necessary to include seed dispersal.*

Comment:
Most DGVMs use plant functional types with mixed dispersal/reproduction traits, particularly when
used for large spatial applications. The example application deals with only two species, and only
the dominant late successional tree species Fagus Sylvatica is tracked, which has a quite narrow
dispersal kernel. Furthermore, the application deals with a homogenous landscape. From what the
authors show and write I am not convinced that/how a continental simulation with multiple
interacting and dispersing species would/can be possible.

*Response:*

*Again we completely agree that only for simulations on the species level, an inclusion of seed dispersal is useful. Currently all dispersal simulation is performed at the master node, if the dispersal simulations for each species are performed at one node per species there should be (theoretically) no reasonable reduction in performance. We also repeat that we have not proven that we can simulate at a continental scale, but only that we are 2-3 orders of magnitude faster in simulating seed dispersal compared to the explicit simulation. Whether this is sufficient for a continental simulation will be shown in successive work .We will rephrased this claim.*

Comment:
From the paper I understood that using FFTM with widely spread transects would not be appropriate in heterogonous areas. SMSM with terrain, on the other hand, would not save enough computation time to be applicable on continental area. Is the plan to use FFTM with transects in homogeneous areas and SMSM in heterogeneous areas? But if so, how would these algorithms then communicate with each other in a continental simulation?

*Response:*
*Though not formulated in the paper (for simplicity), yes this is the main idea. Depending on the parameters of the species specific dispersal kernel, there is a maximum distance that the seeds are transported (theoretically there is no such limit, but given the strong decrease of the tail, this assumption has no influence on the final result). One option is to define a certain area as heterogeneous. The seeds produced in this area are dispersed by the SMSM algorithm, while the seeds of the remaining area are dispersed by the FFTM.  Though for both the seed production is only taken into account for the assigned areas the seed fall will be calculated for the area plus an edge surrounding the area with the width of half a maximum kernel width. In a last step the dispersed seeds of the two methods are added to a final distribution of seeds.  This way there is no complicated communication of the two algorithms, but the edges of the areas that are simulated overlap while the areas of the seed production that go into the algorithm do not.*
*We have presented this reasoning in the Discussion as a last point in 4.2 Comparison of the two dispersal methods.*

Comment:
Given that the paper, the presented ideas and the LPJ-GM implementation are already a substantial contribution, I recommend reconsidering the (over?) statements regarding continental applications and DGVMs e.g. in the last sentence of the abstract and particularly the first sentence in the discussion and talk about DVMs with species and spatial extents exceeding applications of a few ha,  which is a good and sound contribution.
*Response:*
*We have tuned down our statements and clearly show what level of areal coverage the algorithms are can simulate.*

Comment:
Another way could be closing the explanatory gaps, i.e.  (1) discussing issues with DGVMs and how DGVMs, which usually use plant functional types (PFTs) for large scale applications, could be parameterized for the algorithms, (2) discuss the costs/difficulties of an application with a realistic number of interacting species, with differing dispersal traits and (3) explaining how a realistic continental simulation could be assembled with the FFTM and/or the SMSM simulation, given spatial fragmentation and spatial heterogeneity.

*Response:*
*We have done this in the Discussion.*

General comments

1. In many places in the text the authors state that transect simulations lead to similar/only slightly reduced migration speed. However, from the figures/table it seems to be quite a significant underestimation, and the less transects the worse the underestimation of the migration speed (>20%; i.e. in the 3000y application >600y delay). I recommend stating this clearly and to discuss the consequences.

*Response: An underestimation of 20% might look large, but given that most current approaches completely ignore seed dispersal and also that the parameterization of the seed dispersal kernel comes with quite an uncertainty puts the 20% in perspective. The currently reached migration speeds are very likely to be too low as described in the Discussion, which is probably caused by uncertainties in the parameterization of the seed dispersal kernel. However as the aim was to implement the two methods in a DGVM we aimed to keep the kernel similar to the parameterization within TREEMIG to be able to compare results. We did remove the word 'slightly' when mentioning the migration speed reduction.*

Comment:
2. The authors claim that they fulfil the stochastic requirements because they have 200 or so 1km2 grid-cells when comparing to the usual 0.5 x0.5 grid cell. However, this only holds if the spatial heterogeneity caused by the stochastic disturbances and stochastic mortality does not affect tree species migration. In the example application the authors choose Fagus Sylvatica, a dominant late successional tree species, and I can imagine that for this species the stochasticity might indeed play a minor role.
However, what in case of e.g. pioneer, less dominant/more specialised species? These might depend on disturbed areas for establishment – is the transect approach valid for such species? I would find it very helpful to see how the stochasticity and the few available transect cells might affect the spread of such species.

*Response:*
*Pioneer species are typically fast migratory species, hence they would typically be able to colonize the area before the late successional species arrives (if both occupy the same climate space). If on the other hand the late successional species has a larger climate range you are correct that the early successional species will be hindered in its migration into the few spots that are available over a short time. This is true for both a full simulation as well as a simulation along transects. The fact that the seed survival of early successional species is typically higher due to the lower seed mass, should allow the spread still in the small temporal successional gaps. Again the focus here is to introduce the methods. A parameterization of the species that results in a migration speed comparable to observed values is outside the range of this study. However we agree that this is an important point and will discuss it at line #EARLY SUCESSIONAL SPECIES.*

Comment:
It is correct that if applied globally DGVMs usually use 0.5 grid-cells, however when applied as DVMs on continents or regionally the resolution is usually much smaller. See e.g. the dispersal experiments by Snell (2014), and the simulation of European potential natural vegetation with LPJ-Guess (Hickler et al. 2012). 4. I would recommend referring a bit more to relevant literature in some parts of the text, since several of the ideas/methods have already been discussed/used elsewhere. (I mentioned some references in the specific comments list below).
*Response:*
*The maximum resolution is dictated by the climate data available. Given that LPJ-GUESS is parameterized using CRU climate data the simulations at coarse scale are typically performed at 0.5 degree or sometimes using the CRU climatology to bias correct the CRU time series it is*

*performed at 0.25 or 0.1 degree. Even when run with 10 patches and at 0.1 degree would result in 250 patches per 0.5 degree which is comparable to the 200 simulations that we perform at any 50 by 50 km cell. However we now mention that some regional simulations use finer grid cells.*

Comment:
What I miss in the current introduction is a bit more on why migration is missing in DGVMs. The authors state that one reason is the '1D' property, i.e. that cells are not interacting and thus the computation costs of making them interacting. But what should also be mentioned is the problem of parameterisation: DGVMs usually use PFTs, often compiled of species with various different traits with respect to migration (dispersal vectors, competitiveness, generation times, ...) (e.g. Snell et al., 2014).
*Response: We now mention this at line 103.*

Comment :
  Furthermore, if I understood it correctly, the example simulation is for 3000y and the tracked species migrates 100km in that time. Several of the criticized studies with DGVMs (e.g. "land use change on vegetation and ecosystem properties") would use well below 3000y; mostly around 100/200y – given the comparable cheapness of 1D simulations and the mentioned constraints due to parameterisation: wouldn't a 'no dispersal' simulation be sufficient for many simulations with large spatial extent and coarse resolution?

*Response:*
*As already mentioned before (and highlighted in the Discussion) the migration speed that we are calculating are way too small compared to measured values. The aim of this paper was to introduce the method and here we choose to measure our success in response to the migration speed simulated by TREEMIG since we used a similar dispersal kernel. Any real world application will require a new parameterization of the kernel to gain a realistic speed. Currently most simulations are not 'no dispersal' simulations but 'extremely fast dispersal' simulations (given that they use free establishment). Otherwise they could not show any response of vegetation on climate. For short term studies where the time horizon is well below the generation time of the species there is a limited use of a dispersal kernel. Especially for Europe where land use is dominating the vast majority of the landscape and at least in the northern part also plants (often alien) species in forests, any kind of simulations assuming semi-natural conditions are questionable. However to understand current tree distribution in those parts which are still semi-natural and especially to understand forest species history taking seed dispersal into account might be important. We now mention this reasoning as a last point in 4.4.3.*

Comment:
  I would appreciate a more detailed description of the SMSM method. Maybe an illustration? Would this method work with a species with a more pronounced long distance dispersal tail than Fagus Sylvatica? What would this mean regarding computation costs? How to parameterise the SMSM? Could a setting like Fig 5 be simulated with transect at all? Looking at the supplementary figure it seems that the matrix shift method with a terrain has a very small computational gain?

*Response:*
*Thank you for suggesting to add a figure that will illustrate the SMSM, we have now added an example sheet which demonstrates the SMSM additionally to the Matlab code which also gives the full details.*
*With respect to the long tailed distribution: within each SMSM step the maximum dispersal distance increases by one cell. Hence in our example we have a maximum dispersal distance of 10 cells (10 km). A simulation in which you want to allow 20 cells maximum distance require twice as much*

*computation time for the SMSM. For long tailed species the FFTM is certainly better suited, given that this method has no such limitations.*

*The parameterization of the SMSM is described in detail in the Supplementary material. While in general any kind of dispersal kernel can be used in the SMSM, this requires to stack several Gaussian kernel on top of each other and hence would of course increase computational demand. If computation time is an issue, the FFTM is a better choice.*

*Compared to the explicit simulations of seed dispersal, the SMSM is still 3 to 4 times faster. This is much slower than the FFTM which is 1.5 order of magnitudes faster but the improvement is still significant.*

Comment:

I would appreciate more discussion of the limitations and a clearer directive how to apply the algorithms for a continental simulation, if possible. When reading the text I got the feeling that the remedy for the FFTM limitations (heterogeneity/fragmentation, wind directions) is to use the SMSM, but that this method, particularly if used with terrain, is not performant enough for continental applications. Some more buzzwords for the limitation section: parameterisation of SMSM; species parameterisation; fragmentation when using the FFTM; what about ecosystems with many species (i.e.. tropics).

*Response:*
*You read correctly that FFTM is not able to handle landscapes resulting in heterogeneous seed dispersal while SMSM has strong performance constrains. As a matter of fact, the FFTM is still applicable if the barrier is larger than the kernel width since it will place seeds there but if the cells are not suitable then the seeds will not germinate (Baltic sea, alps). The SMSM is only required in areas where we have (or rather know) different dispersal tail lengths depending on the terrain. This might be the case in some valleys in the alps, where seed dispersal acts mainly along the valleys, but not the mountain, but given the typical resolution of the output for continental studies this might not be necessary to apply the SMSM at all, while for finer scale studies SMSM might be the best choice. We are discussing this in the Discussion section together with the mentioning that for some simulations the best option might be to combine the FFTM and the SMSM at the end of 4.2.*

Comment:
Reduction of migration speed by >20%, i.e. in the 3000y simulation > 600y delay. 8. The editor provided me with the model code. Unfortunately I was not able to understand how the simulations were done. There are no hints on how the simulations were conducted, nor was I able to identify the configuration file (instruction script (ins)?) used for the simulations or to find out how/where the transects were defined. I know that it is cumbersome but in the spirit of "good scientific practice" it might be nice to provide and mark the configurations files?

*Response:*
*The transects are defined in the gridlist. Basically while a typical LPJ-GUESS gridlist contains only columns one for the longitude and one for the latitude, in LPJ-GM there are additional columns in which for each species a time is given in which free establishment is allowed see below for the start of a gridlist.*
*23 50   TeBS,101.0,IBS,101.0*
*23.01 50  TeBS,100.0,IBS,10000.0*
*23.02 50  TeBS,100.0,IBS,10000.0*
*The first line indicates that at position 23 degree longitude, 50 degree latitude both species TeBS (temperate broadleaved Summer green tree or beech and IBS Intermediate shade tolerant broadleaved Summer green tree; or birch) are allowed free establishment (hence no seed limitation) at the year 101, which is one year after the initialization phase for nitrogen initialization. Hence this cell would form a refugia for beech. The next cell is located at 23.01 degree longitude and 50*

*degree latitude and beech is only allowed free establishment after the year 10000, hence not within the simulated time of 3000 years, it can only establish at this site if seeds arrive there. Birch is allowed to establish at this site without seed limitation.*

*The transects are defined in a way that only the cells that are on the transects are listed in the gridlist.txt. This way of defining them might not be the most elegant one, but since the current setup of LPJ-GUESS simply cuts the gridlist into as many pieces and distributes them into different directories in which the simulation is performed, this way I did not had to read in a separate file for the refugia definition, and I am sure that the information is linked to the gridcell.*

*The configuration script (the ins file) is similar to the one used in Hickler et al. except that it contains these additional entries at the global level.*

*! migration INSTRUCTION*

*years_total  3000  ! How many years the dispersal simulation is performed*

*domain 23         50         0.01         0.01 ! which domain is simulated and with what resolution*

*param "size_lat" (num  100)  ! how many cells are in the domain along the latitude*

*param "size_lon" (num  100) ! how many cells are in the domain along the longitude*

*dispersal_patchsize 0.99       ! How big a single patch is.*

*if_dispersal_fft  1                 ! whether  FFTM dispersal is performed*

*if_dispersal_float  0               ! whether SMSM dispersal is performed*

*if_dispersal_ext_fft  0        ! whether another variant of FFTM (not described in the paper) is performed*

*stochastic_seed_est_scaler  0.01  ! scaler for the patch size*

*output_interval        10 ! in years save space since not all years are needed in the output*

*Each species contains the following extra parameters which are taken from TREEMIG(here are the values for beech):*

*max_fecundity 29.                 ! maximum fecundity*

*min_height_for_maturity 14.4  ! minimum height for maturity*

*germination_rate 0.3  ! rate of seeds germinating per year*

*max_seed_age 3.3 !         maximal survival times for seeds in seed bank*

*short_range_disp_frac 0.99 !  fraction of seeds that go into short seed dispersal*

*short_disp_alpha 25 !  parameter for short distance dispersal*

*long_disp_alpha  200 ! parameter for long distance dispersal*

*I also would prefer to make the whole model code publicly available. However current policies within the modelling consortium only allows to give access to model code after individual contact with the author. I decided that my unit containing the implementation of the code for the actual migration will be made publicly available (as a supplement to this paper), but there are of course some other small bits and technical issues, like for example the MPI related code that is located in other units.*

Specific comments:

l.1: Maybe consider to adapt the title, since LPJ-GM does not necessarily lead to a more efficient simulation of migration in dynamic vegetation models per se – e.g.: "Simulating migration in dynamic vegetation models efficiently on the example of LPJ- GM" or maybe better "Simulating migration efficiently in the dynamic vegetation model LPJ-GM"

*Response:*
*Yes  we have changed the title to "LPJ-GM 1.0: Simulating migration efficiently in a dynamic vegetation model"*

Comment:

l.21: Most DGVMs do not use species but plant functional types (PFTs)
*Response:*
*We mention this in the second paragraph of the introduction.*

Comment:
l.31: From the last Figure in the supplementary, SMSM with terrain seems to be much
slower than FFTM?
*Response: Yes it is especially now that we have optimized the Matlab code (though at the expense of readability) it is. We deleted the word 'marginally' in the abstract. However it is still faster than an explicit seed exchange.*

Comment:
l.40: "Furthermore, with the transect method both methods"?
*Response: We have reformulated the 'with the transect methods'. We have replaced continents with large regions, since we have not really shown that continents can be simulated with our method. We have made the sentence clearer now.*

Comment:
l.49: DGVMs assume that some instance (i.e. species) of the PFT can establish

*Response: We added a paragraph stating that:*
*"Additionally to the reasons mentioned before, most DGVM applications use plant functional types which comprise typically species with very different traits with respect to migration (e.g. dispersal vectors or seed properties), hence introducing migration would require to split up PFTs into smaller groups and to parameterise the additional properties. "*

Comment:
l.51: Something seems not correct with the embedded sentence – maybe that instead
of the?
*Response:*
*No here that would give a different meaning. We inserted an 'a' and hope the sentence is now easier to read. Line:56*

Comment:
l.53 & 63: Anyway DGVMs usually do not simulate species but only PFTs
*Response:*
*Since we wrote already before that we are only considering species simulating DGVMs we consider this covered.*

Comment:
l.60: When considering ecosystem properties in the future hardly any study would make projections »100y, maybe 200y, but the example in this study uses 3000y. Wouldn't – based on what is shown in this paper – a "no migration between large grid-cells ($0.5\circ$) assumption" be appropriate for studies with ~ 100-200y?
Response: We mention this now with the following sentences. "For periods of less than 50-100 years ahead, which corresponds to at most a few generations of most tree species, the explicit modelling of seed dispersal might be less important for simulating tree distributions, in particular when taking into account the overwhelming influence of human activities. "

Comment:

l.76: Another example of a model even accounting for wind speed and direction: LAVESI-WIND (Kruse et al. 2018)
*Response:*
*Thanks for pointing us to the paper, we are citing it now in line 86.*

Comment:
l.82: What does the spatial heterogeneity refers to in this context – soil and climate? If I understood the set-up right LPJ-GM also does not account for such heterogeneities within the grid cell, only to such with regards to species interactions and stochasticity?
*Response:*
*Yes: Soil and climate, mountains blocking seed transport as well. This sentence should simply highlight that a simple transfer of migration speeds calculated with models at fine scale into models at coarse scale is challenging.*

Comment:
l.94: why every time-step? LPJ-GM only does it once per year?
*Response:*
*Yes of course we mean annually. We change this at line 103.*

Comment:
l.100: If I understood it correctly the presented simulations only simulate two species.
*Response:*
*Yes in our example simulation only two species are simulated but the method can simulate more species in a real application case.*

Comment:
l.101: What would a simulation with several species look like, does each need one FFT/SMSM? What are the resulting costs?
*Response:*
*In the current simulation time that we present in the table we are actually simulating the seed dispersal of both species independently (though it would of course be faster to only simulate one species). Yes each species needs its own FFTM or SMSM to be performed if the migration of several species is to be evaluated, however they could be potentially performed at separate nodes which would decrease calculation time again. We are mentioning this in the Potential furhter improvements section.*

Comment:
l.110-111: Please list a few key references describing LPJ-GUESS 4.0
*Response:*
*We included Smith 2014 and Lindeskog 2013 which are the main references describing the 4.0 version.*

Comment:
l.119: Above and below this node is called master
*Response:*
*We now call it master here as well.*

Comment:
.123: "no seed dispersal"-> "no seed limitation"?
*Response: Yes we added this as well. Line 146*

Comment
l.130: There are species producing seeds throughout the year (see e.g. Owens 1994, Brokaw, 1998)
*Response: Yes this is one of the discretization errors that we have to make. Given unlimited computing power and knowledge of weather conditions and plants behavior, we would perform the FFTM or SMSM daily over the time when seeds are produced. However, as a first improvement of the situation in which most models do not consider seed dispersal at all, we suggest to simulate at an annual time step.*

Comment:
l.140: Here or generally in LPJ-Guess?
Response:
*It is variable but this is the recommended size.*

Comment:
l.153-157: How is this similar to Lischke et al., 2006? Lischke et al. (2006) do not mention LAI but state: "The number of seeds S produced per year by each tree depends on its height, species and mast seeding period."?
*Response:*
*We also use the height of maturity, but no mast seeding period. While Lischke et al. scaled the seeds with height we did scale them with LAI, you are correct this is not the same and we have taken that sentence away as it was meant introduce into the chapter but it is not necessary.*

Comment:
l.175: For Fagus Sylvatica?!
Response:
*Added. Actually we also simulated seed dispersal for Birch but since Birch is set to no seed limitation.*

Comment:
l.181-182: But wouldn't the implementation of wind direction lead to anisotropy and therefore make the FFTM not applicable anymore (E.g. Neupane (2015))?
*Response:*
*The FFTM can apply any shape of seed dispersal kernel, it can just not change it with the landscape. Hence certain wind directions are possible like the kernel used for illustration, which is also skewed, for example by wind. Neupane simulates effects of the landscape on fruit dispersing birds. Such an effect would have to be modelled by the SMSM. If the different wind directions in different parts of the domain (e.g. caused by a certain terrain) is to be taken into account, this also needs to be done by SMSM.*

l. 185: maybe use θ and add the θ = 1 in the text below?
*We changed this equation to make it clearer.*

Comment:
l.186: long term -> long distance?
*Response: Changed.*

Comment:
l.197: but how is the number of seeds defined in this case, since in the next para it is stated that the establishment of seedlings depends on the number of available seeds?
Response:

LPJ-GUESS calculates the number of established individuals per species depending on the light reaching the forest floor. LPJ-GM takes this value and sets it to zero depending on the presence (or rather absence) of seeds. In case of establishment free from seed limitation (in our case the birch), this step is not performed. Hence the species can always establish depending only on the light reaching the forest floor.

Comment:
l.202: "depending stochastically depending"
*Response:*
*Thanks for spotting this repetition. It is fixed.*
Comment:
l.206: "seed bank per and the germination" remove the per?
*Response:*
*Done*

Comment:
l.216: The authors should definitely mention that the method has also already been broadly applied in simulating dispersal. E.g. have a look at Powel (2001) + shortly googling I e.g. found Pueyo et al. (2008) and Prasad et al. (2013) and I assume there are more.
*Response:*
*We are now mentioning that there are a number of applications which already use ffts to simulate dispersal and cite a few of them in the first paragraph of 2.4.*

Comment
l.235: "different wind distributions" -> only if they are valid for the whole simulated area, or?
*Response:*
*We added this remark in line 270.*
l.242: How is this proportion determined?
*Response:*
*A few lines later we point to a derivation of the parameters (in this case this proportion) in the supplementary material S.1.*

Comment:
l.242: 1km2 cell?
*Response:*
*Yes in our application all cells have one km^2 extent.*

Comment:
l.249: How often is this done/ needs to be done to account for long distance dispersal?
*Response:*
*Currently this is done 10 times hence we are reaching a maximum of 10 km.*

Comment:
What happens with the seeds at the boundary of a simulation area?
*Response:*
*For both the FFT as well as the SMSM simulation we extend the area by one kernel width to avoid / minimize edge effects. Basically all seeds that land of the seed domain are lost.*

Comment:
l.255: Figure 2 is not cited in the main text (only in Fig. 5).
*Response:*

*Thanks for spotting this. This sentence must have gotten lost in one of the internal revisions. We now refer to Figure 2 in the description of the simulations in line 320.*

Comment:
l. 266: Wouldn't the heterogeneous landscape be much more crucial to test the applicability of the methods?
*Response:*
*No the idea is to only use the corridors in homogenous landscapes and to speed up the simulation there. In heterogeneous landscapes this simplification is not suitable. Hence we only test the corridors in homogenous landscapes.*

Comment:
l.274-275: And? But?
*Response:*
*And we do not want to strongly increase the migration speed. We have spent a lot of time trying to come up with a better solution like some kind of distance weighted average, however so far we have not found one. Hence we prefer to have a reduction by 20% hence a conservative estimate rather than a strong increase which also was heterogeneous within the simulated area depending on the arrangement of the corridors. We have not given up the hope to come up with a better solution in a real world application.*

Comment:
l.276: Out of interest: how many CPUs were used/ what computing environment?
*Response:*
*We used 200 nodes (with 20 nodes per CPU) at the LUNARC computing facilities.*

Comment:
Would it be possible to add a 'no dispersal and no communication' (i.e. a 1D) simulation for comparison?
*Response:*
*Yes have been working on it and planned to add it to the final version of the paper. As it turned out the differences between the version with communication and without are negligible. We also mention this now in the ms.*

Comment:
l.303: 1km2 grid cells
*Response:*
*We added this.*

Comment:
l.306: somewhat? >20%!

*Response:*
*We have removed the somewhat.*

Comment:
l.309: Maybe add the numbers for the variability
*Response:*
*What we mean here is visual realization that the distance of the points increases. Since we do not use a mean value to estimate the migration front, it is hard to quantify this variability since we have a different variability above compared to below the line.*

Comment:
l.312: Which probably also explains the patterns in the migration front?!
*Response:*
*Yes that is the reason.*

Comment:
l.314: When FFT when FFTM?
*Response*
*It should always be FFTM. Thanks for spotting this.*

Comment:
l.318-319: How do the simulations compare to a simulation without communication between grid cells, i.e. 1D simulations?
*Response:*
*See above the differences to the version with communication are negligible.*

Comment:
l.323: How to specify this parameter when not having a FFTM simulation at hand?
*Response:*
*Here the aim was to parametrize the SMSM in a way that we have a similar migration speed compared to FFTM. In a practical application one would have a certain dispersal kernel and the derivation in Supplement 2 would allow to estimate the parameter to fit a Gaussian kernel. It is also possible to transform the Gaussian kernel to any other shape by adding several Gaussian kernel. If would wanted to do this we would have increased the calculation time for the SMSM. Hence we opted for a more practical approach to get comparable results with the two and still keep the kernel and parameterization from TREEMIG.*

Comment:
How for SMSM with terrain, does this require a simulation without terrain before?
*Response:*
*Well as stated before, one can mimic the function used in the FFTM or one can use a Gaussian dispersal function to start with and calculate the parameter for the SMSM from the distribution. However in our case we wanted to be comparable to TREEMIG, so we choose their function and parameterization. And to avoid to use several Gaussian to approximate the function used in TREEMIG we simply tested in an homogenous area. We are now mentioning this in the text on in chapter 3.3.*

Comment:
What are the cost reductions then?
*Response:*
*When the final dispersal kernel is approximated by stacking several Gaussian dispersal kernel the SMSM has to be performed several times.*

Comment:
l.335-337: I would find it valuable to have the simulation times for the terrain simulation in the table, too!
*Response:*
*All SMSM calculations are with terrain, though the terrain is a homogenous grid or 1s.*
*In the Matlab script we have differentiated between simulation of SMSM with terrain (one extra multiplication) and without, however since we are not planning to use any SMSM without terrain and since the LPJ-GM code always does a terrain, we decided to remove the SMSM without terrain from the Matlab script.*

Comment:

l.341: This would probably not work with transects?!

*Response:*

*Yes, the transects have to be chosen in a way that they are not disrupted by barriers that are larger than the dispersal kernel. The main idea behind using the transects is to use them only in heterogeneous areas where you would simulate the whole area. However some parts of the typically squared domain might be homogenous so one might choose to use transects there as well.*

Comment:

l.358: K is the number of iterations?!

Response:

*Yes thanks for spotting that we did not explain this. We now added it to the text at line …*

Comment:

l.361: "a very similar migration pattern" I would delete the "very"

*Response: Done.*

Comment:

l.362: it is slower by 20%!

*Response:*

*Yes but given the differences in the literature of migration speed within and between measured and simulated migration speed as well as the uncertainty in the parameters of the seed dispersal kernel this is still relatively similar.*

Comment:

l.362-363: in l. 310-312 the authors state that its slower because of the migration path? How do these two different explanations contribute?

*Response:*

*The stochasticity leads to an increase in migration speed if there are surrounding cells right and left that can contribute via diagonal seed exchange to the cells along the transect. We are currently testing the effect of transects being wider than a single cell, but the results of this would make the ms more complex and we will present them in the next application.*

Comment:

l.364: how to parameterize "explicit considerations of wind directions"

*Response:*

*Basically one could calculate different Gaussian distributions in different directions and according to the wind distribution in one area using the considerations in Supplement 2.*

Comment:

l.376: Something is missing in this sentence

*Response:*

*We added an 'or the other' to make it clearer*

Comment:

l.379: Maybe in a DVM? But not in ecology and not to simulate dispersal; the authors should mention some applications - as mentioned above: have a look at Powell (2001) + other references such as Pueyo et al. 2008; Prasad et al. 2013 + I imagine there are much more.

*Response:*

*Yes we agree, we meant DGVMs, this is certainly misleading  and we are now relating to some other applications in the introduction.*

Comment
l.384: "DGVMSs"
*Response: Thanks for spotting this.*

Comment:
l.410-412: 63-85% instead if 85%? I would maybe remove this quantitative comparison. In my understanding the size of the reduction will be dependent on the model and the set-up of the simulations, i.e. on a variety of factors, such as the number of simulated and dispersing species, the resolution, settings of the applied algorithms, etc., and since its two different models and probably very different simulation set-ups, it seems to me to be comparing apples and oranges?
*Response:*
*Yes we certainly agree that there are a variety of factors influencing this and therefore it might be more suitable to not quantify it here. Instead we write that our method leads to a reduction in a similar range depending on the configuration of the corridors (Line 484).*
Comment:
l.413: more pronounced than what or where?

*Response:*
*Thanks for spotting this, the sentence that this was referring to was lost in an internal revision. We are now writing a complete statement at line 486.*

Comment:
l.416: maybe 0.5 and 1.0?
*Response:*
*No here we actually mean 0.1 There are some applications at 1 degree and even 2.5 degree, but when vegetation or even species are in the focus the finer scales are more common.*

Comment:
l.449: From the last Figure in the supplementary it seems that SMSM simulations with terrain are comparably much slower. Is it possible to speed them up with transects?
*Response:*
*The idea is to have only those parts where the area is very complex or in which we are actually able to define different seed dispersal kernel to be used with the SMSM, all other areas should use the FFTM. The SMSM did also speed up if used with corridors. The values in table 1 are actually calculated with the extra one multiplication required for the SMSM with terrain. We therefore decided to remove the SMSM-without-terrain from the figure in Supp.2.*

Comment:
l.462: I would not call 20% slightly
*Response:*
*With respect to the uncertainty both in the parameters available for the seed dispersal kernel as well as the estimates of migration speed in the literature from pollen analysis 20% is still a low uncertainty. However we are removing the word slightly.*

Comment:
l.457: "FTTM" -> "FFTM"
*Response:*
*Thanks for spotting this.*

l.465: unfortunate – I think this would be really interesting, especially when simulating fragmented landscape or non-dominant species
*Response:*
*We absolutely agree and we are already performing test simulations for a further study.*

Comment:
481: missing ")
*Response*
*Thanks for spotting this.*

Comment:
l.486: What was the tested set-up? I assume FFTM? I.e. no 'terrain'? Transects with
50km distance? How many competing/migrating species? All grid cells homogeneous?
How many years?
*Response:*
*We tested using the FFTM (hence without terrain) using 4000 by 4000 grid cells, running for a few years only.*
*Looking at the numbers in table 1 shows that running a full scale simulation with 21000 years over the 3463 ½ degree cells that we typically use for European runs would take a long time:*
*1800 (CPUh per 100000 cells and 3000 years) /100000(cells in the MS)\*3463(half degree cells in Europe) \*50\*50(rough estimation of how many 1km cells are in a half degee cell)) \*21000 years in LGM simulation / 3000 years in testsimulation gives us roughly 10 mill CPUh. Given that my current account allows me 45000 CPUh a month that is currently not feasible and that is why we suggest the transect method. (Actually there might be even more time needed given that there are more than 2 species in the final runs).*

Comment:
l.488: "considerable computation costs" – what were they in the tests (e.g. CPU h per
simulated y)? Are continental applications possible, or are they not possible?
*Response:*
*See above. If the corridors are clever placed yes they are possible and if a more efficient parallelization of the FFT is implemented. In this ms we are not providing a proof for this (we will do in the next where we aim to perform a European simulation). Therefore we do not refer to continental runs anymore. The statement at this point is meant to say that from a memory requirement there is no problem performing the FFTM over large areas.*

Comment:
l.488: plural and singular mix: "a high amounts" + what does "of the FFT as the local simulations"
mean?
*Response:*
*amounts : see calculations above.*
*FFT as local simulation: Currently only one node is taking care of the calculation of the FFT. One could theoretically perform the FFT at each node and use one master node only for collecting the amount of dispersed seeds and performing the communication. Hence there is still some untapped optimization potential.*

Comment:
l.498: what do you mean with "truly mechanistic"? I recommend deleting this statement
*Response:*
*We meant that the migration rates are a result of the dispersal kernel and establishment in a mechanistic way. We agree that the term might be misleading and have removed it.*

Comment:

l.503: "related estimates the Conclusion section"?
*Response:*
*Thanks for spotting that there is an 'in' missing.*

Comment
l.613: Again "10" + something with the formatting
*Response:*
*Thanks for spotting this. It is changed.*

Comment:
l.627: Please provide a legend – even if the figure is only schematic
*Response:*
*We added a legend.*

Comment:
l.635: When looking at the Figure and reading 2.4 I wondered where the 5*10-7 came
from and how this parameter is determined? – Finally I found some information in 3.3
*Response:*
*See responses above. It is a fitted parameter to make the two methods result in comparable
migration speed.*

Comment:
Fig.3 and 4:  I would appreciate if the y-axis of the distance plots on the right would
have similar scales, this would really help for comparison
*Response:*
*We changed the axes.*

Comment:
l.647-648: difficult sentence – maybe: "only taken into account for grid cells ..."?
*Response:*
*Thanks for the suggestion, we reformulated.*

Comment:
l.661: Comparing the dark blue spots in Fig . 2 and the white ones in Fig. 5 the Figures
seem to be mirrored along the diagonal?

*Response:*
*Thanks for spotting this. We checked now and yes it was mirrored.*

Comment : .664: cpu*h = CPU h?
*Response:*
*Changed.*
Comment:
l.671: FFTM with 10: shouldn't this be 64% instead of 67%?
*Response:*
*Thanks for spotting this typo.*

Anonymous Referee #3

Lehsten et al. present a nice and timely study focusing on the implementation of migration into dynamic global vegetation models. They show a way to connect established assumptions of seed dispersal based on former studies with two approaches of enhanced seed dispersal based on Fast Fourier transformation and seed matrix shifting. Together with allowing seed dispersal through specified spatially equidistant corridors they show a nice way of how to reduce computation time while losing some accuracy even though there is no real validation presented. The approach has the potential to be applied in the different DGVMs existing today and is an important contribution to their development. My comments mainly concern 1) the reproducibility of their method and 2) the realism of being able to conduct continental scale simulations.
*Response:*
*Thank you for the effort put in the review of our paper. We will respond and react to your comments as listed below. .*

Comment:
Computation time: My biggest concern here is that the authors simulated only one species migrating. Therefore, they were able to simulate only one growing patch per 1 km2 grid cell. When increasing the species number it is definitively important to also increase the number of growing patches (and probably decrease their size to enable local competition or in other words to avoid an unrealistic growing patch overarching competition which is one of the major selling points of a gap model like in LPJ GUESS typically using 100-1000 m2 growing patches) in return increasing the computation time.
*Response:*
*Actually the whole concept of patches as independent replicates of the vegetation succession at one location is somewhat problematic with our simulation set up.*
*For patches to be independent, there should be no interaction between them. However this is exactly what we want when simulating migration. It seems that the reviewer assumes that we increased the patch size. We did not. The patch size is kept at the standard level of LPJ-GUESS 4.0 of 1000 m2. Similar to all other applications however, this patch represents a larger area. In most applications this patch represents an area as large as the climate grid cell (typically 1 (0.5) degree lon/lat). In our simulation the single patch per gridcell represents an area of 1km2 which is only important for the dispersal kernel.*
*The patches in LPJ-GUESS are mainly introduced to take into account the stochastic heterogeneity of vegetation, that means to decouple the small scale successional stages, to be sure that for example an LAI value for a cell is not too low, simply because the patch is just in an early stage of succession, Hence an averaging of many cells (which have a random disturbance event to restart the succession) will assure that this will not happen. In our case we have only one patch per grid cell, however, in a larger scale simulation one would of course average the output over 0.5 degree and in this case over 200 grid cells with a single patch and hence all successional stages should be present in this 0.5 degree with the correct proportion similar to an application in which a large number of patches would be applied.*

Comment:
The computation time for a continental scale simulation with many different species still has to be determined and could be topic of a follow up study.
*Response:*

*This point has been raised by the other reviewers as well. To run a truly continental simulation with many species is not yet feasible with the presented method - at the current stage of optimisation. However, the main point of the paper was to present the two methods and not to spend a lot of time on optimisation methods. The main time in large scale simulations will be needed not by the simulation of the seed dispersal but by the simulation of the vegetation dynamics given that many cells will have to be calculated. We expect to be able to present continental simulations over the Holocene soon, but have currently no proof for this. Therefore we refrain from mentioning continental scale simulations and refer to 'larger areas'. Basically our tests indicate that the corridors can be placed rather far from each other over the largest areas of Europe, which would allow the continental scale simulations that we refer to. But again since we are not providing data on this we will not mention it in the paper.*

Comment:
Germination rate: It is unclear how sensitive the presented results are in connection to the germination rate used. Obviously the germination rate must influence the speed of migration. The rate of germination directly influences the competitiveness of each species and therefore its dispersal.
*Response:*
*We fully agree with this comment. Here we are using the germination rate used in TREEMIG and compared our results with the results in TREEMIG. In a 'real world' study this should be looked at in detail while here we only want to present the method. We do now mention that this in the Discussion where we changed the title of the section : 'Parameterisation of dispersal kernels ' to Parameterisation of dispersal kernels and other plant parameters'. And we discuss this in Line 534.*

Comment:
Age of maturity: Even though I am totally ok with not taking into account an age of maturity to keep the findings of this study as simple as possible, it is again very obvious that this variable strongly influences the speed of migration. Therefore, this topic needs an extra space in the discussion or some results in the supplement showing e.g. the influence of assuming a minimum age of maturity.
*Response:*
*We agree, that the time a species needs to start reproduction, i.e. generation time, is one of the most*

*important factors influencing migration speed. Instead of fixed age of maturity we use a height*

*threshold for maturity, which makes generation time dependent on growth and thus on environment*

*and competition. We emphasize it in line 568.*

Comments:
Line 132-134: How does the seedbank determine establishment probability and how is environmentally-suitable determined? I would like at least a brief explanation of this crucial aspect.
*Response:*
*While we mention the environmental suitability here, and the probability to establish, we explain the establishment probability in detail in Chapter 2.3.4 Germination.. With respect to the environmental suitability, this is assessed in LPJ-GUESS (and we did not change this part) by using environmental envelopes of some climatological parameters, e.g. minimum temperatures to survive or establish. We now added an inset mentioning this and referring to the LPJ-GUESS publication at line 156- 158.*

Comment:

Line 136 – 150:

So what I see in figure 3 and 4 is that for the 50km corridor approach you have 6 corridors per 0.5 ◦ grid cell (or do neighboring cells share corridors)? And these corridors need 200 simulation cells each of them 1km 2 in size? Assuming that a 0.5 ◦ grid cell is 50x50km I wonder where the positions of your corridors actually are. At the boarders of each grid cell and also diagonal through the middle? It would be helpful to this in Fig. 3 and 4.

*Response: We have inserted a figure (Fig. 3) of the sequence of local dynamics on the corridors, interpolation and dispersal on the grid, where we highlight the corridors (line …) However there seems to be another misconception. We are simulating an area of 100 by 100 km in all simulations. Each time each cell has a size of one by one km. So the cells are completely adjacent. In the simulations with the corridors we are selecting cells (all outside the corridor) for which we do not simulate the vegetation dynamics, but before simulating seed dispersal we interpolate the seed production from neighbouring cells to all cells. Hence in the step of simulating the seed dispersal we have again 100 by 100 cells (adjacent to each other) that produce seeds (though some just have the seed production value from the nearest neighbour). So to come back to your first question, the 100 by 100 km would be roughly a 1 by 1 degree grid cell (in our simulations we need to run LPJ on equal area cells rather than lon lat as usual) and for the 50 km corridors you would have 2 East West, 2 North South, 3 NorthWest – SouthEast and 3 NorthEast –Southwest corridors (lines of cells at which the full vegetation dynamic is calculated).*

Comment:

b) You are able to use only 1 patch per cell, because you are only simulating 1 species migrating. It is important to explain in the discussion that you definitely need more (and probably smaller) patches if you consider more species. It actually scales with species number. Therefore, computation time would be much higher as well. This is contradicting potential continental simulations.

*Response:*

*Testruns have shown that we can have multiple species migrating with a single patch. We cannot see the logic why a single patch can only have a single species migrating, and why the required number of patch repetition scales with species number. In fact the calculation in LPJ-GM simulates the seed dispersal for both simulated species (which probably slows the simulations somewhat), though the birch has free establishment.*

*We agree that more species will require more simulation time. However since this is using LPJ-GUESS and not LPJ it does not scale as strongly with species number since each species can have a multitude of species age cohorts and computation time scales with age cohorts. In a crowded cell there will be less cohorts per species but in general the statement that more species require more computation time holds true. We therefore removed all references to continental simulations in this paper. We hope to present a continental simulation soon in a different paper.*

Comment:

Line 155-157: Even though you cite Lischke et al. I would like to see a brief explanation of the "maximum fecundity" method.

*Response:*

*It is not a method but a single value for maximum fecundity. Basically there is a maximum number of seeds that can be produced per tree of a species for which we have an estimate. This one we multiply with the current LAI divided by the maximum LAI for which we also have a value listed. We now give an example to make this clearer at line 181.*

Comment:

Line 157: Have you performed tests for age of maturity? I guess setting an age of maturity would lower the speed of migration. I am totally ok with not taking this into account, but it would be good to pick up this issue in the discussion e.g. under 4.4.3.

Response:
*As mentioned before we are using height of maturity (see chapters 2.3.1) and now discuss this in the chapter 4.4.3 Parameterisation of dispersal kernels and other plant parameters.*

Comment:
Line 163-164:Please provide explanation and reference for mast fruiting effects.
*Response:*
*We provided an explanation on line 190*

Comment:
Line 188 – 189: So do you use the values for Fagus sylvatica?
*Response:*
*Since the only species that effectively migrates in this paper is Fagus, we do use these values.*

Comment:
*Line 191 – 193: Where can I find values of "loss of germinability"? If these are specific values from Lischke et al. I would suggest to list them in a table as well as similar parameter values. This would really help to reproduce the study.*
*Response:*
*We agree with this and have added one extra table (in the supplementary material S.4) listing all needed parameters needed to reproduce the study.*

Comment:
Line 194-198: I have my problems understanding this whole part. 1. "A year is defined for each species and grid cell before which seed bank constraints are ignored". I do not understand this sentence. 2. I also do not understand the second sentence. I believe you talk about the initial conditions and refugia. It is probably a very crucial part for migration simulations so please provide a few more sentences of explanation.
*Response:*
*We rephrased the sentences to this (and hope it is now clearer):*
*For each grid cell and each year we prescribe whether the species requires seeds to establish. By not requiring seeds in some cells for establishment or not requiring seeds for establishment for some species for all cells we define refugia or in the latter we define that the species' seeds are known to be very far dispersed and hence no explicit simulation of establishment by seeds is required for this species. Technically this is implemented by reading in a list for each cell containing a year from which onwards a species' establishment is not limited by the availability of seeds. Explained in lines 225ff.*
Comment:
Line 207: Explain "age cohort". It has not appeared before and is important to understand the approach.
*Response:*
*The term age cohort comes from the general principles of LPJ-GUESS. Since here we only talk about the establishment of a species we decided to remove the term as it might only confuse the reader if we add a 3 sentence explanation.*
Comment:
Line 199-208:
Please provide equations and an according explanation instead of an example in
*Response:*
*We have added an equation (eq. 5).*
Comment:
b) Moreover, I am quite sure that the germination rate strongly influences results. It is probably important for the speed of migration and definitively for competition and therefore equilibrium biomass. What do you mean with "initial testing"? I don't expect a comprehensive full explanation in the text, but I would like to understand why you have chosen certain parameter values.
*Response:*
*See above, we added a section in the chapter:*

**4.4.3 Parameterisation of dispersal kernels ,and other plant parameters**

Comment:
Line 302 – 303: As suggested above. Please provide the numbers of your parametrization.
*Response:*
*All parameters are listed in an extra table in the supplementary material S4.*

Comment:
Line 442 -443: Is it possible to give a comparison here? What is the computation time for the same setup with standard LPJ-GUESS? I see your comparisons in table 1, but they all refer to simulations which use a master.
*Response:   We have added a simulation without the communication and the difference to the simulation with communication is negligible. We mention this in the paper.*

 Comment:
Line 488 – 490: Have you estimated the CPU hours for this setup? Would be an
interesting information.
*Response:*
*Yes we did. We tested using the FFTM (hence without terrain) using 4000 by 4000 grid cells, running for a few years only.*
 *Looking at the numbers in table 1 shows that running a full scale simulation with 21000 years over the 3463 ½  degree cells that we typically use for European runs would take a long time:*
*1800 (CPUh per 100000 cells and 3000 years) /100000(cells in the MS)\*3463(half degree cells in Europe) \*50\*50(rough estimation of how many 1km cells are in a half degee cell)) \*21000 years in LGM simulation / 3000 years in a test simulation gives us roughly 10 mill CPUh. Given that my current account allows me 45000 CPUh a month, this is currently not feasible and that is why we suggest the transect method. (Actually there might be even more time needed given that there are more than 2 species in the final runs).*
*Again we see quite some opportunities for optimization unused and will hopefully soon provide a continental simulation.*

*Technical comments:*
*Line 22-24: Indicate that this sentence is about plants in the real world*
*Response:*
*We now start the sentence with : Pollen studies have shown that ..... indicating that we are not talking about simulated species anymore.*

Comment:
Line 62-64: For me this is one of the major selling points and I would put it in the abstract as well. You decide.
*Response:*
*Thanks for the suggestion. We would love to do this and in fact we just started a project where we look at the spread of alien species, however what we present here is only considering tree migration. For trees the spread by adjusting to climate change is very slow given the long generation time. We will hopefully soon have a model in which we simulate tree pests spread, there*

*this applies, but in this publication we only consider trees hence the response to current climate change is rather of second order.*

Comment:

Line 202: 2 times the word "depending"

*Response:*

*Thanks for spotting this. We removed it.*

Comment:

Line 203 -205: Confusing sentence. "The probability that a species establishes is proportional to the seed number in the seed bank multiplied by . . .". Wrong formulation.

*Response:*

*We now added an equation instead of this sentence.*

Comment:

Line 206: The word "year" is missing.

*Response:*

*No, the word 'per' was obsolete. Thanks for spotting this. We removed it.*

Comment:

Line 242: I would not expect that every reader knows what a Moore neighborhood is?

*Response:*

*We now added" i.e. the surrounding eight cells".*

[revised manuscript text omitted]

---

## Author Response (AR2)

**Author's response to reviewer comments to: LPJ-GM 1.0: Simulating migration efficiently in a dynamic vegetation model and tracked changes document.**

Referee 1

Suggestions for revision or reasons for rejection (will be published if the paper is accepted for final publication)

Overall Comments:

In general, I found the response to reviewers to be thorough and the authors comments to be informative. Once the authors address the remaining minor comments, this paper will be ready for publication.

Minor comments

Comment:

Throughout – the text needs to be checked for grammar and clarity. In general, check the uses of singular versus plural (e.g., "is" versus "are", L45), the use of "that" or "which" (e.g., L111, L112), unnecessary "the" (e.g., L78), and the use of present versus past tense. Methods and Results should be written in past tense, not present (e.g., L366, L367, should be in past tense).

Response: We went through the whole text and changed the tense as well as the taking care of singular versus plural.

Comment:

Places where the text needs to be revised to improve clarity:

L56 – replace "to have a sufficient amount of seeds at a given location to successfully establish" with "to have seeds present at a given location …"

Response: "they also need to have a sufficient amount of seeds present at a given location" since we want to highlight that it is not enough to just have seeds at the location but the amount needs to be sufficient.

Comment:

L64-67 – please rephrase. These two statements are difficult to understand.

Response:

Rephrased to "The implementation of migration into dynamic vegetation models is not only of interest for the simulation of historical species ranges, it is also of interest for the projection of ecosystem properties in the future since  migration lags might lead to uncertainties in projected ecosystem
properties if the wrong species community is predicted to occur at a certain site (Neilson et al., 2005)."

L76-78 – this statement needs a reference.

Response the statements here are discussed in Snell et al. 2014 and we added the reference.

L110 – please include … "between patches within each grid cell" to the end of this sentence.

Respone:

We added this part.

L114 – please consider rephrasing this "allow simulating species migration of several species
simultaneously", as it is awkward.

Response: we rephrased it to:

"However, to the knowledge of the authors, there is no implementation of a migration scheme into a
DGVM which allows simulations with a large extent, takes migration within the grid cell into account
and includes feedbacks between all simulated species. "

L121 – 123 – I would delete this sentence, as it is pretty vague and doesn't help a reader understand
LPJ-GUESS. Especially as much more useful information is presented subsequently.

Response: we deleted the sentence.

L140 – not at the end of every year? What is a "migration year"?

We tried to speed the computing up by only producing seeds after the first 100 years, where we
simulate vegetation with no seed or N-limitation (described in the paragraph below). We agree that
the term migration year is missleading. We changed the sentence to :

"Seeds are produced potentially in each grid cell at the end of each year after the first 100 years (see
below)"

L161 – "are simulated", needs to be deleted from this sentence.

Response: we deleted the words.

L166 – 173 – this is an example, of where the explanation found in the response to reviewers was much clearer, than what is in the manuscript. Please rephrase this. Also, it is not clear how Figure 3 explains the comment, "LPJ-GM represents a 0.5 x 0.5 degree cells with 200 simulation cells". Shown is an 11 x 11 box, which is 121 grid cells, of which only a subset are actually simulated.

Response: We added a paragraph which gives an overview of the steps involved in the calculation of the migration.

"We demonstrate this in Fig. 3 where a single 11 km by 11 km large grid cell is separated in to 11 by 11 smaller grid cells with similar climate. The local dynamics and seed production is only simulated along the transects (grey or green cells in left panel of Fig.3). As a next step the seed production is interpolated onto all cells for which no local dynamics, was calculated and the seed dispersal is simulated. Finally, seedling establishment is simulated, but only in the grid cells on the corridors (more details for the different steps are given below). "

L223 – please add in the value for loss of germination.

Response: all values are given in the supplementary material. To keep matters simple we refer to Lischke et al. 2006, from which we took the Seed bank dynamics. And instead of explaining it in detail with equations we simply list the parameters (which are equal to the ones in Lischke et al. ) in the supplementary material. We now refer explicitly to the supplementary material in this sentence.

L225 – 227 – this statement is confusing and needs to be rephrased.

Response: we rephrased the sentences to:

For each grid cell and each year we prescribe whether the species requires seeds to establish. By not requiring seeds for establishment we define refugia, or we define that the species' seeds are known to be very far dispersed and hence no explicit simulation of establishment by seeds is required for this species.

L237 – The choice of pi as a symbol in this equation is confusing. To improve clarity and understanding, please use a different symbol.

Response: We changed the symbol to P.

L238 – "seed number" or do you mean "number of seeds"?

Response: Yes we changed this

Figure 1 – the coloured scales need to be labeled, and units included. Is this number of seeds?
Probability of dispersal? Why does one panel go from 0 – 1000, and the other go from 0 – 14 (x 10-
3)?

Response:

As there was limited place available, we added this information in the title.

L292 – instead of referring to grid sizes in degrees latitude/longitude, it would be helpful to mention
size (to make it easier to compare to the following statement about maximum dispersal distances of
200 m).

Respone:

We added that the 200m would be approximately 0.002 degree longitude latitude(at the Equator).

L318 – dispersal ability is not the correct term. This implies the ability of the tree to disperse. Perhaps
"differences in available habitat", or "differences in barriers to dispersal"?

Response: this does not represent "differences in available habitat", but different dispersal kernels at
different points of the landscape, or barriers in the landscape. Hence dispersal ability is the correct
term here.

L321 – seeds can reach those areas, however if they do reach it – they don't germinate. Perhaps this
confusion is between biology (i.e., in reality, these areas would be a city or a parking lot – seeds
arrive but don't germinate because it is not suitable), and programming (i.e., in the model, it is
simulated as seeds not arriving).

Response: No it seems the reviewer did not understood the set up. Seeds can NOT reach these areas.
We have set the dispersal ability (kernel width) to zero the whole point of this simulation is to show
the effect of spatially varying dispersal kernels. The cells in  the middle of the figure have the same
climate as mentioned in the text as the surrounding cells. To stick with the images given by the
reviewer, it is not a parking lot where seeds arrive but can not establish, but it is a greenhouse which
would have good growing conditions but seeds can not enter as the glass wall does not let them
pass. Hence it is exactly as we describe it in the text.

All figures – Scales are missing a label. X and Y axes should be capitalized.

Figure 1 we added the labels, Figure 2 displays seed permeability, hence the scale has no unit.

Figure 5 and 6 Each left panel figure is labelled Year of arrival and the scale displays the year. This
should be sufficiently clear. If we would add 'year of arrival' as a caption to the scale the place would
not suffice.

We capitalised all X and Y labels.

L325 – "local dynamics on the corridors" is unclear. Please clarify you mean that you are simulating
vegetation successional dynamics.

Response: we clarified this already in the description of the figure before, and now we also changed
the figure description to vegetation successional dynamics.

L336 – 337 – this sentence is unclear - "neglected the points within the first 5 km". Does this relate to
the starting location? Or the ending position, if they migrated < 5 km?

Response: It relates to all points within 5km from the starting location. We added this information.
The sentence now reads:

"To avoid founder effects we neglected all points within first 5 km from the starting location (the
refugium)."

L372 – 373 – this statement (and subsequent statements, e.g., L394-395, 427) about the reduction or
increase in computing time by percentages is unclear. Table 1 is very helpful and clear, but these
summary statements are not. Perhaps because the Table includes + and – so it is clear if the total
times were increasing or decreasing? A reduction BY 88% is different than a reduction TO 88%.

We went through the statements and made sure that we correctly used the words TO and BY.

L392 – please clarify here, that the migration rate is reduced when you use corridors.

Response: We added that the decrease in migration rate is caused by the corridors.

There are numerous statements in the discussion which are awkward, and need to be edited for
clarity. E.g.,

L448, L460 – 461, L463 – 464, L482 – 484, L516 – 518, L531-532, L568-570, L575 – 578,

Response: We went through the mentioned passages and adjusted them and hope that they are now
clearer.

L492 – The approach by Snell (2014) did not use a reduced number of patches. There were 400
patches per grid cell.

Response: We changed this sentence to:

" Snell (2014) approached the discretization problem for the DGVM LPJ-GUESS by   assuming that
the numerous replicates of the vegetation dynamics on a patch are randomly distributed over the area
of the grid cell (using 400 patches)."

###############################################################################

Referee 2

I appreciate the steps taken to clarify the applicability of LPJ-GM 1.0.

Unfortunately, it is very difficult to follow the tracked changes. In particular in section 4.3 and 4.4.3,
but e.g. also in section 1 and 2, several paragraphs are marked as new/old, although it appears to be
exactly the same text as before).

This makes a reassessment more difficult than necessary, and it particularly suggests much more
changes than the authors have really applied!

Response: As several authors were writing in the same version, and some changes needed to be
accepted for the author making the changes to see that his suggestions were accepted, we generated
a new track changes version using Words internal document comparison tool, where we compared
the original word document which was submitted before to the version with all the changes. We
expected that this procedure should assure that all changes and only the changes are highlighted we
are sorry if this was not the case. This time we did not use the document comparison tool but only
the track changes and did not successively accept changes.

Nevertheless, in my opinion most comments have been addressed and I am mainly left with one
important aspect:

From the text provided in the manuscript I still would not be able to fully really assess when and how
to apply which algorithm:

1. In 2.7 the authors refer to the performance test of the two methods SMSM and FFTM in in
Supplementary 2, showing that "SMSM is still up to an order of magnitude slower than the FFTM".
They pronounce that the implementation is in a different environment, but state: "However in a
general sense we can see no reason why they should not reflect the performance differences
between the algorithms."

Given this, when looking at the last Figure in Supp 2, I do not understand how the computation time
for SMSM in Table 1 can nearly equal those of FFTM (and even perform better with transects than
FFTM).

Response: We agree with the referee that this is counterintuitive and we should have picked that up
in the first place. We went through all original files and looked at the time stamps to assure that the
values in the table are correct. They are correct. According to the Matlab documentation, the fft
calculation of Matlab is based on the same the fftw library (Matlab manual links to fftw.org),
therefore we assumed that the running times should be comparable. Apparently they are not. Either
the FFTW implementation used by Matlab is much faster (better optimized) than the way that we
implement this in C++ (relative to SMSM), or the SMSM implementation in Matlab is slower (which is unlikely). One could also think that the actual calculation is rather unimportant and the majority of
time is needed for the actual seed transfer between gridcells, but this can also not be the case since
in the last line of table 1 we simulated the same amounts of seed transfers between cells as in the
other ones.

We are now discussing this in the discussion section and suggest that the reader only looks at the
relative increase of computation time since the differences between the different methods are not
represented in the computation time demands on the computing cluster.

2. In the responses to the review the authors state "A simulation in which you want to allow 20 cells
maximum distance require twice as much computation time for the SMSM. For long tailed species
the FFTM is certainly better suited, given that this method has no such limitations." -- I would
strongly recommend to make this also explicit in the discussion (e.g. add this to first paragraph of
4.2).

Response: We have added this now to the first paragraph. The added sentence is as follows:

Additionally the SMSM restricts the long tail of the distributions by the number of iterations, as the
seeds can travel only travel one grid cell per iteration step.

3. In the responses to the review the authors state "the idea is to only use the corridors in
homogeneous landscapes and to speed up the simulation there. In heterogeneous landscapes this
simplification is not suitable."

I would appreciate if this is also stated that explicitly in the discussion.

In addition, thinking about this statement of the authors I really wonder why the SMSM is shown
with transects at all, since its only preferable for heterogeneous areas and in these it cannot use
transects?

I would also appreciate if it would be explicitly stated in the text (e.g. 4.2) that SMSM with
heterogeneous area as in Fig.6 cannot be simulated with transects.

Response: We covered these points in the discussion in a new paragraph which answers the
questions.

"The two approaches that we present differ in their ability to simulate heterogeneous landscapes (in
terms of permeability). We suggest using the FFTM with corridors in homogenous landscapes (to
speed up the computation) and to use the SMSM without in heterogeneous landscapes. In cases where parts of the domain are heterogeneous (e.g. the regions around a mountainous area) and other parts of
the domain are homogenous (e.g. lowlands), the cells can be arranged in a way that they cover the
whole area in the heterogeneous part and only corridors in the homogenous part. In this setting the
SMSM can still be used for the whole domain and an improvement of computation time can be
achieved by only simulating the local vegetation dynamics in the homogenous parts of the domain."

4. Although not fully comparable with the other simulations, it would be interesting to have the
simulation depicted in Fig.6 also listed in Table 1, or at least to have mentioned the computation
time required for this simulation in 3.3.

What was the computation time required for the simulation with SMSM and the non-homogeneous
dispersal area?

Response: we added that information in the text.

==

Additionally, I list a couple of minor comments/suggestions, mainly typos/issues in single sentences:

l39: "where the local dynamics is simulated" -> are?

Response: changed.

l55: check sentence ".. replace existing vegetation – the processes gap models describe successfully –
but they .." Maybe "vegetation – that/which"?

Response:

Changed to:

However, in real ecosystems species need not only to establish and replace existing vegetation, which
the processes in gap models describe successfully, but they also need to have a sufficient amount of
seeds present at a given location to successfully establish.

l94-96: check grammar/sentence: "If it is assumed in contrast that the simulated forest is uniformly
distributed in the cell, with each time step some seeds reach the neighbour cell, leading to a
resolution dependent speed up of migration."

Response: Changed to

If, on the other hand, a uniformly distributed forest in the cell is assumed in the simulation some seeds reach the neighbour cell with each time step, leading to a resolution dependent speed up of migration.

l109: "while Snell et al. (2014)" its --> Snell (2014) not Snell et al. (2014)

Response: Changed.

l122 and

Response:

This sentence has been deleted on the request of the other referee.

l151: I still do not agree that the temporal resolution of (tree species) seed production is necessarily annual (particularly not in the tropics), and suggest to e.g. stop after "annual basis.", or add "in temperate forest", or similar (again see e.g. Owens 1994, Brokaw, 1998))

Response: We stop the sentence after "annual basis".

l157: "temperatures; see; Smith et al. 2001" -> "temperatures; see Smith et al., 2001"?

Response: changed.

l161: check sentence: "previous applications simulate a certain number of replicate patches are simulated per grid cell"

Response: We removed the 'simulated per'

l163: "1000m2" -> "1km2" for consistency

Response: It would have to be changed to 0.001 km2, we consider 1000m2 easier to envision.

l177: introduce LAI abbreviation when used first

Response thanks for spotting that we missed the brackets in the sentence which indicate that LAI

stands for the acronym. We added them.

l179-180: why do you cite (Lischke et al., 2006) here? there is no LAI in that paper

Response: We removed the reference.

l218: "( 0.99" space

Response: Thanks for spotting this.

l250: "rarely"? ++ missing bracket ++ order of references?

Response: Thanks for spotting this.

l392: "...for simulations with transects"

Response: added.

l483: "which resulted a computing time" -> "... resulted in a ..."

Response: we changed the sentence to.

This method led to a computing time reduction of 30-85%  compared to the full simulation similar to
our transect methods (which resulted a computing time reduction in a similar range depending on the
configuration of the corridors).

l562: "TreeMig ;Lischke" -> "TreeMig; Lischke"

Response: Thanks for spotting this.

l578: check sentence: "where alien species a planted."

Response: We removed that part.

l585: still mix of plural and singular "require a relatively high amounts"

Response: Thanks for spotting this.

Fig.5: I would appreciate if the y-axis of the distance plots on the right would have similar scales, this
would really help for comparison

Response:

We fixed that.

Kruse, S., ..., in review, 2018.

=> published during the mean time

Response: We adjusted the references.

Table 1: still has 67% instead of 64% for FFTM with 10

Response Sorry this must have gotten lost in the last revision. It is correct now.

Supplementary 3

- readability: sentences are cut onto two pages

Response: we added a page break.

- should be 931.6 not 9317

Response Changed.

[revised manuscript text omitted]